# Exploring Semantic-constrained Adversarial Example with Instruction Uncertainty Reduction

Jin Hu[1,2]    Jiakai Wang[2✉]    Linna Jing[1]    Haolin Li[3]    Haodong Liu[3]
Haotong Qin[4]    Aishan Liu[1]    Ke Xu[1,2]    Xianglong Liu[1,2]

[1]State Key Laboratory of Complex & Critical Software Environment (CCSE), Beihang University
[2]Zhongguancun Laboratory    [3]School of Computer Science and Engineering, Beihang University
[4]Department of Information Technology and Electrical Engineering, ETH Zurich
hujin@buaa.edu.cn   wangjk@mail.zgclab.edu.cn

## Abstract

Recently, semantically constrained adversarial examples (SemanticAE), which are directly generated from natural language instructions, have become a promising avenue for future research due to their flexible attacking forms, but have not been thoroughly explored yet. To generate SemanticAEs, current methods fall short of satisfactory attacking ability as the key underlying factors of semantic uncertainty in human instructions, such as *referring diversity*, *descriptive incompleteness*, and *boundary ambiguity*, have not been fully investigated. To tackle the issues, this paper develops a multi-dimensional **ins**truction **u**ncertainty **r**eduction (**InsUR**) framework to generate more satisfactory SemanticAE, *i.e.*, transferable, adaptive, and effective. Specifically, in the dimension of the sampling method, we propose the residual-driven attacking direction stabilization to alleviate the unstable adversarial optimization caused by the diversity of language references. By coarsely predicting the language-guided sampling process, the optimization process will be stabilized by the designed ResAdv-DDIM sampler, therefore releasing the transferable and robust adversarial capability of multi-step diffusion models. In task modeling, we propose the context-encoded attacking scenario constraint to supplement the missing knowledge from incomplete human instructions. Guidance masking and renderer integration are proposed to regulate the constraints of 2D/3D SemanticAE, activating stronger scenario-adapted attacks. Moreover, in the dimension of generator evaluation, we propose the semantic-abstracted attacking evaluation enhancement by clarifying the evaluation boundary based on the label taxonomy, facilitating the development of more effective SemanticAE generators. Extensive experiments demonstrate the superiority of the transfer attack performance of InSUR. Besides, it is worth highlighting that we realize the reference-free generation of semantically constrained 3D adversarial examples by utilizing language-guided 3D generation models for the first time.

## 1  Introduction

Adversarial example (AE), showing that small perturbations can impact the performance of deep learning models, is broadly focused due to its potential to promote model robustness and secure applications in practice. A series of studies has uncovered several forms of AEs, including physical-world AEs [1, 2, 3, 4], transfer AEs [5, 3] and naturalistic AEs [6, 7], as well as the applications in evaluating real-world recognition [8, 9], autonomous driving [10, 11, 12] or LLM systems [13, 14].

While most adversarial example research focuses on finding AEs around existing data, generating AEs from natural language instructions without referenced data has not yet been thoroughly explored, *i.e.*, to find *Semantic-Constrained Adversarial Examples* (SemanticAE). Specifically, given a certain

39th Conference on Neural Information Processing Systems (NeurIPS 2025).

natural language description, we aim to generate the data that corresponds to its real semantic meaning but is hardly to be correctly recognized by deep learning models trained in related tasks. Recent works have employed techniques related to naturalistic AEs to accomplish a similar objective [15, 16, 17], , but the de facto potential of SemanticAE has still not been fully released in performing transferable, adaptive, and effective attacks, limiting the applicability. In light of the recent advancements in language-driven multimodal intelligence and the increasing demand for alignment [18, 19, 20], it is necessary to take a step further in SemanticAE generation and facilitate more versatile AE generation.

To push the boundary of the current technology, we focus on the key underlying factor limiting the adversarial capability of SemanticAEs: the inherent uncertainty within human instructions that defines semantic constraints. We categorize three major forms of uncertainty in instructions: ❶ *Referring diversity* introduces a barrier in SemanticAE optimization via the multi-step generative models, since it leads to the inconsistent language-guidance that the adversarial optimization should collaborate with. ❷ *Descriptive incompleteness*, which conceptualizes the gap between the precise model of the attack scenario and the instructions given by potential users, restricts the application scenarios. ❸ *Boundary ambiguity* of the semantic constraint is hard to characterize in task definitions, affecting the evaluation of SemanticAE generators.

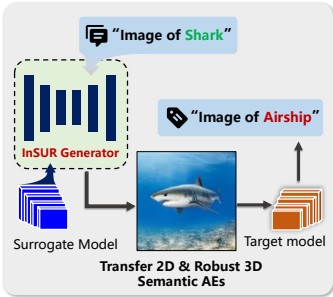

Figure 1: SemanticAEs are generated directly by instructions.

We propose a multidimensional **ins**truction **u**ncertainty **r**eduction (**InSUR**) framework to tackle the issues and generate more transferable, adaptive, and effective SemanticAE. Specifically, for referring diversity, we propose residual-driven attacking direction stabilization via the novel ResAdv-DDIM sampler that stabilizes optimization through coarsely predicting the language-guided sampling process, releasing the capability of multistep diffusion models on adversarial transferability and robustness. For descriptive incompleteness, we propose the context-encoded attacking scenario constraint for both 2D and 3D generation problems by scenario knowledge integration, tackling the scenario adaptation problem by addressing the descriptions' incompleteness problem, achieving the first 3D SemanticAE generation. For boundary ambiguity, we propose the semantic-abstracted attacking evaluation enhancement based on label taxonomy. Our contribution can be summarized as:

- We conceptualize the SemanticAE generation problem and propose a multi-dimensional instruction uncertainty reduction framework, InSUR, to address the challenges.
- In the dimension of the sampling method, we propose the residual-driven attacking direction stabilization to achieve better adversarial optimization. In task modeling, we propose the context-encoded attacking scenario constraint to realize scenario-adapted attacks. In generator evaluation, we propose the semantic-abstracted attacking evaluation enhancement to facilitate the development of SemanticAE generators.
- Extensive experiments demonstrate the superiority in the transfer attack performance of generated 2D SemanticAEs, and for the first time, we realize the reference-free generation of 3D SemanticAE by utilizing language-guided 3D generation models.

## 2   Backgrounds

**Adversarial Example Generation**   Adversarial attack generating algorithms can be categorized as iterative optimization in the data space, e.g. *FGSM* [21], *PGD* [22], *AutoAttack* [23], iterative optimization in the latent space of generative models, *e.g. NAP* [24], *AdvFlow* [25] and *DiffPGD* [7], training a neural network for generation [26, 2, 27], and applying zero-order optimizations [28, 9]. Adversarial examples may not be robust in the physical world. For physical attacks, expectation-over-transformation (EoT) [29] and 3D simulation [30, 31] are proposed to bridge the digital-physical gap. Research has also focused on attacking unknown models, *i.e*, transfer attacks [5, 32]. An extended technical background and related works are provided in Appendix A.

**Semantic-constrained Adversarial Example**   [33] first proposes the *Unrestricted Adversarial Example*(UAE), of which the restriction is defined by the human's cognition instead of $l_p$-norm on existing data, and proposes a generative learning method for its generation. A line of studies further develops optimization techniques in latent spaces [34, 24, 35], and terms it as *Natural Adversarial Example* (NAE), while another line of study focuses on constructing the perceptual constraints for UAEs. A difference between them is that NAE studies also focus on generating

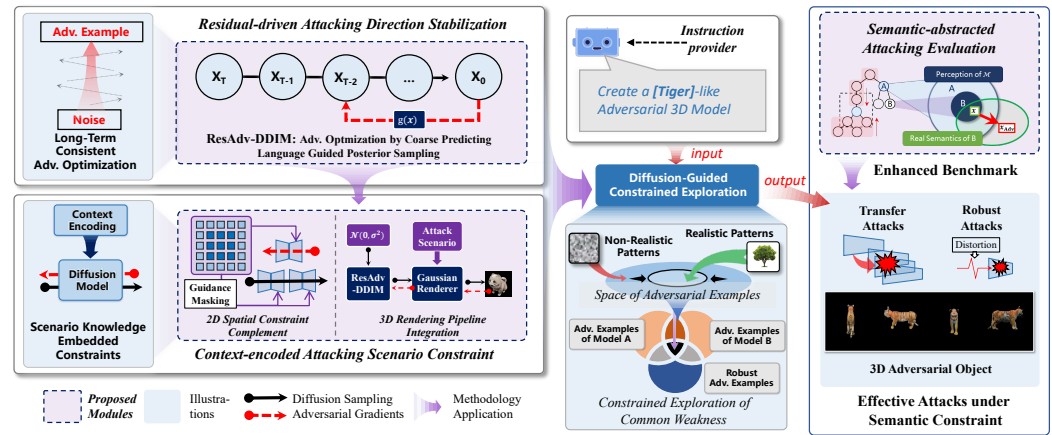

Figure 2: Overview of multi-dimensional instruction uncertainty reduction (**InSUR**) framework.

diverse-distributed adversarial examples without referencing a static image. We formulate the adversarial example generating task constrained by natural language's semantics as the *semantically-constrained adversarial example generation* problem. Recent works ([7, 35, 36, 16, 17]) focus on integrating the pre-trained diffusion model and iterative optimization to constrain the naturalness and improve transferability. Furthermore, generating 3D adversarial examples that are more aligned with the physical world and satisfy the semantic constraints is still an open problem.

## 3 Methodology

### 3.1 Problem Formulation and Analysis

**Semantic-Constrained Adversarial Example (SemanticAE) Generation Problem** We define SemanticAE generation problem as generating an adversarial example $x_{\text{adv}}$ that fools the target model and satisfies the semantics constraint defined by the user's instruction Text. Formally, we formulate the SemanticAE generation problem as follows:

$$find\ x_{\text{adv}} \in \mathcal{S}(\text{Text})\ \text{s.t.}\ \mathcal{M}(x_{\text{adv}}) \in A_{\text{Text}}, \tag{1}$$

where $\mathcal{S}(\text{Text})$ is the set of data with semantic meaning corresponding to Text, $\mathcal{M}$ represents the target model, and $A_{\text{Text}}$ defines the types of target model's output that are conceptually different from Text, representing a successful attack. In the strict black-box setting, which is the focus of this paper, both $\mathcal{S}$ and $\mathcal{M}$ are unknown to the generation algorithm.

The goals of SemanticAE generation are *to build a red-team model $\mathcal{G}$ that automatically finds the alignment problem between the intelligent model $\mathcal{M}$ and the implicit semantics $\mathcal{S}(\text{Text})$ reflecting the social consensus or the physical world.* This goal leads to the following constraints: firstly, to achieve automatic alignment with limited supervision, the instruction Text is not required to characterize semantic constraints precisely. Secondly, from the perspective of data value, the generated SemanticAE $x_{\text{adv}}$ should be able to perform transfer attacks.

**Challenges in SemanticAE Generation** As shown in the middle card of Figure 2, generative or diffusion models can constrain the pattern of generated AEs and facilitate transfer attacks with better in-manifold constraint [37]. We take a step further in SemanticAE, focusing on the inherent challenge related to instruction uncertainty: ❶ Reference diversity challenges adversarial optimization. The language guidance that is learned from the mapping between Text and $\mathcal{S}(\text{Text})$ is non-linear since $\mathcal{S}(\text{Text})$ is diverse. This makes collaborating with adversarial optimization and the diffusion model for better transfer attacks and robust attacks a non-trivial problem. ❷ Descriptive incompleteness requires scenario-knowledge integration for scenario-adapted generation. The challenges are identifying the missing contexts in pretrained models and establishing practical knowledge embedding methodologies, thereby further eliminating the reference diversity from the external perspective. ❸ Boundary ambiguity makes defining $\mathcal{S}$ and $A$ for evaluating the generator also challenging, which lies in the fact that inappropriate evaluation leads to inaccurate results.

**Multi-dimensional Instruction Uncertainty Reduction (InSUR) Framework** As shown in Figure 2, for the reference diversity problem, we propose the residual-driven attacking direction

stabilization with the designed ResAdv-DDIM sampler. For the contextual incompleteness, we propose the context-encoded attacking scenario constraint methods for scenario-knowledge integration in representative 2D and 3D SemanticAE generation tasks. Moreover, since the unclear semantic boundary makes evaluating the generator difficult, semantic-abstracted attacking evaluation enhancement is proposed to facilitate further developments of SemanticAE generation.

## 3.2 Residual-driven Attacking Direction Stabilization with ResAdv-DDIM

**Semantic-constrained Optimization Problem** Referring to the generative-model-based adversarial examples, we solve SemanticAE generation by maximizing the loss $\mathcal{L}_{\text{ATK}}$ under the constraint of the posterior sampling process defined by the natural language guidance Text. However, if the posterior sampling process is more complex, *e.g.*, multi-step diffusion de-noising, tackling this maximization problem is challenging. Recent work utilizes a deterministic sampling process, *e.g.*, DDIM [38], as a constraint defined by Text [7, 15]. For simplicity, we denote the sampling step as $f_{\theta,\Delta T}(x_t) \sim q_\theta(x_{t-\Delta T}|x_t, x_0, \text{Text})$, and such optimization can be formulated as:

$$\max \mathcal{L}_{\text{ATK}}(\mathcal{M}(\underbrace{f_{\theta,\Delta T} \circ f_{\theta,\Delta T} \circ \cdots \circ f_{\theta,\Delta T}}_{T/\Delta T \text{ times}}(x_T))), \tag{2}$$

where $\circ$ denotes function composition. However, this may trigger the robust problem of $f$, since it is hard to determine whether $x_0$ is an adversarial example of $f$ or $\mathcal{M}$. This results in the instruction misalignment of SD-NAE shown in section 4.4. Also, this optimization is computationally expensive. Another solution is to tackle the challenges by approximating the gradient [7] or directly altering the sampling process [16], which can be re-formulated as:

$$x_{t-\Delta T} = f_{\theta,\Delta T}(\arg \max_{x_t'} \mathcal{L}_{\text{ATK}}(\mathcal{M}(x_t'))), \ \forall t \in [\Delta T, t_s), \tag{3}$$

where $t_s$ is a selected intermediate step, and the $\max_{x_t}$ optimization could be a single iteration. An advantage is that, since the maximization algorithm does not retrieve the information of $f$, or $f$ has been protected from adversarial attacks, it alleviates the robustness problem of $f$. However, the optimization used $\nabla_{x_0}\mathcal{L}_{\text{ATK}}$ to approximate $\nabla_{x_t'}\mathcal{L}_{\text{ATK}}$, and the inconsistency introduces noise to the optimization. As shown in Figure 3, the optimization direction may vary, or be non-linear, with respect to different $x_t$. This misalignment makes the adversarial pattern optimized in the initial denoising stage ineffective in the latter stage, thereby limiting the multi-step regularization opportunity of diffusion models for better transfer attacks.

Overall, this technical challenge originates from the conflict between (1) the accurate estimation of $\nabla_{x_t}\mathcal{L}_{\text{ATK}}$, causing the robust problems of the language guidance defined by $f$ and the computational problems, and (2) the approximated estimation of $\nabla_{x_t}\mathcal{L}_{\text{ATK}}$, causing the non-optimality of attack optimization. We solve the problem by improving the approximation with the novel *ResAdv-DDIM* posterior sampler, enabling the discovery of more robust adversarial patterns with better regularization.

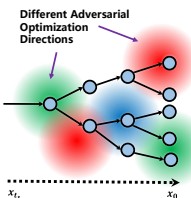
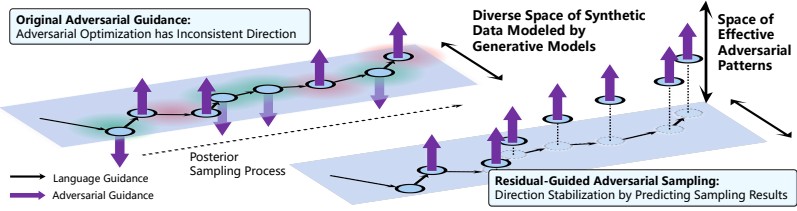

Figure 3: Inconsistent adv. direction problem.

Figure 4: Residual-driven attacking direction stabilization. *ResAdv-DDIM* is designed for the efficient and thorough exploration of new adversarial patterns constrained by multi-step sampling processes.

**Residual-Guided Adversarial DDIM Sampler** Inspired by *Learning to Optimize* [39], we handle the challenge by **predicting a coarse sketch of the future-step denoising result $x_0$** for estimating the attack optimization direction with $\mathcal{L}_{\text{ATK}}$. Our key insight is that since multi-modal models acquire general capabilities through training in the task of *predicting diverse human responses* and *complementing missing information across diverse data*, we should fully leverage the model's intrinsic multi-granularity predictive capabilities to achieve stable generation under semantic uncertainty.

Specifically, we leverage DDIM's multi-step posterior sampling capabilities to achieve a coarse prediction of $x_0$ from current $x_t$, *i.e.*, $g_\theta(x_t)$, which allows for a more accurate estimate of $\nabla_{x_t}\mathcal{L}_{\text{ATK}}$ compared to directly using $\nabla_{x_0}\mathcal{L}_{\text{ATK}}$. We formulate the generation process as:

$$g_\theta(x_t) = \underbrace{f_{\theta,\Delta T_1} \circ f_{\theta,\Delta T_2} \circ \cdots \circ f_{\theta,\Delta T_k}}_{k \text{ times}, \, k \ll T/\Delta T}(x_t), \text{ where } \sum_{i=1}^{k} \Delta T_i = t \tag{4}$$

$$x_{t-\Delta T} = f_{\theta,\Delta T}(\arg\max_{x_t} \mathcal{L}_{\text{ATK}}(\mathcal{M}(g_\theta(x_t)))), \; \forall t \in [\Delta T, t_s].$$

The notation is the same as Eq 2. $g_\theta(x_t)$ is the coarse estimation of $x_0$, and $k$ is a small number of iterations that could be selected from $\{1, 2, 3, 4\}$. Since the sampling process takes a residual shortcut to $x_0$, we name it as *Residual-Guided Adversarial DDIM Sampler* (**ResAdv-DDIM**).

To further establish the concrete adversarial attack algorithm, we further propose the following method. **(1) Constraining the Semantics**. To ensure the generated sample satisfies $x_{\text{adv}} \in \mathcal{S}(\text{Text})$, we constrain the discrepancy of the sample trajectory with the $l_2$-norm between DDIM-generated samples and the adversarially optimized samples after determining $x_{t_s}$:

$$||\text{Denoise}_{\text{DDIM}}(x_{t_s-\Delta T}) - \text{Denoise}_{\text{Adv}}(x_{t_s-\Delta T})||_2 < \epsilon. \tag{5}$$

Such constrained optimization problem could be performed by simultaneously sampling with $\text{Denoise}_{\text{DDIM}}$ and $\text{Denoise}_{\text{Adv}}$, and clipping $x'_t$ in each step. **(2) Adaptive Attack Optimization.** To solve the maximization in Eq. 4, we introduce an early-stop mechanism that terminates optimization:

(1) after the first iteration, and the estimated probability of unsuccessful attack $< \xi_1$, *or*,

(2) at the first iteration, when this probability satisfies a stricter threshold $\xi_2 < \xi_1$.

The probabilities are estimated from $\mathcal{M}(g_\theta(x_t))$. These conditions reduce the expected number and lower bound of optimization steps, while maintaining attack performance alongside the denoising step, which degrades the adversarial capability. Lower threshold could result in better attack performance and slower optimization, and we set $\xi_1 = 0.1$, $\xi_2 = 0.01$ as a feasible setting across all experiments. In addition, we integrate momentum optimization, introduced in [17, 5], to improve the attack transferability. Detailed implementation and analysis are shown in Appendix B.1.

### 3.3 Context-encoded Attacking Scenario Constraint for 2D and 3D Generation

In the application scenarios, the instruction Text might be ambiguous or incomplete, which requires integrating learned guidance with external knowledge. For effective task adaptation, we provide knowledge embedding strategies on the key data structures that collaborate with the ResAdv-DDIM sampler, achieving better 2D SemanticAE generation and realizing 3D SemanticAE generation.

**Spatial Constraint Complement for 2D SemanticAE Generation**   We focus on the problem of the incompleteness of the spatial constraint given by the instruction Text representing the object label. Specifically, an effective SemanticAE generator shall leverage the optimization space of the image backgrounds and generate patterns that amplify the attack's effectiveness. However, the diffusion model's conditional background generation is overly uniform because the attack functionalities were not considered during the original training. To solve the problem, we encode the context of the attack application through the fine-grained control of the denoising guidance. We leverage the application of guidance-masking from edit area control of single guidance [40] to the re-distribution of multiple guidance, and embed the masking into the key guidance function $f_\theta$ applied in both posterior sampling and adversarial optimization process in ResAdv-DDIM (as shown in Figure 2). Together with the deterministic DDIM, $f_\theta$ is formulated as:

$$f_{\theta,\Delta T}(x_t) = \sqrt{\bar{\alpha}_{t-\Delta T}/\bar{\alpha}_t} \left(x_t - \sqrt{1-\bar{\alpha}_t}\epsilon_\theta(x_t, t)\right) + \sqrt{1-\bar{\alpha}_{t-\Delta T}} \cdot \epsilon_\theta(x_t, t),$$
$$\epsilon_\theta(x_t, t) = (1-M) \cdot \epsilon_{\theta,\text{Unconditional}}(x_t, t) + M \cdot \epsilon_{\theta,\text{Conditional}}(x_t, t, \text{Text}), \tag{6}$$

where $\alpha$ defines the noise ratio in the diffusion model, $\epsilon_\theta$ is the noise estimating network, and $M$ is the guidance masking that regularizes the spatial distribution of semantic guidance Text. Detailed definition of $M$ is in Appendix B.1, and the integration is illustrated in Figure 2.

**Differentiable Rendering Pipeline Integration for 3D SemanticAE Generation**   3D Data is valuable for world modeling [41, 42]. We focus on the problem of generating 3D SemanticAE $x_{adv}^{(3D)}$ for target models $\mathcal{M}$ operated under the 2D inputs. To fill the gap between the 3D scenarios 2D target models, additional physical rendering knowledge shall be efficiently encoded. Leveraging the proposed ResAdv-DDIM sampler and the advancements in Gaussian-splatting [42], we fill the gap between the 3D generation and 2D target models. As shown in Figure 5, we optimize latents with the

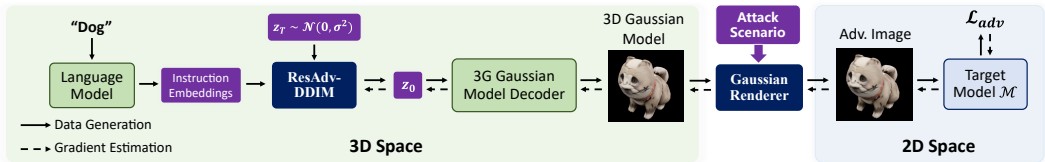

Figure 5: 3D optimization pipeline. Scenario knowledge is encoded with the Gaussian renderer.

gradient back-propagated through the 3D data structure and integrate the scenario knowledge during the differentiable rendering process. The optimization pipeline could be implemented concisely with the *Trellis* [43] framework, which is a recently published diffusion-based 3D access generation framework. For simplicity, we reformulate the *Trellis* sampling and rendering process as:

$$\mathbf{pos} = \text{Coords}(\mathcal{D}_{slat}(\boldsymbol{z}_0^{slat})), \ \text{Model}_{\text{GS}} = \mathcal{D}_{\text{GS}}(\boldsymbol{z}_0, \mathbf{pos}), \ x = \text{Renderer}_{\text{GS}}(\text{Model}_{\text{GS}}, \text{Camera}),$$

$$\mathcal{D}_{\text{GS}} : \{(z^i, pos^i)\}_{i=1}^L \rightarrow \{\{(x_i^k, c_i^k, s_i^k, \alpha_i^k, r_i^k)\}_{k=1}^K\}_{i=1}^L, \quad (7)$$

where $\boldsymbol{z}_0$ and $\boldsymbol{z}_0^{slat}$ are latents sampled by the diffusion model, and are represented by sparse and dense tensors, respectively. $\mathcal{D}_{slat}$ is the coarse structure decoder, $\text{Coords}$ transforms the voxel to point positions $\mathbf{pos}$, $\mathcal{D}_{\text{GS}}$ is the refined structure decoder that decodes each vertex into multiple Gaussian points and $\text{Renderer}_{\text{GS}}$ renders the Gaussian model to 2D images $x$ with the camera parameter. For SemanticAE generation, the refined feature generation process $\{z_T, z_{T-\Delta T}, ..., z_0\}$ is replaced with ResAdv-DDIM sampling in Eq. 4 with the rendering model embedded in $g_\theta$:

$$g_\theta(z_t, \mathbf{pos}, \text{Camera}) := \text{Renderer}_{\text{GS}}(\mathcal{D}_{\text{GS}}(f_{\theta, \Delta T_1} \circ \cdots \circ f_{\theta, \Delta T_k}(z_t, \mathbf{pos}), \mathbf{pos}), \text{Camera})$$

$$z_{t-\Delta T} := f_{\theta, \Delta T}(\arg\max_{z_t} \mathbb{E}_{\text{Camera} \sim P_{\text{Cam}}}[\mathcal{L}_{\text{ATK}}(\mathcal{M}(g_\theta(z_t, \text{Camera}, \mathbf{pos})))]) \quad (8)$$

We use the EoT method with gradient accumulation to optimize $z_t$ for unknown camera positioning, *i.e.*, samples Camera from $P_{\text{Cam}}$ in each iteration. The scenario knowledge is encoded in $P_{\text{Cam}}$ and the rendering background. Due to the stabilized guidance in ResAdv-DDIM, the gradient of the previous steps could be utilized as current-step gradient estimation, and therefore, fewer EoT steps are required. Since texture and localized positioning perturbation are enough for adversarial attacks, $z_0^{slat}$ is not included in the parameter space and serves as a semantic anchor. With the collaborative constraint of 3D diffusion and renderer model, semantically-constrained and multi-view adapted SemanticAEs are efficiently generated. Detailed implementation is shown in Appendix B.2.

### 3.4 Semantic-abstracted Attacking Evaluation Enhancement with Label Taxonomy

The evaluation of SemanticAE generator requires the benchmark to judge whether $x \in \mathcal{S}(\text{Text})$ and define $A_{\text{Text}}$, which determines the adversarial attack and semantic alignment performance of the generator, and is still a blank in practice. To address the issue, we provide a task construction method for automatic evaluation based on the application goal of the SemanticAE generation task. Note that our method evaluates the SemanticAE generator instead of the adversarial example.

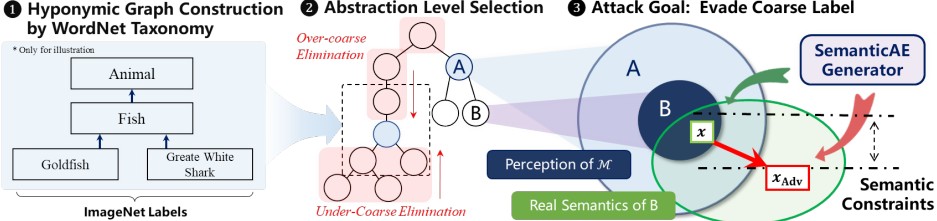

Figure 6: Construction of abstract label evasion evaluation task.

The task is constructed based on semantic abstracting with the label taxonomy. Firstly, the attack targets of existing non-target evaluation methods based on ImageNet labels are often too simple, while the constraint space of SemanticAE is relatively loose, making it easy for the attack generation model $\mathcal{G}$ to achieve successful attacks easily. For example, it is unreasonable to use the ImageNet label "tiger-shark" as the misclassification category $A_{\text{Text}}$ for the instruction Text "great-white-shark", since achieving successful attacks in this task may not show the capability of successful attacks in real scenarios. To clarify the boundary, we re-construct the evaluation label with a better abstraction level by leveraging *WordNet* [44] taxonomy. As shown in Figure 6, we firstly construct the hyponymic

graph based on the hyponymic relation defined by *WordNet*, then select the proper abstraction level, and finally define the attack goal as the evasion attack on the abstracted label. Specifically, under the definition of SemanticAE generation, the evaluation task is formulated as:

$$\text{Text} := \text{"Realistic image of [AbstractedLabel], specifically, [label]"},$$

$$A_{\text{Text}} := \{\text{label}_{\text{Adv}} \mid \text{AbstractedLabel} \notin \mathbf{Ancestors}(\text{label}_{\text{Adv}})\}, \tag{9}$$

$$\text{AbstractedLabel} \in \{c \in \mathbb{L}' \mid \exists l \in \mathbb{L} \text{ s.t. } c \in \mathbf{Ancestors}(l) \wedge \mathbf{Count_{Children}}(c) > 0\},$$

where $\mathbb{L}$ is the transitive closure of *ImageNet* labels on the hyponymic graph, the construction of AbstractedLabel is equivalent to: (1) Remove overly coarse-grained labels through annotation to obtain the label subset $\mathbb{L}'$. (2) For each linear path, select the node with the lowest height as a candidate label, constraining the upper bound of the abstracting level. (3) Eliminate descendant labels of the candidate labels to constrain the lower bound of the abstracting level.

Secondly, from the perspective of semantic constraint evaluation, using another deep-learning model for evaluation, *e.g.*, *CLIP*, will limit the benchmark to the robust region of such models. Drawing upon previous discussions and attempts in evaluation enhancement [17], we further conceptualize the sub-task of non-adversarial exemplar generation. As shown in Figure 6, the adversarial generator $\mathcal{G}$ is required to simultaneously generate a nearby sample $x_{\text{exemplar}} \in \mathcal{X}_{\text{exemplar}}$ as a proof that $x_{\text{adv}}$ complies with semantic constraints. We further propose the evaluation method based on attack success rate and pair-wise semantic metric as a complement to the single-image assessments:

$$ASR_{\text{Relative}} = \frac{\sum_{i=1}^{K} \text{Attack Success}(x_{\text{adv}}^{(i)}) \wedge \text{Classification Correct}(x_{\text{exemplar}}^{(i)})}{K \cdot \text{Accuracy}(\mathcal{X}_{\text{exemplar}})} \in [0, 1], \tag{10}$$

$$\text{SemanticDiff}_{\mathcal{S}} = \langle x_{\text{exemplar}}, x_{\text{adv}} \rangle_{\mathcal{S}},$$

where $K$ is the amount of samples, $\mathcal{S}$ is a visual similarity metric, such as *LPIPS* or *MS-SSIM*.Measuring local similarity is easier since high-level feature extraction, which could be attacked, is less required. By assuming the generator $\mathcal{G}$ is not motivated towards finding a *positive adversarial example*, achieving a high score on both metrics can sufficiently show both adversarial capability and instruction compliance of $\mathcal{G}$. Notably, $ASR_{\text{Relative}}$ evaluation metric imposes **a more rigorous** assessment for the masked language guidance in Section 3.3, as it eliminates the confounding variations in benign example generation methods through regularizing with Classification Correct($x_{\text{exemplar}}$).

## 4 Experiments

### 4.1 Experiment Settings

**Tasks and Baselines**    We evaluate different generation methods by generating 6 samples for each label in the ImageNet 1000-class label evasion task and the proposed abstracted label evasion task. The baseline method is constructed by combining **diverse ❶** Surrogate models, **❷** Transfer attack methods (MI-FGSM [5], DeCoWA [45]), and **❸** Diffusion-based naturalistic AE generation methods (AdvDiff [16], SD-NAE [15], VENOM [17]). Note that DeCoWa develops on top of MI-FGSM. For fairness, we incorporate the same pretrained diffusion model as SD-NAE and VENOM. For 3D SemanticAE we evaluate the classification models' performance on the generated video showing the rotating object. Detailed implementation and settings are shown in Appendix C.

**Evaluation Metrics**    Referenced image quality assessment metrics, including *LPIPS* [46] and *MSSSIM* [47], are been employed to measure the similarity between $x_{\text{exemplar}}$ and $x_{\text{adv}}$ under the proposed non-adversarial exemplar evaluation. Also, we supplement the non-reference image quality assessment *CLIP-IQA* [48] for the generated images. We do not use *FID* and *IS* as a primary metric since their adapted vision backbones are simple and might be adversarially attacked, and keeping the feature distribution consistent is not the primary goal of the SemanticAE generation. For attack evaluation, we computed the classification accuracy and $ASR_{\text{Relative}}$, defined in Eq. 10, across diverse targets set $\mathcal{T} = \{$*ResNet50* [49], *ViT-B/16* [50], *ConvNeXt-T* [51], *ResNet152*, *InceptionV3* [52], *Swin-Transformer-B* [53]$\}$. The first three models are used individually as surrogates in experiments. The main paper presents the average $ASR_{\text{relative}}$ and accuracy (ACC) on the target model, while the detailed transfer attack performance on different target models is shown in Appendix D.1. In addition, we fuse the input-transformation-based module *DeCoWa* with ResNet50 as one of the surrogate models to evaluate the collaboration capability of the generation algorithms. 2D / 3D generation times are benchmarked on a single 4090 or A800 GPU, respectively, by generating 100 samples with abstracted labels, and are presented in the results with standard deviation.

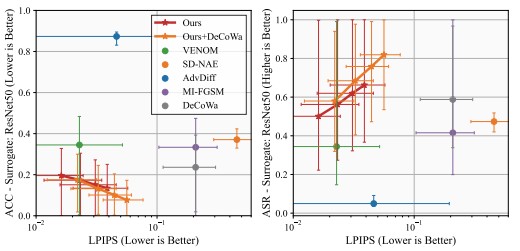

Figure 7: ImageNet label results.

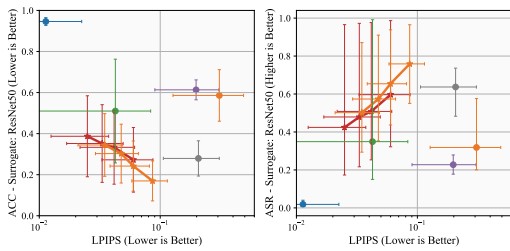

Figure 8: Abstracted label results.

Table 1: Results on more surrogate models. Appendix D shows our Pareto optimality in each setting.

| Attacker Settings | | ImageNet Label SemanticAE Generation Task | | | | | Coarse Label Evasion Task (Section 3.4) | | | | | |
| Surrogates | Method | Acc.↓ | ASR↑ | Clip$_Q$↑ | MSSSIM↑ | LPIPS↓ | Acc.↓ | ASR↑ | Clip$_Q$↑ | MSSSIM↑ | LPIPS↓ | Time(s)↓ |
|---|---|---|---|---|---|---|---|---|---|---|---|---|
| ResNet50 | MI-FGSM | 33.4% | 41.5% | 0.548 | 0.880 | 0.201 | 61.3% | 22.8% | 0.551 | 0.885 | 0.198 | 1.43$_{\pm0.02}$ |
| | AdvDiff | 87.3% | 4.9% | 0.634 | 0.939 | 0.046 | 94.6% | 1.8% | 0.621 | 0.992 | 0.011 | 19.6$_{\pm0.01}$ |
| | SD-NAE | 37.1% | 47.4% | 0.841 | 0.433 | 0.457 | 58.6% | 31.8% | 0.771 | 0.599 | 0.308 | 24.43$_{\pm0.14}$ |
| | VENOM | 34.5% | 34.4% | 0.795 | 0.972 | 0.023 | 51.0% | 34.9% | 0.779 | 0.951 | 0.043 | 3.09$_{\pm0.52}$ |
| | Ours | 15.1% | 62.0% | 0.815 | 0.961 | 0.031 | 35.2% | 47.9% | 0.808 | 0.958 | 0.033 | 7.26$_{\pm2.57}$ |
| ViT-B | MI-FGSM | 32.7% | 42.4% | 0.524 | 0.855 | 0.205 | 58.2% | 25.7% | 0.521 | 0.860 | 0.207 | 1.46$_{\pm0.01}$ |
| | AdvDiff | 65.6% | 30.3% | 0.638 | 0.430 | 0.390 | 94.1% | 2.2% | 0.628 | 0.972 | 0.026 | 20.5$_{\pm0.01}$ |
| | SD-NAE | 33.7% | 51.7% | 0.844 | 0.441 | 0.459 | 56.1% | 33.6% | 0.787 | 0.609 | 0.300 | 24.5$_{\pm0.10}$ |
| | VENOM | 30.5% | 40.6% | 0.796 | 0.977 | 0.021 | 46.3% | 40.3% | 0.780 | 0.958 | 0.040 | 3.07$_{\pm0.33}$ |
| | Ours | 10.9% | 69.7% | 0.815 | 0.956 | 0.038 | 28.7% | 55.4% | 0.814 | 0.955 | 0.039 | 7.23$_{\pm2.15}$ |
| ConvNeXt | MI-FGSM | 31.9% | 44.9% | 0.543 | 0.877 | 0.204 | 41.2% | 46.4% | 0.532 | 0.88 | 0.202 | 1.46$_{\pm0.01}$ |
| | AdvDiff | 52.9% | 44.4% | 0.636 | 0.312 | 0.471 | 93.3% | 3.2% | 0.627 | 0.985 | 0.017 | 19.9$_{\pm0.11}$ |
| | SD-NAE | 22.4% | 67.3% | 0.848 | 0.432 | 0.458 | 53.6% | 36.1% | 0.782 | 0.603 | 0.308 | 24.5$_{\pm0.10}$ |
| | VENOM | 28.8% | 44.6% | 0.796 | 0.978 | 0.020 | 42.6% | 45.0% | 0.785 | 0.961 | 0.037 | 3.14$_{\pm0.65}$ |
| | Ours | 9.1% | 75.8% | 0.817 | 0.958 | 0.036 | 28.6% | 57.4% | 0.812 | 0.957 | 0.036 | 6.96$_{\pm1.96}$ |
| ResNet50 +DeCoWa | MI-FGSM | 23.6% | 58.8% | 0.535 | 0.869 | 0.201 | 27.9% | 63.7% | 0.474 | 0.870 | 0.207 | 3.01$_{\pm0.04}$ |
| | AdvDiff | 67.9% | 27.5% | 0.597 | 0.761 | 0.293 | 93.4% | 3.2% | 0.629 | 0.934 | 0.081 | 20.8$_{\pm0.18}$ |
| | SD-NAE | 26.6% | 61.7% | 0.845 | 0.421 | 0.470 | 64.4% | 26.3% | 0.782 | 0.568 | 0.331 | 24.3$_{\pm0.07}$ |
| | VENOM | 36.1% | 31.4% | 0.805 | 0.968 | 0.027 | 56.2% | 28.1% | 0.796 | 0.944 | 0.048 | 4.93$_{\pm2.67}$ |
| | Ours | 10.1% | 75.8% | 0.810 | 0.947 | 0.044 | 30.3% | 57.4% | 0.808 | 0.943 | 0.048 | 10.7$_{\pm3.60}$ |

\* Attack performance is **averaged** across targets. Detailed results with the specified target models are shown in Appendix D.

## 4.2 Overall Performance Evaluation

**2D SemanticAE Generation** The overall results on 2D SemanticAE generation is shown in Figure 7 and 8 with the strength of semantic constraint of our method set in $\epsilon = \{1.5, 2, 2.5, 3\}$ and $\epsilon = \{2, 2.5, 3, 4\}$ via Eq. 5, respectively. Multiple $\epsilon$ are applied since it is difficult to control and align the distortion strength for baselines. The min/max ASR across target models and the standard deviation of LPIPS across generated images are plotted as bars. Table 1 shows the results of our method set as $\epsilon = 2.5$. More results are in the appendix. Overall, in **any** of the 4 surrogate and 2 task settings, **InSUR** is able to achieve **at least 1.19×** average ASR and **1.08×** minimal ASR across **all target models in** $\mathcal{T}$, and maintains with lower LPIPS (unsuccessful baseline generation with avg. ASR < 5% are not considered), showing the **consistent superiority**. The Pareto improvement shown by the figure is more significant. Moreover, ❶ $\epsilon$-based semantic constraint in Eq. 5 achieves more consistent LPIPS across images generated in identical settings. ❷ For $Clip_Q$, our method performs better in the challenging abstracted-label evasion task, while SD-NAE is higher in original tasks.

**3D SemanticAE Generation** We export the video visualization of the object under MPEG4 encoding, and evaluate the attack performance by reading it. The surrogate and target models are both ResNet50. The results are in Table 2. There is no 3D SemanticAE previously available. It shows that our results show satisfactory attack performance, validating the

Table 2: 3D generation results.

| Generator | Acc. | ASR | MSSSIM | LPIPS |
|---|---|---|---|---|
| Non-Adversarial | 21.5% | — | — | — |
| Ours w/o ResAdv | 17.9% | 45.1% | 0.658 | 0.261 |
| Ours | 2.8% | 92.2% | 0.665 | 0.258 |

cross-task scalability of **InSUR**. Note that since 3D-diffusion research is still under development, the clean accuracy on the generated 3D samples is not high, while making **InSUR** a growable research.

## 4.3 Key Ablation Studies

**Residual Approximation** We evaluate the effects of residual approximation steps $k$ in the adversarial guidance $g$ in Eq 4, under the 2D abstracted label task and the 3D task. In implementation, $k$ is controlled by the upper-bound parameter $K$ in Eq. 15 in Appendix B.1. The performance improvement is significant and consistent compared to the result without future-step sampling prediction ($K = 0$), which represents the original DDIM sampling with adversarial guidance. The naturalness metrics, including $Clip_Q$, MSSSIM, and LPIPS, are also slightly improved. Moreover, attributed to

the adaptive iteration mechanism, more accurate estimation leads to lower optimization steps, and therefore, the increase in time consumption is sub-linear. Detailed results on more settings are shown in the Appendix D.3. Parameter and time consumption analysis are shown with the implementation details in Appendix B.

The effect of different $k$ on the process of adversarial optimization is shown in Figure 9. The setting of Text is *a dog sitting on the floor*, and the evasion label is *dog*. As $k$ increases, the estimated ASR after adversarially sampling $x_t$ increases earlier. Since the white-box ASR in both settings is nearly 100%, the improvements are from: (1) by eliminating the *referring diversify*, the initial sampling steps receive more accurate guidance. (2) More effective adversarial optimization on earlier diffusion denoising steps provides better on-manifold regularization, leading to better visual quality and better adversarial transferability.

Table 3: Ablation of residual approximation

| Surro. | $K$ | $\overline{\text{Acc.}}$ | $\overline{\text{ASR}}$ | $\text{Clip}_Q$ | MSSSIM | LPIPS | Time |
|---|---|---|---|---|---|---|---|
| ViT-B | 0 | 43.1% | 43.3% | 0.794 | 0.941 | 0.055 | $4.53_{\pm0.19}$ |
| | 1 | 33.5% | 54.7% | 0.812 | 0.942 | 0.049 | $6.87_{\pm1.48}$ |
| | 2 | 31.3% | 56.6% | **0.816** | 0.942 | 0.049 | $7.44_{\pm1.92}$ |
| | 3 | 29.2% | 57.5% | 0.807 | 0.943 | 0.048 | $7.75_{\pm2.37}$ |
| | 4 | **27.7%** | **60.1%** | 0.813 | **0.944** | **0.047** | $7.87_{\pm2.31}$ |
| ResNet50 +DeCoWa | 0 | 39.8% | 47.1% | 0.764 | 0.946 | 0.053 | $5.65_{\pm0.38}$ |
| | 1 | 36.3% | 50.4% | 0.812 | 0.954 | 0.040 | $9.64_{\pm2.64}$ |
| | 2 | 32.8% | 52.6% | **0.818** | **0.955** | **0.039** | $10.7_{\pm3.27}$ |
| | 3 | 30.4% | 53.9% | 0.817 | **0.955** | **0.039** | $11.2_{\pm3.67}$ |
| | 4 | **29.5%** | **54.2%** | 0.814 | **0.955** | **0.039** | $11.5_{\pm4.00}$ |
| 3D Gen. | 0 | 17.9% | 45.1% | – | 0.658 | 0.261 | $7.49_{\pm1.49}$ |
| | 1 | 3.4% | 90.2% | – | 0.658 | 0.262 | $30.4_{\pm35.02}$ |
| | 2 | 3.4% | 91.2% | – | 0.659 | 0.261 | $32.7_{\pm39.17}$ |
| | 3 | 2.9% | 91.2% | – | **0.671** | **0.255** | $41.1_{\pm53.75}$ |
| | 4 | **2.8%** | **92.2%** | – | 0.665 | 0.258 | $40.1_{\pm59.25}$ |

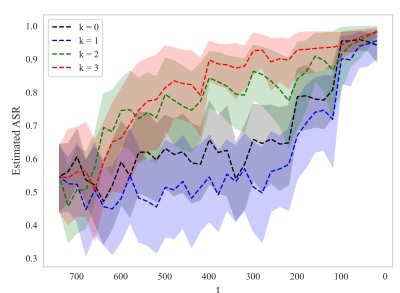

Figure 9: Estimated ASR of $\hat{x}_0$ after adv. sampling $x_t$, under different settings of $k$ in Eq. 4. Earlier increase of estimated ASR leads to better naturalness and higher attack transferability.

**Spatial Masking of Language Guidance** We evaluate the effect of guidance masking by setting $M_{\text{edge}}/M_{\text{mid}}$ to different values in Eq. 6, and setting to 1.0 represent removing the masking. The results are shown in Table 4 and Figure 10, indicating that (1) decreasing $M_{edge}$ leads to an increase in unconditional guidance, which enriches the background diversity as shown in the figure, and may lead to the improvements of $\text{Clip}_Q$. (2) When the budget $\epsilon$ is small ($\epsilon = 2$) and the optimization space is relatively narrow, diversifying the background is beneficial. Note that $\epsilon$ is large, the improvement is marginal since there is no need for expanding the optimization space. Overall, the guidance masking design improves diffusion models' adaptability to strongly constrained SemanticAE generation.

Table 4: Ablation of guidance masking.

| $\epsilon$ | $M_{\text{edge}}/M_{\text{mid}}$ | $\overline{\text{Acc.}}$ | $\overline{\text{ASR}}$ | $\text{Clip}_Q$ | MSSSIM | LPIPS |
|---|---|---|---|---|---|---|
| 2 | 0.0 | **32.8%** | **52.3%** | **0.813** | **0.958** | 0.035 |
| 2 | 0.1 | 34.5% | 50.2% | 0.809 | 0.957 | 0.035 |
| 2 | 1.0 | 36.9% | 48.3% | 0.788 | **0.958** | 0.034 |
| 4 | 0.0 | **16.6%** | **75.9%** | 0.804 | 0.901 | 0.087 |
| 4 | 1.0 | 17.3% | **75.9%** | 0.775 | **0.904** | 0.084 |

Instruction: Realistic image of boat, specifically, canoe. Evasion label: boat.

| 0.0 | 0.1 | 0.2 | 0.4 | 0.6 | 1.0 |

Figure 10: Vis. of different guidance masking. The value under images denotes $M_{\text{edge}}/M_{\text{mid}}$.

## 4.4 Visualization of Generated SemanticAEs

We selected the 2D / 3D samples with $x_{\text{exemplar}}$ correctly classified and visualized in Fig. 11 and Fig. 12. 2D samples are generated with the original ImageNet-label and the surrogate model is DeCoWa+ResNet. 3D samples are generated and visualized as the main experiment. The results are coherent with the main table, showing that MI-FGSM is not natural, SD-NAE disturbs more global semantics, while ours achieves both global semantic preservation and naturalness. Through observation, our method generates local in-manifold patterns to achieve the strong attacks, *e.g*, adding fog in the castle image or altering the lightning in the jellyfish image. For 3D results, although the MSSSIM and LPIPS metrics are not excellent in the main experiment, the generated 3D objects are natural and follow the semantic constraint. More visualizations are shown in Appendix E.

## 4.5 Discussions

**Extending to Attacking Large Vision Language Models** The InSUR framework is agnostic to specific target models and tasks, and can be directly applied to adversarial attack evaluation

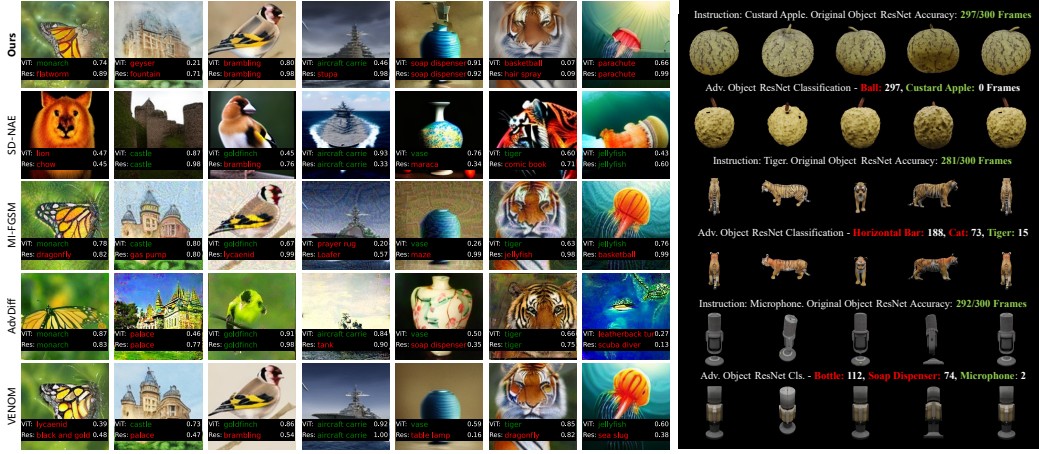

Figure 11: Comparison of 2D SemanticAEs.

Figure 12: 3D SemanticAEs

in Visual Question Answering (VQA) scenarios. To illustrate the capability, we conduct VQA experiments using OpenAI CLIP [54] model *RN50* (with DeCoWA augmentation) as the surrogate, using *stabilityai/stable-diffusion-2-1-base* [55] as the diffusion generator, and generate adversarial examples for the VQA problem. The question and options are *"What is in the picture:"* and *{Police car, Ambulance, Taxi, School Bus}*. We evaluate **weak-to-strong** transferability across OpenAI CLIP variants {RN50, RN101, RN50x4, RN50x16, ViT-B/32, ViT-B/16} and larger vision language models (VLM), LLaVA [56] and Qwen-7B [57]. The results are in Table 5, showing that our technical contribution is also effective in the VQA transfer attack task.

Table 5: $ASR_{relative}$ (%) of attacks. The surrogate model is ResNet50 CLIP model.

| Residual Approximation | Target Model | | | | | | | |
|---|---|---|---|---|---|---|---|---|
| | RN50 | RN101 | RN50x4 | RN50x16 | ViT-B/32 | ViT-B/16 | LLaVa1.5-7B | Qwen2.5VL-7B |
| ✗ ($K = 0$) | 77.50 | 27.50 | 30.00 | 11.11 | 12.50 | 10.53 | 11.11 | 16.22 |
| ✓ ($K = 3$) | 97.50 | 57.50 | 62.50 | 27.78 | 40.00 | 47.37 | 22.22 | 35.14 |

**Broader Applications** Our approach also has the potential to extend to real-world adversarial attacks under the 3DGS representations [31], providing technical tools for security-related research. We tested our generated 3D SemanticAE in with lightning, material, and camera settings with *Blender* environment, and the ResNet50 ACC is **59.1%**, **20.0%**, for the two settings in Figure 13, respectively. Although the attack performance is worse than the original rendering (ACC=7.7%), the relative improvement is significant. Our framework can also be integrated with existing 3D scene generation pipelines [58], enabling more effective incorporation of diffusion-based adversarial optimization into world models. This

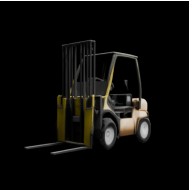
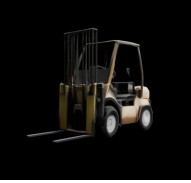

(a) w/o ResAdv.  (b) with ResAdv.

Figure 13: *Blender* re-rendered 3D SemanticAEs with altered lightning.

supports safety-critical scenario generation for applications such as autonomous driving [59, 60] and embodied agents [61]. On another front, recent work [62] demonstrates that diffusion-generated hard samples can achieve higher sample efficiency, especially with adversarial guidance, suggesting that our InSUR framework also holds promise for supporting the generation of adversarial training data.

## 5 Conclusion

This paper proposes multi-dimensional uncertainty reduction frameworks for SemanticAE generation, pushes the boundary of the 2D generation, and opens the door to 3D generation. The proposed technology consistently improves the attack performance and has the potential to scale to other tasks. Moreover, we believe it could provide valuable insights for the test-time scaling of the red-teaming framework. For limitations, there is scope for improvement in the generation quality, the evaluation on larger models, and the application in the real world, suggesting future research avenues of the SemanticAE generation algorithms based on concrete generative models and the scenario adaptation methods oriented to real-world scenarios.

## Acknowledgments and Disclosure of Funding

This work was supported by the National Natural Science Foundation of China (NSFC) under Grant 62476018, and was supported by Zhongguancun Laboratory.

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

# Contents

# A Technical Background and Related Works

## A.1 Diffusion models

**Diffusion Model** This work mainly applies the diffusion model as the language-guided data generator. As a brief review, diffusion models establish the theoretical and technical route of estimating the score function $\nabla_X \log p(X)$ of the data distribution $X$ by training UNet models on the dataset $\{x\}$, and sampling the data from the score function with numerical methods. One of the key insights of diffusion models is alleviating the training difficulty by disturbing the data $x$ gradually with noise, and models it as a forward process from $x_0 \sim X$ to $x_T \sim \mathcal{N}(0, I)$ with the Markov process $X_t = \sqrt{1 - \beta_t} X_{t-1} + \sqrt{\beta_t} \mathcal{N}(0, I)$ scheduled by $\boldsymbol{\beta}$. The training is performed by learning a denoising process, *i.e.*, learn to predict the distribution $q_\theta(x_{t-1}|x_t)$ with the neural network $\epsilon_\theta$. During inference, the data is sampled from $x_T \sim \mathcal{N}(0, I)$ to $x_0$ step by step with $q$. DDIM model reinterprets the forward process as $p(x_t|x_{t-1}, x_0)$, which abandons the Markov property, and constructs the sampling method by finding $q_\theta(x_{t-1}|x_t, x_0)$. It also provides the theoretical grounding for the step jumping in the sampling process. The DDIM [38] sampling procedure could be formulated as (deterministic version):

$$x_{t-\Delta T} = \sqrt{\bar{\alpha}_{t-\Delta T}/\bar{\alpha}_t} \left( x_t - \sqrt{1 - \bar{\alpha}_t} \epsilon_\theta(x_t, t) \right) + \sqrt{1 - \bar{\alpha}_{t-\Delta T}} \cdot \epsilon_\theta(x_t, t), \tag{11}$$

where $\bar{\alpha} = \prod_{s=1}^{t}(1 - \beta_s)$, and $\Delta T$ is the step interval. We adapted this notation in the main paper. The language-guided generation problem is modeled by adding the conditional term Context that corresponds to the data $x$ and sampling the data by $\epsilon_\theta(x_t, t, \text{Context})$, where Context is the encoding given by the language model. To balance the generation diversity and the instruction following, conditional and unconditional guidance are integrated, *i.e.*, $\epsilon_\theta = -\omega\epsilon_\theta(x_t, t, \text{Unconditional}) + (1 + \omega)\epsilon_\theta(x_t, t, \text{Context})$, in the *classifier-free guidance* [63].

**Discussions** In the application in content edit [40], masking has been applied to constrain the editing area, *i.e.* $x_{t-1} = \text{Mask} \cdot x_{\text{t,edit}} + (1 - \text{Mask}) \cdot x_{\text{t,original}}$. In our 2D generation task, we take a further step to model the interaction between conditional and unconditional guidance, and achieve the new function of re-distributing the spatial strength of the semantic constraint. In recent years, training techniques that enable single-step generation have been proposed [64, 65]. Although our method is designed with multi-step diffusion models, our method still has practical value since (1) there is a performance gap between multi-step and single-step diffusion models, and (2) as the adversarial optimization is performed with local perturbations, sampling in small step sizes might provide a better regularization for transfer attacks.

## A.2 Generating Hard Samples from Language

Language-guided adversarial example generation is still under development. We categorize two major paradigms. ❶ Perturbing the input text without altering the language-guided data generation model. A line of study has focused on utilizing the language-guided image edition model to evaluate the robustness of visual models [66, 67, 68]. Compared to adversarial attacks, they focus more on enriching the dataset than finding hard examples for victim models, while the adversarial capability is relatively limited. Related to the discussion in Section 3.2 of the main paper, overly perturbing the text guidance may lead to unsatisfactory semantic alignment. To solve this problem, [69] integrates the additional *CLIP* supervision on the generated image. ❷ Altering the language-guided data generation model by introducing the adversary capability. Although it has difficulties in global optimization, this paradigm has an advantage in the fine-grained control of the generated samples.

We focus on the latter paradigm, since (1) Perturbing the text input only will limit the capability of the generation method within the semantic knowledge learned by both discriminative and generative models. *e.g.*, a *CLIP*-based semantic supervision may not attack the *CLIP* model itself. (2) Optimizing the input text for the entire data generation process with the adversarial feedback is time-consuming. These features are crucial from the perspective of **SuperAlignment** [70], *i.e*, creating efficient methodologies that utilize small models with limited capability to build the oversight framework for larger models. Also, we believe that the two methods could be implemented into an integrated red-team system, *i.e*, the language-space reject sampling as the high-level generation method and the latent-space optimization as the mid or low-level generation module. Therefore, our proposed evaluation task focuses on the second paradigm, serving as an evaluation of the key component of the red-team framework.

## A.3 Transfer Attack Methodology

Transferable attack denotes finding adversarial examples that can attack other unknown or even stronger black-box models, which is practical for revealing real-world AI safety problems. The general method is to find a surrogate model(s) in the related task and optimize the adversarial example for it with the regularization. From the perspective of the optimization pipeline, regularization on three modules has been investigated to improve transferability: ❶ The gradient decent. Momentum is the general method to boost the transferability. ❷ The surrogate model. Model ensemble can effectively enhance transferability by combining gradient features from multiple distinct models [71, 72]. Constructing a proper objective using the Sharpness-Aware Minimization (SAM) [73] or the attention mechanism [3], to model the cross-model vulnerability, is also beneficial for transferability. ❸ The input transformation. Transfer improvement could be achieved by inserting a random transformation between the current-step adversarial example and the surrogate model's input, the [74, 45]. This could be regarded as improving the diversity of the surrogate model. From the theoretical perspective, decreasing the cooperation within the adversarial pattern [75] and in-manifold constraint [37] is beneficial for transferability. In the adversarial example generation pipeline, the generative-model-based method, which this work focuses on, could be regarded as a replacement of gradient descent with a better in-manifold constraint. We construct baselines and evaluation tasks based on this background.

## A.4 3D Generation and Gaussian Splatting

3D geometric data employs diverse representations that are crucial in 3D generation. These representations can be categorized into three types: ❶ explicit representations, such as point clouds [76] and meshes [77], ❷implicit representations, such as Neural Radiance Fields (NeRFs) [41], ❸hybrid representations, such as 3D Gaussian [42]. Currently, 3D Gaussian splatting is widely used in 3D generation due to the geometric editability, high-frequency detail capture capability and real-time rendering efficiency. Each 3D Gaussian point can be represented by the following parameters: position $x$, spherical harmonics coefficients $c$, opacity $\alpha$, a rotation matrix $r$, and a scaling matrix $s$. Then, 3D Gaussian points can be projected onto the image plane via the viewing transformation $W$ and the Jacobi affine approximation matrix of the projection transformation matrix in geometry. In terms of appearance, we can calculate the color of every pixel in the 2D image by blending N ordered points overlapping the pixel using spherical harmonics coefficients $c$ and opacities $\alpha$.

With the development of 3D representations, 3D adversarial examples for different 3D structures have also been advanced [78, 79, 80]. Unlike other adversarial methods, which are based on the geometric or texture data of existing objects, for the first time, we realize the reference-free generation of semantically constrained 3D adversarial examples by utilizing language-guided 3D generation models. We implement language-guided 3D adversarial example generation through the ResAdv-DDIM sampler referencing the optimization pipeline of Trellis in latent space and decode into 3D Gaussian formats. Then, we achieve 3D adversarial attacks on 2D target models using 3D Gaussian rendering.

# B Detailed Implementation of InSUR Framework

Adversarial attacks often require reengineering existing tools to achieve new functions, and their implementation is not simple, especially when achieving new characteristics. Therefore, we provide the design principles and the key techniques in the main paper, and supplement the detailed implementation in this section as a complement. In the following subsections, we describe the implementation of the SemanticAE generator based on the different scenarios, and then provide more details about the *Semantic-abstracted Attacking Evaluation Enhancement*.

## B.1 2D Image Generation

**ImageNet Object Generation with Guidance Masking**   For the classical image generation problem, we start with the original DDIM sampling process:

$$x_{t-\Delta T} = \sqrt{\bar{\alpha}_{t-\Delta T}/\bar{\alpha}_t} \left( x_t - \sqrt{1 - \bar{\alpha}_t}\epsilon_\theta(x_t, t) \right) + \sqrt{1 - \bar{\alpha}_{t-\Delta T}} \cdot \epsilon_\theta(x_t, t), \tag{12}$$

where $\bar{\alpha} = \prod_{s=1}^{t}(1 - \beta_s)$ is the sampling parameter, and $\Delta T$ is the step interval. As described in section 3.3, masked guidance is adapted as the conditional diffusion guidance, which formulates the denoise function $x_{t-\Delta T} := f_{\theta,\Delta T}(x_t)$ as:

$$f_{\theta,\Delta T}^{(t,\text{Text})}(x_t) = \sqrt{\frac{\bar{\alpha}_{t-\Delta T}}{\bar{\alpha}_t}} \left( x_t - \sqrt{1 - \bar{\alpha}_t}\epsilon_\theta(x_t, t, \text{Text}) \right) + \sqrt{1 - \bar{\alpha}_{t-\Delta T}} \cdot \epsilon_\theta(x_t, t, \text{Text}),$$

$$\epsilon_\theta(x_t, t, \text{Text}) := (1 - M) \cdot \epsilon_{\theta,\text{Unconditional}}(x_t, t) + M \cdot \epsilon_{\theta,\text{Conditional}}(x_t, t, \text{Text}).$$
(13)

The guidance mask is defined as a matrix with different values in the border elements:

$$M_{ij} := \begin{cases} M_{\text{mid}} & \frac{h}{16} \leq i < \frac{15h}{16}, \frac{w}{16} \leq j < \frac{15w}{16}, \\ M_{\text{edge}} & \text{otherwise.} \end{cases}$$
(14)

In the main paper, we use the simplified notation of $f$ without parameters Text and $t$ explicitly written. Here we use the detailed notations. For the experiments, we adapted $M_{\text{mid}} = 3.0$ and $M_{\text{edge}} = 0.3$ in the main experiment and selected $M_{\text{edge}} = \{0.0, 0.3, 3.0\}$ in the ablation study.

**Residual Approximation**   Recall that ResAdv-DDIM defines a residual estimation function $g$ for the adversarial feedback of the target model. We construct the step size of $g$ as evenly distributed:

$$g_\theta^{(t,\text{Text})}(x_t) := \underbrace{f_{\theta,\Delta T_1}^{(\Delta T_1,\text{Text})} \circ f_{\theta,\Delta T_2}^{(\Delta T_2 + \Delta T_1,\text{Text})} \circ \cdots \circ f_{\theta,\Delta T_k}^{(t,\text{Text})}}_{k \text{ times}}(x_t), \text{ where } \sum_{i=1}^{k} \Delta T_i = t,$$

$$k := \lceil \frac{t}{\lfloor T/K \rfloor} \rceil, \quad \Delta T_i(t) = \begin{cases} \lfloor T/K \rfloor, & 1 < i \leq k \\ t \mod \lfloor T/K \rfloor, & i = 1 \end{cases},$$
(15)

where $K$ is the maximal iteration number, corresponding to $Iter_{\max}$ in the ablation study in the main paper. For brevity, we define $T$ as the number of sampling steps of the original DDIM generator and use the sampling interval of the original DDIM generator as the unit of $\Delta T$. We let $t_s$ (default set as $0.75$) be the start step of the adversarial optimization, and the sampling process with adversarial optimization is formulated as:

$$x_{t-\Delta T} = \begin{cases} f_{\theta,\Delta T}^{(t,\text{Text})} \left( \arg\max_{x_t} \mathcal{L}_{\text{ATK}}(\mathcal{M} \circ g_\theta^{(t,\text{Text})}(x_t)) \right), & t \leq t_s, \\ f_{\theta,\Delta T}^{t,\text{Text}}(x_t), & t > t_s. \end{cases}$$
(16)

**Collaborating with Guidance Masking**   To generate attack-related image backgrounds without disturbing the foreground semantics, as the goal of *Context-encoded Attacking Scenario Constrain*, we let the exemplar generation $x'_t \rightarrow x'_{t-\Delta T}$ communicate with the adversarial generation $x_t \rightarrow x_{t-\Delta T}$. Specifically, at the beginning of the adv. optimization in step $t_s$, we set the benign sample generated from $x'_{t_s} \leftarrow x_{t_s}$ as the initialization of the constraint anchor. At the end of the adversarial optimization, the optimized background is written back $x'_0 [M = M_{edge}]_{\text{Select}} \leftarrow \frac{1}{2}(x_0 + x'_0) [M = M_{edge}]_{\text{Select}}$ after the generation.

**Adaptive Optimization Iteration**   To improve the efficiency in multi-step diffusion-based adversarial optimization, we implement the adaptive iteration mechanism for ResAdv-DDIM by early-stopping the optimization problem in Eq 16, which is formulated as:

$$n := \begin{cases} M, & t = t_s \vee t < 4, \\ m, & \text{otherwise.} \end{cases}$$

$$\text{PerformOptimize} := \underbrace{(i \leq n \ \wedge \ \arg\max_{l \notin A_{\text{Text}}} P(\mathcal{M}(x_t) = l) > \xi_1)}_{\text{If confidence} \leq \xi_1, \text{ early stop. Maximal optimize iterations is n.}} \ \vee$$

$$\underbrace{(i = 1 \ \wedge \ \arg\max_{l \notin A_{\text{Text}}} P(\mathcal{M}(x_t) = l) > \xi_2)}_{\text{If confidence} \leq \xi_2, \text{ do not optimize at the first step.}},$$
(17)

where $i$ is the current iteration number, and $n$ stands for the maximal adversarial optimization iteration in the single diffusion step. We set $\xi_1 = 0.1$, $\xi_2 = 0.01$, and set $m = 3$ and $M = 10$ for diversified

Table 6: Analysis of the different setting of $\xi$

| $\xi_1$ | 0.15 | 0.15 | 0.15 | 0.1 | 0.1 | 0.1 | 0.05 | 0.05 | 0.05 |
| --- | --- | --- | --- | --- | --- | --- | --- | --- | --- |
| $\xi_2$ | 0.02 | 0.01 | 0.005 | 0.02 | 0.01 | 0.005 | 0.02 | 0.01 | 0.005 |
| ResNet50 ACC$\downarrow$ | 0.0224 | 0.0192 | 0.0214 | 0.0246 | 0.0246 | 0.0246 | 0.0224 | 0.0256 | 0.0246 |
| ViT-B/16 ACC$\downarrow$ | 0.2949 | 0.2885 | 0.2885 | 0.2853 | 0.2724 | 0.2714 | 0.2404 | 0.2382 | 0.2511 |

strategy in different sampling steps, *i.e.*, the initial and final steps are set with a higher maximal iteration number ($n = M$).

In Table 6, we present a parameter analysis of $\xi_1$ and $\xi_2$ using ResNet50 as the surrogate model on the Abstract Label Evasion Task. In white-box attacks, all tested threshold values achieve strong performance (accuracy after attack < 3%). In black-box attacks, smaller $\xi_1$ values yield higher transferability, because lower $\xi_1$ leads to more aggressive adversarial optimization in early denoising steps—optimization that tends to be more transferable (this is consistent with the design principle of ResAdv-DDIM, which aims to enhance early-step optimization). However, this also increases computational cost. A practical deployment could balance effectiveness and efficiency.

**Attack Loss Construction**    For the classification task, given the incorrect label set $A_{\text{Text}}$, we implement the loss as:

$$\mathcal{L}_{\text{ATK}} := -\text{Logits}[\hat{L}_{\text{Tar}}]. \tag{18}$$

Where Logits is the model's prediction, and $\hat{L}_{\text{Tar}}$ is the estimated highest confidence label in the incorrect label set $A_{\text{Text}}$. We maximize $\mathcal{L}_{\text{ATK}}$ to perform the attack. For the current denoising step $x_t$, the logits are estimated based on the residual approximation function and the surrogate model, *i.e.*, $\text{Logits} = \mathcal{M}(g_\theta^{(t,\text{Text})}(x_t))$. To further eliminate guidance fluctuation, we adapt the fixed $\hat{L}_{\text{tar}}$ after its initialization in the initial steps of attack optimization (*target label update delay*). Note that our adversarial attack method is not designed for the classification task only, and the loss construction could be substituted regarding the specific adversarial example generation task.

The overall SemanticAE generation algorithm is constructed by further integrating *global perturbation constraints*, *momentum gradient optimization*, *target label update delay* (after $t < t_k = 0.4T$) through the bi-level optimization. The pseudo-code is shown in Algorithm 1. The default configuration for other parameters is coherent with related baselines, *i.e.*, $\beta = 0.5, s = 0.7, T = 100$.

---

**Algorithm 1:** ResAdv-DDIM

**Require:** Input: $\text{Text}, \mathcal{M}, A_{\text{Text}}, f, \epsilon$, optimization parameters $T, K, t_s, t_k, \beta, s$

   Init $x_T \sim \mathcal{N}(0, \boldsymbol{I}), v_x \leftarrow 0, x'_T \leftarrow x_T$.

   **for** $t = T, ..., 1$ **do**

     **if** $t \leq t_s$ **then**

       **for** $i = 1, 2, ..., n$ **do**

         **if** $\neg\text{PerformOptimize}$ *(Eq. 17)* **then**

           **break**               /* Adaptive Iteration */

         **if** $t = t_s \lor t < t_k$ **then**

           $\hat{L}_{\text{Tar}} = \arg\max_{L \in A_{\text{Text}}} \mathcal{M}(g_\theta^{(t,\text{Text})}(x_t))_{\text{logits}}[L]$   /* Target Update */

         $v_x \leftarrow \beta v_x - (1 - \beta)\nabla_{x_t}\mathcal{L}_{\text{ATK}}$.       /* Momentum Updates */

         $x_t \leftarrow x_t + s * v_x$.            /* Adversarial Optimization */

     **if** $t = t_s$ **then**

       $x'_{t-1} \leftarrow x_{t-1}$            /* Determine Exemplar on Step $t_s$ */

     $x_{t-1} \leftarrow f_{\theta,1}^{(\text{t},\text{Text})}(x_t)))(x_t)$       /* DDIM Sampling Step for $x_{\text{adv}}$ */

     $x'_{t-1} \leftarrow f_{\theta,1}^{(\text{t},\text{Text})}(x_t)))(x'_t)$.      /* DDIM Sampling Step for $x_{\text{exemplar}}$ */

     $x_{t-1} \leftarrow x_{t-1} + \min\{\epsilon, ||x'_{t-1} - x_{t-1}||_2\} \cdot \frac{x'_{t-1} - x_{t-1}}{||x'_{t-1} - x_{t-1}||_2}$. /* Semantic Constraint */

   **return** $x_0, x'_0$.

---

By counting, the step number of the forward or backward process of diffusion-UNet is less than $(2mt_s + 8M)K + t_s + T$. The maximal memory cost of the backward process is $\lceil \frac{t_s}{\lfloor T/K \rfloor} \rceil \times$ the parameters in the feature maps of the diffusion-UNet, combined with other modules, including the surrogate model, the input transformation, and the VAE-decoder in the optimization pipeline. In practice, the time consumption is significantly lower than the upper bound due to the adaptive iteration mechanism, and the lower bound is characterized by:

**Proposition B.1** (Lower Bound of the Diffusion Step). *For ResAdv-DDIM with the total sampling step $T$, the parameter of approximate iterations in $g$ as $K$, the timestep of start adversarial optimization as $t_s$, the lower bound of the diffusion step is:*

$$(\frac{K \cdot t_s}{T} + 3) \cdot \frac{t_s}{2} + T, \tag{19}$$

*if $K$ and $t_s$ are set as $K|T$ and $\frac{T}{K}|t_s$.*

*Proof.* From $K \mid T$ and $\frac{T}{K}|t_s$, let $T = K \cdot d$ where $d \in \mathbb{Z}^+$, $t_s = d \cdot m$ where $m \in \mathbb{Z}^+$. Under the optimal implementation, the forward process in the early-stop judgment and the optimization step are reused. Consider the case of always early-stopping, the total number of approximate iterations in $g$ is $\sum_{t=1}^{t_s} \lceil \frac{t}{\lfloor T/K \rfloor} \rceil$, we have:

$$\sum_{t=1}^{t_s} \lceil \frac{t}{\lfloor T/K \rfloor} \rceil = \sum_{t=1}^{dm} \left\lceil \frac{t}{d} \right\rceil = \sum_{k=1}^{m} \sum_{t=1}^{d} \left\lceil \frac{kd + t - d}{d} \right\rceil = \sum_{k=1}^{m} (d \cdot k) = \frac{dm(m+1)}{2}. \tag{20}$$

By combining the total denoising step of $x_{\text{adv}}$ and $x_{\text{exemplar}}$ generation, the total step is:

$$\frac{dm(m+1)}{2} + t_s + T = \frac{t_s}{2} \cdot (\frac{K \cdot t_s}{T} + 1) + t_s + T = (\frac{K \cdot t_s}{T} + 3) \cdot \frac{t_s}{2} + T. \tag{21}$$

This completes the proof. □

## B.2 3D Object Generation

**Base 3D Generation Method (Trellis)** : Representing 3D data as matrices is inefficient and impractical due to computational problems. Our method is developed based on the *Trellis* model, which bridges the gap between 3D structure and the diffusion process with the structured latent (SLAT). The overall generation process could be formulated as:

$$\begin{aligned}
\mathbf{z_0^{slat}} &= \text{Diffusion Sampling}(\epsilon_{slat}, \mathbf{z_t^{slat}}, \text{Text}), \mathbf{z_T^{slat}} \sim \mathcal{N}(0, I) \\
\mathbf{pos} &= \text{Coords}(\mathcal{D}_{slat}(\mathbf{z_0^{slat}})), \ \text{Coords} : \mathbb{R}^{b \times h \times w \times d} \to \mathbb{R}^{b \times n \times 2} \\
\mathbf{z_0} &= \text{Diffusion Sampling}(\epsilon, \mathbf{z_T}, \mathbf{pos}, \text{Text}), \mathbf{z_T} \sim \mathcal{N}(0, I) \\
\text{Model}_{\text{GS}} &= \mathcal{D}_{\text{GS}}(\mathbf{z_0}, \mathbf{pos}), \ \mathcal{D}_{\text{GS}} : \{(z^i, pos^i)\}_{i=1}^{L} \to \{\{(x_i^k, c_i^k, s_i^k, \alpha_i^k, r_i^k)\}_{k=1}^{K}\}_{i=1}^{L} \\
x &= \text{Renderer}_{\text{GS}}(\text{Model}_{\text{GS}}, \text{Camera}).
\end{aligned} \tag{22}$$

where $\mathbf{z_0}$ and $\mathbf{z_0^{slat}}$ are latents sampled by the diffusion model, and are represented by sparse and dense tensors, respectively. $\mathcal{D}_{slat}$ is the coarse structure decoder, Coords transforms the voxel to point positions $\mathbf{pos}$, $\mathcal{D}_{\text{GS}}$ is the refined structure decoder that decodes each vertex into multiple Gaussian points, and $\text{Renderer}_{\text{GS}}$ renders the Gaussian model to 2D images $x$ with the camera parameter. Since the refined position is also encoded in $\mathbf{z_0}$, we implement the proposed ResAdv-DDIM with the noise estimation network $\epsilon$ that models the refined structure.

$\text{Renderer}_{\text{GS}}$ is the Gaussian renderer proposed in Gaussian splatting. Due to its advantage in optimization, we select this representation as the intermediate 3D data structure for gradient estimation. Specifically, the 3D Gaussian with center point $p$ can be expressed as the Gaussian function:

$$G(p) = e^{-\frac{1}{2} p^T \Sigma^{-1} p} \tag{23}$$

To get a 2D projection image $x$ from a 3D Gaussian point in a world coordinate with a viewing transformation $W$, the 2D covariance matrix $\Sigma'$ as:

$$\Sigma' = JW\Sigma W' J', \tag{24}$$

where $J$ is the Jacobi affine approximation matrix of the transformation matrix. To render the entire Gaussian model, the color of every pixel in the 2D image $x$ is computed by blending $N$ ordered points overlapping the pixel using the following equation:

$$C = \sum_{i \in N} c_i \alpha_i \prod_{j=1}^{i-1} (1 - \alpha_j), \qquad (25)$$

where $c_i$ is the color of each point evaluated by spherical harmonics (SH) color coefficients and $(\alpha_i)$ is determined by a 2D Gaussian with covariance $\Sigma$ and optimizable per-point opacity.

**Residual Approximation with EoT**  We adapt the expectation-over-transformation to bridge the 2D and 3D adversarial generation. By integrating the renderer, the residual approximation $g$ and the adversarial optimization are represented as:

$$g_\theta^{(t,\text{Text})}(z_t, \mathbf{pos}, \text{Camera}) :=$$
$$\text{Renderer}_{\text{GS}} \left( \mathcal{D}_{\text{GS}}(f_{\theta,\Delta T_1}^{(\Delta T_1, \text{Text})} \circ \cdots \circ f_{\theta,\Delta T_k}^{(t,\text{Text})}(z_t, \mathbf{pos}), \mathbf{pos}), \text{Camera} \right) \qquad (26)$$
$$z_{t-\Delta T} := f_{\theta,\Delta T}^{(t,\text{Text})} \left( \arg\max_{z_t} \mathbb{E}_{\text{Camera} \sim P_{\text{Cam}}} \left[ \mathcal{L}_{\text{ATK}}(\mathcal{M}(g_\theta^{(t,\text{Text})}(z_t, \text{Camera}, \mathbf{pos}))) \right] \right)$$

The camera settings are sampled based on the original configuration in *Trellis* framework that defines $P_{\text{Cam}}$, *i.e.*:

$$\Delta_{\text{yaw}} \sim \mathcal{U}\left(-\frac{\pi}{4}, \frac{\pi}{4}\right),$$
$$\Delta_{\text{pitch}} \sim \mathcal{U}\left(-\frac{\pi}{4}, \frac{\pi}{4}\right),$$
$$\mathbf{eye} = 2 \cdot \begin{bmatrix} \sin(\theta_{\text{yaw}} + \Delta_{\text{yaw}}) \cos(\theta_{\text{pitch}} + \Delta_{\text{pitch}}) \\ \cos(\theta_{\text{yaw}} + \Delta_{\text{yaw}}) \cos(\theta_{\text{pitch}} + \Delta_{\text{pitch}}) \\ \sin(\theta_{\text{pitch}} + \Delta_{\text{pitch}}) \end{bmatrix},$$
$$\mathbf{R} = \text{LookAt}(\text{From}=\mathbf{eye}, \text{To}=(0,0,0), \text{UpAxis}=Z),$$
$$\text{Extrinsics} = \begin{bmatrix} \mathbf{R} & \mathbf{t} \\ \mathbf{0} & 1 \end{bmatrix}, \qquad (27)$$
$$\text{Camera} = [\text{Extrinsics}, \text{Intrinsics}_{fov=40°}],$$

where the eye position $\mathbf{eye}$ is sampled on the sphere with $r = 2$, and the camera is toward the coordinate origin. Based on the practice of expectation-over-transformation(EoT) [29], the inner-loop optimization is performed by averaging the gradient from different sampled cameras:

$$Cam[1, 2, ..., E] \leftarrow \text{Sample}(P_{\text{Cam}}).$$
$$grad = \frac{1}{E} \sum_{i=1}^{E} \nabla_{z_t} \mathcal{L}_{\text{ATK}}(\mathcal{M} \circ g_\theta^{(t,\text{Text})}(z_t, \mathbf{pos}, \text{Camera})) \qquad (28)$$
$$v_z \leftarrow \beta v_x + (1 - \beta) \cdot grad$$
$$\mathbf{z}_t \leftarrow \mathbf{z}_t + s * v_z,$$

where $E$ is the step of EoT with the default value 1. For the optimization-related configurations, we maintained most of the parameters in 2D generation: $\epsilon = 10$, $K = 4$, $M = m = 30/E$, $\xi_1 = \xi_2 = 0.01$, $\beta = s = 0.5$ and $t_s = 0.75T$. The target label delay update mechanism is not applied. Diffusion-related configurations are coherent with the original pipeline, *i.e.*, $T = 50$.

**Overall time Consumption**  We conducted a timing analysis on a single GPU using 100 labels from the ImageNet-Label dataset. The results are shown in Tabel 7. Gradient computation and residual approximation account for the majority of the computational cost. Meanwhile, *EoT* sampling is implemented via multi-view rendering using the Gaussian splatting renderer, while the rendering parameter computation does not occupy CUDA execution time. The variation in generation time arises from the adaptive number of optimization steps and differences in the size of sparse tensors in the Trellis framework. Specifically, in single iteration, fluctuation in computation speed is primarily caused by inconsistent data sizes in the Sparse Tensor. For the entire optimization, the fluctuation in

Table 7: GPU Time Consumption for Adversarial Optimization

| Process | Single Iteration (ms) | Entire Optimization (ms) |
|---------|----------------------|--------------------------|
| Denoise Sampling | $123.321 \pm 36.878$ | $2994.933 \pm 269.841$ |
| Residual Approximation | $147.256 \pm 76.243$ | $7040.238 \pm 8915.112$ |
| Gaussian Decoding | $21.547 \pm 11.868$ | $1029.394 \pm 1185.636$ |
| Gaussian Rendering | $28.556 \pm 7.706$ | $1364.221 \pm 1732.996$ |
| Surrogate Model | $6.777 \pm 0.745$ | $323.782 \pm 423.503$ |
| Backward Process | $208.471 \pm 78.039$ | $6261.577 \pm 12119.871$ |

entity, physical_entity, object, ungulate, whole, animal, organism, vertebrate, vascular_plant, instrumentality, mammal, placental, carnivore, vehicle, herb, self-propelled_vehicle, amphibian, canine, domestic_animal, electronic_equipment, device, container, covering, conveyance, commodity, monkey, abstraction, consumer_goods, structure, invertebrate, artifact, ruminant, invertebrate, matter, wheeled_vehicle, arthropod, causal_agent, reptile, equipment, implement, even-toed_ungulate, garment, game, diapsid, primate, protective_covering, relation, restraint, ape, natural_object, psychological_feature, geological_formation, attribute, starches, obstruction, aquatic_vertebrate, old_world_monkey, process, barrier, new_world_monkey, substance, communication, establishment, feline, tool, clothing, food, solid, piece_of_cloth, brass, screen, shelter, grouse, machine, vessel, craft, arachnid, fabric, durables, thing, place_of_business, reproductive_structure, plant, event, material, fastener, woody_plant, measure, home_appliance, mechanism, seafood, cognition, part, organ, group, game_equipment, shape, rodent, military_vehicle, area, mechanical_device, substance, nutriment, amphibian, salamander, support, produce, natural_elevation, mollusk, crustacean, aquatic_mammal, signal, indefinite_quantity, act, public_transport, hand_tool, medium, box, state, kitchen_appliance, edible_fruit, toiletry, shellfish, ware, utensil, fur, foodstuff, cloak, big_cat, footwear, ball, instrument, person, measuring_instrument, sports_equipment, stick, worker, insect, computer, lepidopterous_insect, vine

Figure 14: Overly polysemous tags filtered out

computation speed is also attributed to the adaptive optimization steps. Specifically, the average count of optimization steps for the inner optimization is: 47.810 (std=63.566). Attributed to the proposed early stopping mechanism, easier attack cases require fewer iterations and thus less time. The large variance in generation time reflects significant differences in attack difficulty across tasks, which highlights the effectiveness of our adaptive step design.

### B.3 Construction and Analysis of Semantic-Abstracted Evaluation Task

**Abstracted Label Set Construction**  As described in the main paper, we construct the coarse label set based on three steps: (1) hyponymic graph based on the hyponymic relation defined by *WordNet*, (2) select the proper abstraction level, (3) define the attack goal as the evasion attack on the abstracted label. Specifically, the hyponymic graph is defined as a directed graph $G = (V, E)$, $E \subset V \times V$, where each vertex $V$ represents a word, each edge $e : v_1 \rightarrow v_2$ denotes that the word $v_2$ is a hypernym of $v_1$. We denote the ImageNet label set as the vertex set $\mathbb{L} \subset V$, and construct the subgraph $G' = (V', E')$ as

$$V' = \mathrm{TC}_G(L) := \left\{ v \in V \mid \exists u \in L, \exists k \in \mathbb{N}, (u \xrightarrow{k} v) \in E^+ \right\}$$
$$G' := (V', E') \rangle \quad \text{where} \quad E' = \{(u, v) \in E \mid u, v \in L'\} \tag{29}$$

Where TC denotes the transitive closure and $G'$ is the induced subgraph. Next, we select the abstraction level. We first annotated and filtered out the following over-coarse labels from the vertex set $V'$, resulting in the label set $L'$. We also filter out polysemous tags listed in Figure 14.

Then, for each linear path, we select the node with the lowest height (or has more than one direct hyponym) as a candidate label, constraining the upper bound of the abstracting level, and eliminate

descendant labels of the candidate labels to constrain the lower bound of the abstracting level. These labels are represented as the AbstractedLabel. We construct the SemanticAE generation task by evading the AbstractedLabel, which is formulated as:

$$
\begin{aligned}
&find\ x_{\text{adv}} \in \mathcal{S}(\text{Text})\ \text{s.t.}\ \mathcal{M}(x_{\text{adv}}) \in A_{\text{Text}}, \\
&\quad \text{Text} := \text{"Realistic image of [AbstractedLabel], specifically, [label]"}, \\
&\quad A_{\text{Text}} := \{\text{label}_{\text{Adv}} \mid \text{AbstractedLabel} \notin \textbf{Ancestors}(\text{label}_{\text{Adv}})\}, \\
&\text{AbstractedLabel} \in \{c \in \mathbb{L}' \mid \exists l \in \mathbb{L}\ \text{s.t.}\ c \in \textbf{Ancestors}(l) \wedge \textbf{Count}_{\textbf{Children}}(c) > 0\},
\end{aligned}
\tag{30}
$$

For simplicity, we use the term $c \in \textbf{Ancestors}(l)$ to denote there exists a path from $l$ to $c$, and the term $\textbf{Count}_{\textbf{Children}}(c) > 0$ to denote the in-degree of $c > 0$. We select the abstracted label with more than 3 hyponym Image labels as the label set in the experiment. The abstracted labels and the corresponding hyponym imagenet labels are shown in Table 9.

**Discussions on the ASR$_{\text{relative}}$ metric.** We design $ASR_{\text{relative}}$ to match the goal of facilitating the red-teaming framework, as described in Section 3.1 of the main paper The following proposition describes the relation between the $ASR_{\text{relative}}$ evaluation metrics of SemanticAE generator and the application scenario of the multi-round reject-sampling-based data generation pipeline.

**Proposition B.2** ($ASR_{\text{relative}}$ *characterize the upper-bound probability of the successful attack*)**.** *For any adversarial sampling algorithm $K$ that generates the SemanticAE with $ASR_{\text{relative}} = p$ on the evaluated black-box model $\mathcal{M}_T$ and the surrogate model $\mathcal{M}_S$, there exists an attack algorithm that **achieves the successful attack with at least probability** $p \cdot (1 - \epsilon)$ on $\mathcal{M}_T$, the average generation times less than $1/p_s$, and the maximal generation times $\left\lceil \frac{\log(\epsilon)}{\log(1-p_s)} \right\rceil$, for any $0 < \epsilon < 1$, if the following assumption holds true:*

1. ***Non-Positive Attack**: For the sample generated from the instruction* Text *and the generation algorithm does not perform adversarial optimization towards misleading $\mathcal{M}$ towards the labels corresponding to* Text*, the correct classification leads to semantic alignment,* i.e., *$\mathcal{M}(x) \in L_{\text{Text}} \to (x + \delta) \in \mathcal{S}(\text{Text})$, where $L_{\text{Text}}$ is the correct label corresponding to* Text *and $L_{\text{Text}} = \overline{A_{\text{Text}}}$, and $\delta$ is a small perturbation with $\delta > \langle x_{\text{exemplar}}, x_{\text{adv}} \rangle$.*

2. *The sampling algorithm can generate a sample satisfying $\mathcal{M}(x) \in L_{\text{Text}}$ at least a probability of $p_s$ by only accessing the instruction* Text*.*

3. ***Adjacency Assumption**: $P(\mathcal{M}_T(x_{\text{adv}}) \in L_{\text{Text}} \mid \mathcal{M}_T(x_{\text{exemplar}}) \in L_{\text{Text}}) > P(\mathcal{M}_T(x_{\text{adv}}) \in L_{\text{Text}} \mid \mathcal{M}_S(x_{\text{exemplar}}) \in L_{\text{Text}})$, where $\mathcal{M}_S$ is the surrogate model, $\mathcal{M}_T$ is the target model.*

4. ***Consistency Assumption**: $ASR_{\text{relative}} = P(\mathcal{M}_T(x) \in A_{\text{Text}} \mid \mathcal{M}_T(x) \in L_{\text{Text}})$ for the instruction* Text *given in the attack scenario.*

*Proof.* We construct the *Las Vegas*-style sampling algorithm with the surrogate model $\mathcal{M}_S$. The sampling algorithm is re-run if $\mathcal{M}_S(x_{\text{exemplar}}) \notin L_{\text{Text}}$, until it reaches the upper-bound iteration $\left\lceil \frac{\log(\epsilon)}{\log(1-p_s)} \right\rceil$.

For the case of $\mathcal{M}_S(x_{\text{exemplar}}) \in L_{\text{Text}}$, based on assumption (1), $x_{\text{adv}} \in \mathcal{S}(\text{Text})$. Based on assumptions (3) and (4), we have

$$
\begin{aligned}
P(\mathcal{M}_T(x_{\text{adv}}) \in A_{\text{Text}}) &= P(\mathcal{M}_T(x_{\text{adv}}) \in A_{\text{Text}} \mid \mathcal{M}_S(x_{\text{exemplar}}) \in L_{\text{Text}}) \\
&= 1 - P(\mathcal{M}_T(x_{\text{adv}}) \in L_{\text{Text}} \mid \mathcal{M}_S(x_{\text{exemplar}}) \in L_{\text{Text}}) \\
&> 1 - P(\mathcal{M}_T(x_{\text{adv}}) \in L_{\text{Text}} \mid \mathcal{M}_T(x_{\text{exemplar}}) \in T_{\text{Text}}) \quad \text{(by Assumption (3))} \\
&= P(\mathcal{M}_T(x_{\text{adv}}) \in A_{\text{Text}} \mid \mathcal{M}_T(x_{\text{exemplar}}) \in T_{\text{Text}}) \\
&= ASR_{\text{relative}} = p \quad \text{(by Assumption (4))}
\end{aligned}
\tag{31}
$$

Therefore, with probability $p$, $x_{\text{adv}}$ is a successful attack. Since $P(\mathcal{M}_S(x_{\text{exemplar}}) \in L_{\text{Text}}) > p_s$, the final success attack rate is :

$$
P_{\text{final}} = p \cdot (1 - (1 - p_s)^{\left\lceil \frac{\log(\epsilon)}{\log(1-p_s)} \right\rceil}) > p \cdot (1 - (1 - p_s)^{log_{1-p_s}\epsilon}) = p \cdot (1 - \epsilon)
\tag{32}
$$

The expected execution time of the sampling is:

$$\mathbb{E}[T] = \sum_{k=1}^{\lceil \frac{\log(\epsilon)}{\log(1-p_s)} \rceil} (1-p_s)^{k-1} p_s < \sum_{k=1}^{\infty} (1-p_s)^{k-1} p_s = p_s \cdot \frac{1}{(1-(1-p_s))^2} = \frac{1}{p_s}$$

This completes the proof. □

We acknowledge that the evaluation might still not be adequate as it requires the assumption of *Non-Positive Attack*. However, the defect of the original non-reference evaluation is already shown in our experiments (detailed in Appendix D.4).

## C  Detailed Experiment Settings

### C.1  2D Experiment Settings

**Baseline Descriptions**   Our baseline method is constructed based on the categorization of the transfer attack method in Appendix A.3 and based on the discussions in Appendix A.2. Since the proposed module belongs to the intersection of the diffusion-based adversarial attack and the optimization methodology, we select the classical MI-FGSM [5], which is the base method of recent transfer attacks, and three diffusion-based adversarial optimization methods that are recently proposed and are suitable for SemanticAE , including AdvDiff [16], VENOM [17] and SD-NAE [15]. SD-NAE applies the gradient back-propagation optimization over the full diffusion steps, with the optimization formula as follows. It belongs to the first optimization paradigm discussed in Section 3.2

$$\max_{\text{TextEmbedding}} \mathcal{L}_{\text{ATK}}(\mathcal{M}(\underbrace{f_{\theta,\Delta T}^{\text{TextEmbedding}} \circ f_{\theta,\Delta t}^{\text{TextEmbedding}} \circ \cdots \circ f_{\theta,\Delta t}^{\text{TextEmbedding}}}_{T/\Delta T \text{ times}}(x_T))), \quad (33)$$

AdvDiff and VENOM alter the latent embedding $x_t$ during the diffusion denoising process, and belong to the second optimization paradigm discussed in Section 3.2. Compared to AdvDiff, VENOM applied the additional momentum mechanism and the early-stopping mechanism to the optimization process. Also, it tries to resample $x_T$ based on the adversarial direction of $x_0$ when the optimization fails on the surrogate model. Our method, besides the technical improvement discussed in the main paper, adds a more fine-grained scheduling of optimization steps based on the new optimization paradigm. And we do not add the resample mechanism since it is not redundant if our method is used as a sampling module in an attack/data generation system that could perform reject sampling based on the application scenarios.

We integrate an input-transformation-based method (DeCoWA [45]) as the surrogate model to evaluate the collaboration capability of our method and other methods. We set the *num_warping=2* for diffusion-based and our attacks, and *num_warping=10* for MI-FGSM. The latter setting is consistent with the overall attack pipeline evaluated in the *DeCoWA* paper.

For the diffusion baselines, AdvDiff adapts the classifier guidance and takes the image class as the input of the diffusion model, while VENOM and SD-NAE adapt the classifier-free guidance. SD-NAE alters the selected text embedding by solving the maximization problem described in the main paper, and therefore $\max \mathcal{L}_{\text{ATK}}(\mathcal{M}(f_{\theta,\Delta T} \circ f_{\theta,\Delta t} \circ \cdots \circ f_{\theta,\Delta t}(x_T)))$, Advdiff adapts the approximated optimization $'_{t-\Delta T} = f_{\theta,\Delta T}(\arg\max_{x'_t} \mathcal{L}_{\text{ATK}}(\mathcal{M}(x'_t)))$, and VENOM introduces conditional optimization and momentum mechanisms on it to stabilize the optimization. The diffusion model applied in VENOM and SD-NAE is *bguisard/stable-diffusion-nano-2-1*, and *latent-diffusion/cin256-v2* is applied for AdvDiff. We implemented our code based on the baselines VENOM and SD-NAE, and adapted the same diffusion model for consistent evaluation. In principle, our method can scale to larger models and is evaluated on the *Trellis* 3D generation model.

**Surrogate and Target Models**   As described in the main paper, we adapt the target model set as $\mathcal{T} = \{$*ResNet50* [49], *ViT-B/16* [50], *ConvNeXt-T* [51], *ResNet152*, *InceptionV3* [52], *Swin-Transformer-B* [53]$\}$, and the surrogate model set as {ResNet50, ViT-B/16, ConvNext-T, ResNet50+DeCoWA}.

**Loss Function**   We employ the same loss function as the original implementation for the baseline methods. For our model, we set the loss function in both 2D and 3D SemanticAE generation task as:

$$\mathcal{L}_{\text{Atk}}(\text{logits}, A_{\text{Text}}, L_{Tar}) = \text{LogSoftMax}(\text{logits})[L_{\text{Tar}}] - \frac{1}{|A_{\text{Text}}|} \sum_{i \in A_{\text{Text}}} \text{LogSoftMax}(\text{logits})[i],$$
(34)

Where $L_{\text{Tar}}$ is the currently selected label (the label with the highest confidence in the set $A_{Text}$), and log softmax denotes $\log \frac{e^x}{\sum e^x}$.

**Image Quality Assessment Metrics**   To evaluate semantic constraints, the pairwise semantic metric is proposed to measure the similarity between $x_{\text{exemplar}}$ and $x_{\text{adv}}$, which defines as follows in the main body of our work:

$$\text{SemanticDiff}_{\mathcal{S}} = \langle x_{\text{exemplar}}, x_{\text{adv}} \rangle_{\mathcal{S}},$$
(35)

$\mathcal{S}$ is a visual similarity metric; we employed LPIPS and MS-SSIM for evaluation. Parameters of the evaluation metrics are adapted as common practice. For MS-SSIM, we adapt $kernel size = 11$ and $\sigma = 1.5$. For LPIPS, we adapt *AlexNet* as the local feature extractor. For $\text{Clip}_Q$, we use the implementation of *piq* [81] and adapt *openai/clip-vit-base-patch16* as the image embedding extractor.

## C.2   3D Experiment Settings

This section details the comprehensive framework established for the evaluation of 3D video generation models, encompassing dataset preparation, video synthesis methodologies for both benign and adversarial examples, and the metrics employed for performance assessment.

**Dataset Preparation**   The ImageNet dataset, while extensive, contains labels with fine-grained semantic distinctions that can be challenging for text-to-3D video generation models to differentiate effectively. A coarse-graining procedure was applied to the Original Imagenet labels to address this.

Specifically, a predefined mapping, detailed in Table 9, was utilized to merge semantically similar labels. This process involved replacing the Original ImageNet labels with their corresponding Abstracted Labels. The resultant collection of these processed Abstracted Labels served as the prompt dataset for the subsequent video generation tasks. Let $\mathbb{L}$ be the set of Original ImageNet labels and $\mathcal{T}_{coarse}$ be the set of Abstracted Labels. The mapping function $M : \mathbb{L} \rightarrow \mathcal{T}_{coarse}$ transforms each Original ImageNet label to its Abstracted Label. The set of prompts used for generation is $\mathcal{P} = \{t | t \in \mathcal{T}_{coarse}\}$.

**3D Video Generation**   Two categories of video samples were generated: clean samples and adversarial examples. Clean video samples were synthesized using the TRELLIS[43] model. For each prompt $p \in \mathcal{P}$, a corresponding clean video $V_{clean}$ was generated. Adversarial examples were generated based on the TRELLIS[43] model (version *TRELLIS-text-base*), employing a ResNet-50 model, pre-trained on ImageNet, as the surrogate model for guiding the adversarial attack. The generation of these adversarial examples was performed using our proposed methodology. During each generation instance, an abstracted text label $p \in \mathcal{P}$ was used as the input prompt. The target label for the attack was set to any Original ImageNet label $l_{target} \in \mathbb{L}$ such that $M(l_{target}) = p$. A constant perturbation strength, denoted as $\epsilon$, was maintained at 10.0 across all adversarial generation processes. All videos are captured by the original rendering pipeline, *i.e.*, a camera surrounding the object.

**Evaluation Metrics**   The evaluation of the generated videos involved a frame-by-frame analysis using a pre-trained ResNet50 classifier.

For a given video $V$, consisting of $N$ frames $\{f_1, f_2, \ldots, f_N\}$, each frame $f_i$ was individually classified by the ResNet-50 model. This yields a sequence of Original Imagenet labels, $c_i = \text{ResNet50}(f_i)$. Each $c_i$ was then mapped to its Abstracted Label $t_i = M(c_i)$. The overall model prediction for the video $V$, denoted as $P_V$, was determined by the mode of these frame-level Abstracted Label predictions:

$$P_V = \text{mode}(\{t_1, t_2, \ldots, t_N\})$$

where $\text{mode}(\cdot)$ returns the most frequently occurring element in the set.

A video $V$ was deemed correctly classified if its model prediction $P_V$ matched its ground truth Abstracted Label $G_V$. It was observed that, under certain parameter configurations (particularly for varying $K$), a minority of video generation attempts might fail. To ensure a rigorous and controlled comparison, a data curation step was implemented. The intersection of successfully generated videos across all conditions – clean samples ($V_{clean}$) and all sets of adversarial examples ($V_{adv}^{(0)}, V_{adv}^{(1)}, V_{adv}^{(2)}, V_{adv}^{(3)}, V_{adv}^{(4)}$) – was taken. Only videos present in this intersection were considered for the final evaluation. This ensures that performance metrics are calculated over an identical set of video instances, thus isolating the impact of the varied adversarial generation parameters.

Following the procedures outlined above, the Accuracy (ACC) and Attack Success Rate (ASR) were calculated. The specific mathematical formulations ASR are provided in the main body of this work.

## D Detailed Results and Discussions

### D.1 Transfer Attack Analysis

**Experimental settings** We select VENOM, MI-FGSM, SD-NAE, and AdvDiff as baseline methods, employing four surrogate models: ResNet50, DeCoWa, ConvNext-T, and ViT-B/16. Six target models are evaluated: ResNet50, ResNet152, ConvNext-T, ViT-B/16, Swin-B, and InceptionV3. For Abstracted Label tasks, our method adopts four different perturbations $\epsilon = \{2, 2.5, 3, 4\}$, while Original Imagenet label tasks use $\epsilon = \{1.5, 2, 2.5, 3\}$. For each surrogate-target model pair, adversarial examples are crafted using both our method and baselines. Evaluation metrics include Attack Success Rate (ASR), Accuracy (ACC) and LPIPS (lower values indicate better perceptual quality).

**Data presentation** Due to the inconsistency between different surrogate-target model pairs, we chose to present the experimental results using subplots. The x-axis represents the selected surrogate models, and the y-axis represents the attacked target models, with a total of 24 subplots. The experimental results are shown in Figure 20, 21, 22, 23.Figure 20, 22 display the relationship between LPIPS and ASR/ACC for examples generated by different methods in the abstracted label task. Figure 21, 23 show the relationship between LPIPS and ASR/ACC for examples generated by different methods in the original Imagenet label task. In the figures, data points of different colors represent different methods. The data points of our method under varying perturbation strengths are connected by lines, and its trend is fitted with a black dashed line. As a supplement, the numerical results with the standard deviation of the final metric over 6 random seeds are shown in Table 11 and Table 11, demonstrating the stability of the results.

**Analysis** Based on the observations from Figure 20, 21, the following conclusions can be derived:

- In our method, as the perturbation parameter $\epsilon$ gradually increases, ASR also rises, but the LPIPS increases as well. The relationship between ASR and the Natural logarithm of LPIPS essentially forms a straight line with a positive slope.

- The adversarial examples produced by SD-NAE (yellow markers) exhibit higher LPIPS in almost all cases, indicating that this method introduces more noticeable perturbations. However, compared to other baseline methods, SD-NAE does not always achieve a higher attack success rate.

- The adversarial examples produced by MI-FGSM (purple markers)have lower LPIPS than SD-NAE, but still significantly higher than our proposed attack method.

- The LPIPS of adversarial examples generated by VENOM (green markers) is comparable to our method. However, at similar LPIPS levels, our method achieves a higher ASR in most cases.

- AdvDiff (blue markers) produces adversarial examples with the lowest LPIPS, indicating better stealthiness. However, its attack success rate is significantly lower than all other methods.

**Conclusion** To summarize, **in all 48 settings** of this study, except for two cases (the first row, first column and first row, fourth column in Figure 20) where our method was slightly worse than the VENOM method, the remaining ASR-LPIPS curves of our method were all located above and to the

left of the comparison methods. In Figure 22, 23, all the methods, except for the VENOM method in the two subfigures of Figure 22(the first row and first column, and the first row and fourth column), are on the ACC-LPIPS curves of our method. This means our method achieved higher attack success rates at the same LPIPS levels. These results clearly demonstrate that our method outperforms the comparison methods in most cases **(46/48)**. In addition, these exceptions both occurred in white-box attack scenarios, which were not our primary focus. VENOM achieved higher white-box accuracy by repeatedly resampling through rejection, while we treated this as a module separate from the Adversarial Sampling algorithm.

Therefore, our **InSUR** framework shows significant superiority in adversarial transferability.

### D.2 Attack Robustness Analysis

**Experimental settings** : This experiment discusses the performance of our method compared to baseline methods when facing adversarial defenses. The defense methods used are JPEG and DiffPure. JPEG applies lossy compression to images (with a quality factor of 75 in this experiment), while DiffPure removes adversarial perturbations by feeding samples into a diffusion model for regeneration.

**Data presentation** : Figure 24, 25 show the results of our method and comparison methods on both defended and undefended models. Solid dots represent undefended results, while hollow dots represent defended results, with arrows indicating the impact of applying defenses.

**Analysis** :

- **MI-FGSM**: JPEG is a rule-based defense method, whereas DiffPure is a defense based on the in-manifold assumption. Due to the lack of in-manifold regularization, MI-FGSM-generated adversarial examples are less robust against DiffPure, showing a noticeable performance drop. This can be seen in Figure 24 vs. Figure 25.

- **AdvDiff**: As diffusion-based adversarial example generation method,it exhibit strong robustness against DiffPure (also diffusion-based).Figure 25 shows that their attack success rates (ASR) even increase after DiffPure defense, though they still underperform our method.

- **SD-NAE**: SD-NAE is also a diffusion-based adversarial example generation method, so it exhibits strong robustness against DiffPure. Besides, SD-NAE applies perturbations through text embeddings, which results in better performance in transfer attacks. However, this does not necessarily indicate an advantage, as the visualization results suggest that it may lead to uncontrollable semantic deviations. The high transfer attack success rate could be attributed to inherent changes in global semantics. Additionally, its optimization for white-box attacks is inadequate. For undefended white-box models, the success rate is lower than that of our method and VENOM.

**Conclusion** : Our method shows a decrease in ASR in most cases when facing JPEG and DiffPure defenses, but this effect is limited, and in the face of JPEG defense, our method is the only one able to keep the attack success rate ASR consistently above 80% when the target model is the same as the surrogate model (see Figure 24). When the DiffPure defense leads to a decrease in the attack success rate of our method and an increase in the attack success rate of the SD-NAE method, we are still able to ensure that at least one data point of our method outperforms the SD-NAE (e.g., the subplot in the first column of the fifth row of Figure 25).

In summary, our method enhances the optimization exploration capability for adversarial examples while maintaining In-Manifold Regularization(outperforms the comparison methods all cases **(48/48)**). This ensures that our approach remains effectively aggressive against defense methods.

### D.3 On the Role of Residual-driven Attacking Direction Stabilization in 2D and 3D SemanticAE Applications

**Boosting Multi-diffusion-step Regularized Adversarial Optimization** Recall that, to solve the problem of **collaborating the adversarial optimization with the diffusion model for better transfer attacks and robust attacks**, we introduce the residual approximation in *ResAdv-DDIM*.

We further analyze it in the more refined evaluation tasks. The experiment is conducted with the surrogate model of **ViT-B/16** with 6 different target models on the abstracted label evasion task. The parameters of the maximal approximate iteration $K$ and the $t_s/T = 0.25, 0.5, 0.75$ have been evaluated. The results are shown in Figure 15. Since altering the parameter changes the behavior of the semantic alignment, the semantic difference measurement is included, and the results are plotted in a figure. $K = 0$ indicates the setting without the residual approximation.

The figure shows that:

- When $t_s/T = 0.25$, both LPIPS and ASR are higher, and the introduction of the residual approximation lets the balance point between LPIPS and ASR offset, or more biased towards LPIPS optimization. This is due to (1) since the white box results, shown in the upper left figure, indicate that the successful ASR has already been achieved, and there is no need to improve the performance of adversarial optimization. Therefore, improvement is on LPIPS. (2) Since the residual approximation is not designed for regularizing transfer attacks, the transfer attack performance declines in ResNet and Inception models.

- When $t_s/T \in \{0.5, 0.75\}$, the result is different. Specifically, (1) the white-box results indicate that the adversarial optimization is non-optimal. (2) The residual approximation improves the performance of the white-box attacks and significantly improves the performance of transfer attacks. Improvements exist in both LPIPS and ASR.

- Simply increasing the step (the blue arrows) of undergoing diffusion steps of adversarial optimization may not improve adversarial transferability. However, it may decrease adversarial optimization performance under the same adaptive optimization mechanism, as the diffusion process may purify the adversarial optimization.

- Increasing $t_s$ together with the residual approximation solves the collaboration problem between adversarial optimization and diffusion purification, leading to highly transferable SemanticAE.

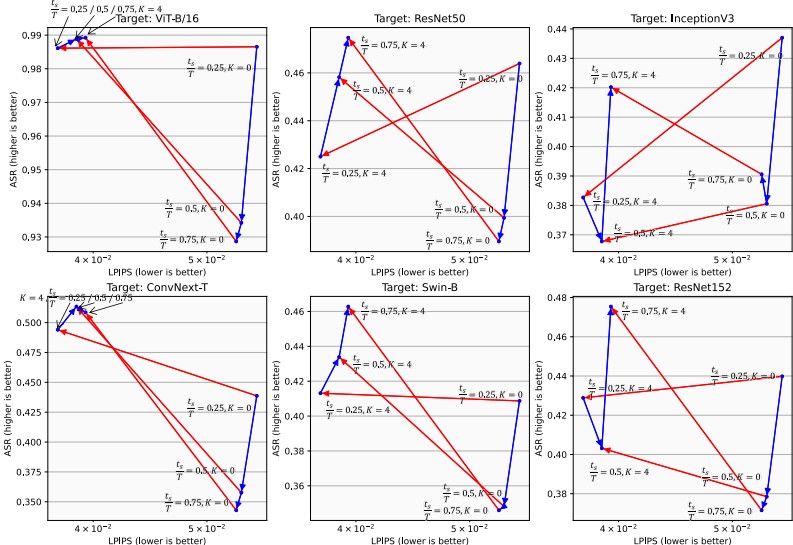

Figure 15: Parameter analysis with surrogate model ViT-B/16. Red arrows denote the performance before and after adding the residual approximation for $g(x)$. In each setting of $\frac{t_s}{T} \geq 0.5$, applying residual approximation achieves significant improvements (The upper left corner indicates a strictly superior direction). **This result is coherent with the design goal of ResAdv-DDIM.**

**Collaboration with EoT** Different from 2D optimization, the global gradient of 3D models cannot be obtained within a single iteration. Therefore, the EoT optimization is applied, accumulating the gradients across different perspectives. This further challenges the adversarial optimization capability. However, as shown in Table 8, with the residual approximation ($K > 0$), our method can collaborate well with $EoT$ with different numbers of EoT steps, resulting in different gradient

optimization step sizes. The visualized comparison is shown in Figure 16 and the supplementary videos in https://semanticAE.github.io. The attack performance on different views is significantly higher than without residual approximation ($K = 0$), showing that the necessity of the residual approximation under diffusion+EoT generation pipeline.

Table 8: Comparison between residual approximation and expectation over transformation (EoT). The total iteration (EoT step * gradient descent step) is consistent. The gradient optimization stepsize is larger if the EoT step is higher.

| EoT step | 5 | 3 | **1** | 1 | 1 | 1 | 1 |
|---|---|---|---|---|---|---|---|
| Residual approximation step (K) | 4 | 4 | **4** | 3 | 2 | 1 | 0 |
| ASR | **0.922** | 0.913 | **0.922** | 0.912 | 0.912 | 0.902 | 0.451 |

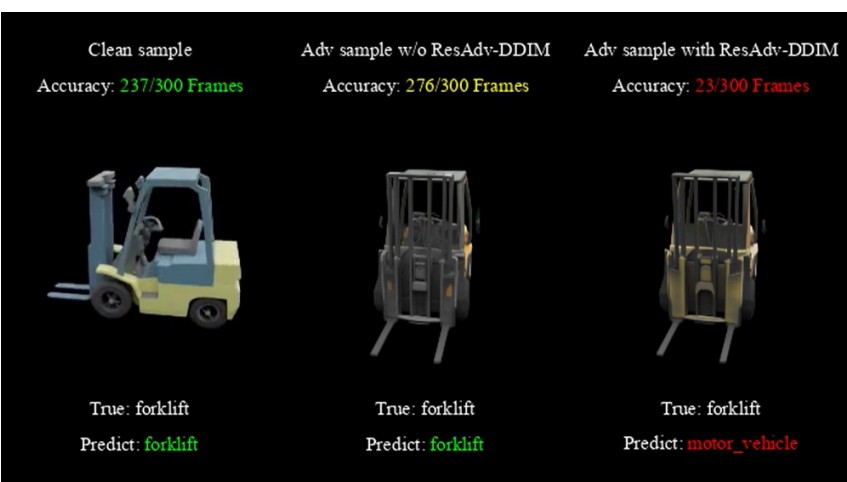

Figure 16: 3D Visual Results Ablation.

## D.4 On the Potential Adversarial Transferability to Semantic Evaluator

We analyze the character between and find the clue of the reference-free semantic evaluation metric being transferred in SemanticAE evaluation. Our experiment is based on hypothesis testing. Specifically, we evaluate the results from our method under the setting of Appendix D.1, which is 16 rounds of generation for each of the original ImageNet label evasion task and the abstracted label evasion task. Additionally, we evaluate the clip-score [82] metric with the settings:

$$Prompt = \begin{cases} \text{ImageNetLabel} & Task = ImageNet \\ \text{AbstractedLabel, ImageNetLabel} & Task = Abstracted \end{cases} \quad (36)$$

The backbone of the clip-score is *ViT-B/32*. Then we apply the linear regression on the clip-score, as the clip Semantic metric, and the LPIPS score on the factor of **whether the surrogate selects ViT-B/16**. The results is shown in Figure 17, Figure 18, and Figure 19. The following hypothesis is rejected with high confidence ($p < 0.02$):

> Under the same generation settings, the CLIP semantic evaluation results are independent of the surrogate model selection.

Due to the high p-value associated with LPIPS, its correlation with surrogate model selection is relatively low. Although there are differences in model configurations and training tasks, both clip-score and the surrogate model adopt the **ViT** architecture. Therefore, we have reason to believe that the transfer attack has affected the evaluation of semantic similarity metrics.

We believe that the potential adversarial transferability to the deep-learning-based semantic evaluation model is difficult to tackle within the models, especially for the potential adversarial example that

could perform weak-to-strong attacks. As discussed in Appendix B.3, our exemplar-based evaluation task provides an alternative way to show the semantic alignment, avoids the requirement of non-referencing semantic evaluation, and directly shows the adversarial capability of the generated data.

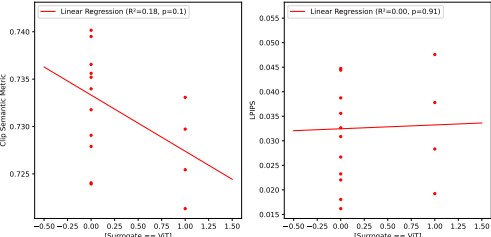

Figure 17: Results on the Original ImageNet Label Evasion SemanticAEs

Figure 18: Results on the Abstracted Label Evasion SemanticAEs

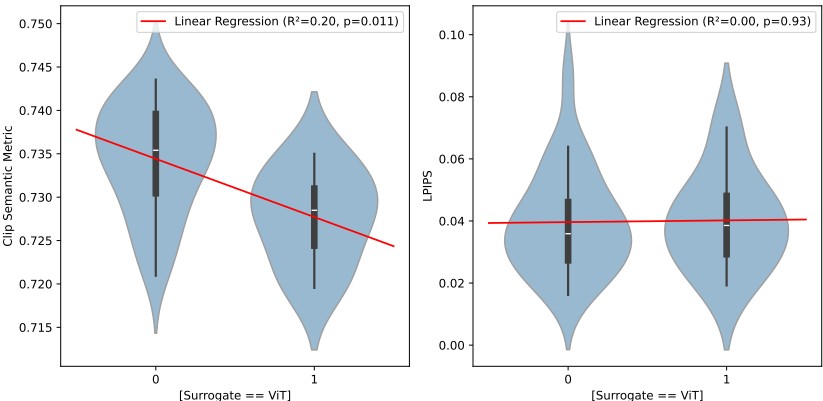

Figure 19: Combined distribution of the semantic metric on the ImageNet and abstracted label evasion SemanticAEs. With high confidence ($p < 0.02$), the *CLIP* semantic metric is affected by the attack transferability.

## E    Results Visualization

### E.1    2D Visualized Results

To intuitively visualize the samples generated under the semantic constraints of original ImageNet label and abstracted label, we select SemanticAEs $x_{\text{adv}}$ and corresponding nearby samples $x_{\text{exemplar}}$ of seven original ImageNet labels (monarch, castle, goldfinch, aircraft carrier, vase, tiger, jellyfish) and six abstracted labels (aircraft, bag, beetle, bird, boat, fish) for visualization, as shown in Figure 26 and 27. The surrogate model is DeCoWa + ResNet. For our attack method. We set different parameters $\epsilon = \{1.5, 2, 2.5, 3\}$ and $\epsilon = \{2, 2.5, 3, 4\}$, respectively, for 2D ImageNet-label evasion attacks and 2D abstracted-label evasion attacks, which determines the strength of the semantic constraint. We also present the generation results of the baselines. For a single image $I$, we report confidence and mark the classification label $y$ with different colors employing ResNet50 and ViT-B/16 as target models. In 2D ImageNet-label evasion attacks, Green is for the same classification label $y$ and ImageNet label corresponding to the semantic constraint; otherwise, red. In 2D Abstracted-label evasion attacks, Green indicates that the classification label $y$ belongs to the abstracted label corresponding to the semantic constraint. Therefore, a successful attack is defined as follows: for a pair of $x_{\text{adv}}$ and $x_{\text{exemplar}}$, $x_{\text{exemplar}}$ is classified correctly (green), while $x_{\text{adv}}$ is classified incorrectly (red).

**Analyzing 2D Visualized Results**    Based on the observations of Figure 26 and  27, the following conclusions can be derived:

- For our attack method, as the parameter $\epsilon$ gradually increases, signifying a progressive weakening of the semantic constraint strength, the number of successful attacks correspondingly increases. However, this also results in unnaturalness of SemanticAEs. For instance, the $x_{\text{adv}}$ with the original ImageNet label "castle" has a castle with a blurred top when $\epsilon = 3$, and the $x_{\text{adv}}$ with the abstracted label "ship" exhibits a blurry mast when $\epsilon = 4$.
- SemanticAEs generated by SD-NAE disturb more global semantics, which leads to semantic drift and dissimilarity between the $x_{\text{exemplar}}$ and $x_{\text{adv}}$. It is consistent with the results shown in Appendix D.1: SD-NAE has a lower MS-SSIM score and a higher LPIPS score, with the same surrogate model, compared with other attack methods.
- As for MI-FGSM, noticeable noise can be observed in the background area of the SemanticAEs, which are not natural. In the main paper, MI-FGSM corresponds to a lower CLIP-QAI score related to noisiness.
- AdvDiff generates adversarial examples with artifacts at the edges in ImageNet-label evasion attacks, such as the $x_{\text{adv}}$ of label "castle" and "aircraft carrier", which is a manifestation of low image quality. In abstracted-label evasion attacks, SemanticAEs hardly attack target models successfully.
- The adversarial examples of VENOM are natural. However, they struggle to effectively attack the ViT-B/16 model.

In conclusion, our method generates local in-manifold patterns to achieve strong attacks.

### E.2  3D Visualized Results

**Image Visualization**   To qualitatively substantiate the efficacy of our proposed methodology, we conducted a visual comparison of the generated video samples. Initially, ten unique labels were randomly selected from the complete set of Abstracted Labels. Subsequently, for each of these selected labels, the corresponding clean video sample and the adversarial video sample generated with $K = 4$ were chosen. From each of these videos, five frames were extracted at equidistant intervals. The visual results of these extracted frames are presented in Figure 28. A comparative analysis of each pair of clean and adversarial examples reveals that the adversarial counterparts maintain a high degree of visual similarity to the benign videos. Despite this perceptual resemblance, the adversarial examples demonstrate a high probability of inducing misclassification by the ResNet-50 model, thereby underscoring the effectiveness of our approach in generating robust yet inconspicuous adversarial attacks.

Furthermore, we performed a comparative analysis of adversarial examples generated under varying $K$ parameter settings to demonstrate the rationale behind our parameter selection. Specifically, we selected the same video instance from the clean samples, the adversarial examples generated with $K = 0$, and those generated with $K = 4$. A single frame was extracted from each of these three video versions for visualization, as depicted in Figure 16. It is observable from the figure that when $K = 0$, the classification outcome for the adversarial example paradoxically improves compared to the clean sample, signifying a failure of the adversarial attack. Conversely, for $K = 4$, the adversarial example exhibits visual characteristics closely resembling those of the $K = 0$ sample. However, its efficacy in deceiving the classifier is significantly enhanced. This comparative visualization corroborates the superiority of our chosen parameter configuration ($K = 4$) in achieving a strong attack effect while preserving visual quality.

**Video Visualization**   We performed a comparative analysis of adversarial examples generated under varying $K$ parameter settings to demonstrate the rationale behind our parameter selection. Specifically, we selected three video instances, identified by their primary content as the *forklift* video, the *llama* video, and the *volcano* video. For each of these videos, we considered the clean sample, the adversarial example generated with $K = 0$ (without residual approximation), and that generated with $K = 4$. The videos are shown in https://semanticAE.github.io.

It is observable from our quantitative analysis that the outcomes for $K = 0$ varied across the different videos. For the *forklift* video, the classification accuracy of the adversarial example generated with $K = 0$ paradoxically improved compared to its clean sample, signifying a failure of the adversarial attack under this specific setting. In contrast, for both the *llama* and *volcano* videos, the adversarial examples generated with $K = 0$ achieved lower classification accuracies than their respective clean samples, indicating some attack effect. However, their accuracies were still notably higher than those

of the adversarial examples generated with $K = 4$. This demonstrates that for the *llama* and *volcano* videos, while $K = 0$ initiated an attack, its deceiving capability was considerably weaker than that of $K = 4$.

Conversely, for $K = 4$, the adversarial examples for all three videos (*forklift*, *llama*, and *volcano*) exhibit visual characteristics closely resembling those of the $K = 0$ samples. However, their efficacy in deceiving the classifier is significantly enhanced across all instances. This comparative visualization and analysis corroborate the superiority of our chosen parameter configuration ($K = 4$) in achieving a strong attack effect while preserving visual quality.

## F    Boarder Impacts

While our goal is to catalyze the development of the red-teaming framework and trustworthy AI, we acknowledge that the proposed technology might be misused, including: (1) extending the proposed transfer attack improvement methods to jailbreak multi-modal LLMs. (2) extending the proposed 3D attack methods to generate physical adversarial examples to attack biometric authentication systems. However, our framework is not directly designed for these scenarios and requires further integration.

To protect from potential attacks in applications, we suggest developing the following closed-loop framework as a complement to traditional defense methods tested in Appendix D:

- Collect data generated by the proposed **InSUR** framework.
- Annotate the data with human feedback or rule-based models.
- Improve the alignment of the multi-modal models through fine-tuning on the dataset.

As a tool for data generation, we believe our framework is more beneficial for the model holder. For responsibility, we will release the code of **InSUR** framework *after* the paper is published for reference.

Table 9: Abstracted Label Mapping from ImageNet Numerical IDs

| Abstracted Label | Original ImageNet Label IDs |
| --- | --- |
| dog | 151, 152, 153, 154, 155, 156, 157, 158, 159, 160, 161, 162, 163, 164, 165, 166, 167, 168, 169, 170, 171, 172, 173, 174, 175, 176, 177, 178, 179, 180, 181, 182, 183, 184, 185, 186, 187, 188, 189, 190, 191, 192, 193, 194, 195, 196, 197, 198, 199, 200, 201, 202, 203, 204, 205, 206, 207, 208, 209, 210, 211, 212, 213, 214, 215, 216, 217, 218, 219, 220, 221, 222, 223, 224, 225, 226, 227, 228, 229, 230, 231, 232, 233, 234, 235, 236, 237, 238, 239, 240, 241, 242, 243, 244, 245, 246, 247, 248, 249, 250, 251, 252, 253, 254, 255, 256, 257, 258, 259, 260, 261, 262, 263, 264, 265, 266, 267, 268 |
| bird | 7, 8, 9, 10, 11, 12, 13, 14, 15, 16, 17, 18, 19, 20, 21, 22, 23, 24, 80, 81, 82, 83, 84, 85, 86, 87, 88, 89, 90, 91, 92, 93, 94, 95, 96, 97, 98, 99, 100, 127, 128, 129, 130, 131, 132, 133, 134, 135, 136, 137, 138, 139, 140, 141, 142, 143, 144, 145, 146 |
| musical_instrument | 401, 402, 420, 432, 486, 494, 513, 541, 546, 558, 566, 577, 579, 593, 594, 641, 642, 683, 684, 687, 699, 776, 822, 875, 881, 889 |
| furnishing | 423, 431, 453, 493, 495, 516, 520, 526, 532, 548, 553, 559, 564, 648, 703, 736, 741, 765, 794, 831, 846, 854, 857, 861, 894 |
| motor_vehicle | 407, 408, 436, 468, 511, 555, 569, 573, 575, 609, 627, 656, 661, 665, 675, 717, 734, 751, 803, 817, 864, 867 |
| snake | 52, 53, 54, 55, 56, 57, 58, 59, 60, 61, 62, 63, 64, 65, 66, 67, 68 |
| fish | 0, 1, 2, 3, 4, 5, 6, 389, 390, 391, 392, 393, 394, 395, 396, 397 |
| building | 410, 425, 449, 497, 498, 580, 598, 624, 663, 668, 698, 727, 762, 832 |
| saurian | 38, 39, 40, 41, 42, 43, 44, 45, 46, 47, 48 |
| ball | 429, 430, 522, 574, 722, 747, 768, 805, 852, 890 |
| headdress | 433, 439, 452, 515, 518, 560, 667, 793, 808 |
| beetle | 300, 301, 302, 303, 304, 305, 306, 307 |
| timepiece | 409, 530, 531, 604, 704, 826, 835, 892 |
| shop | 415, 424, 454, 467, 509, 788, 860, 865 |
| dish | 925, 926, 933, 934, 962, 963, 964, 965 |
| cat | 281, 282, 283, 284, 285, 286, 287 |
| weapon | 413, 456, 471, 657, 744, 763, 764 |
| bottle | 440, 720, 737, 898, 899, 901, 907 |
| fungus | 991, 992, 993, 994, 995, 996, 997 |
| spider | 72, 73, 74, 75, 76, 77 |
| ship | 403, 510, 628, 724, 833, 913 |
| boat | 472, 554, 576, 625, 814, 914 |
| turtle | 33, 34, 35, 36, 37 |
| bag | 414, 636, 728, 748, 797 |
| housing | 500, 660, 663, 698, 915 |
| crab | 118, 119, 120, 121 |
| wolf | 269, 270, 271, 272 |
| fox | 277, 278, 279, 280 |
| bear | 294, 295, 296, 297 |
| aircraft | 404, 405, 417, 895 |
| armor | 465, 490, 524, 787 |
| wheel | 479, 694, 723, 739 |
| fence | 489, 716, 825, 912 |
| personal_computer | 527, 590, 620, 681 |
| roof | 538, 853, 858, 884 |
| overgarment | 568, 617, 735, 869 |
| skirt | 601, 655, 689, 775 |
| bread | 930, 931, 932, 962 |
| squash | 939, 940, 941, 942 |

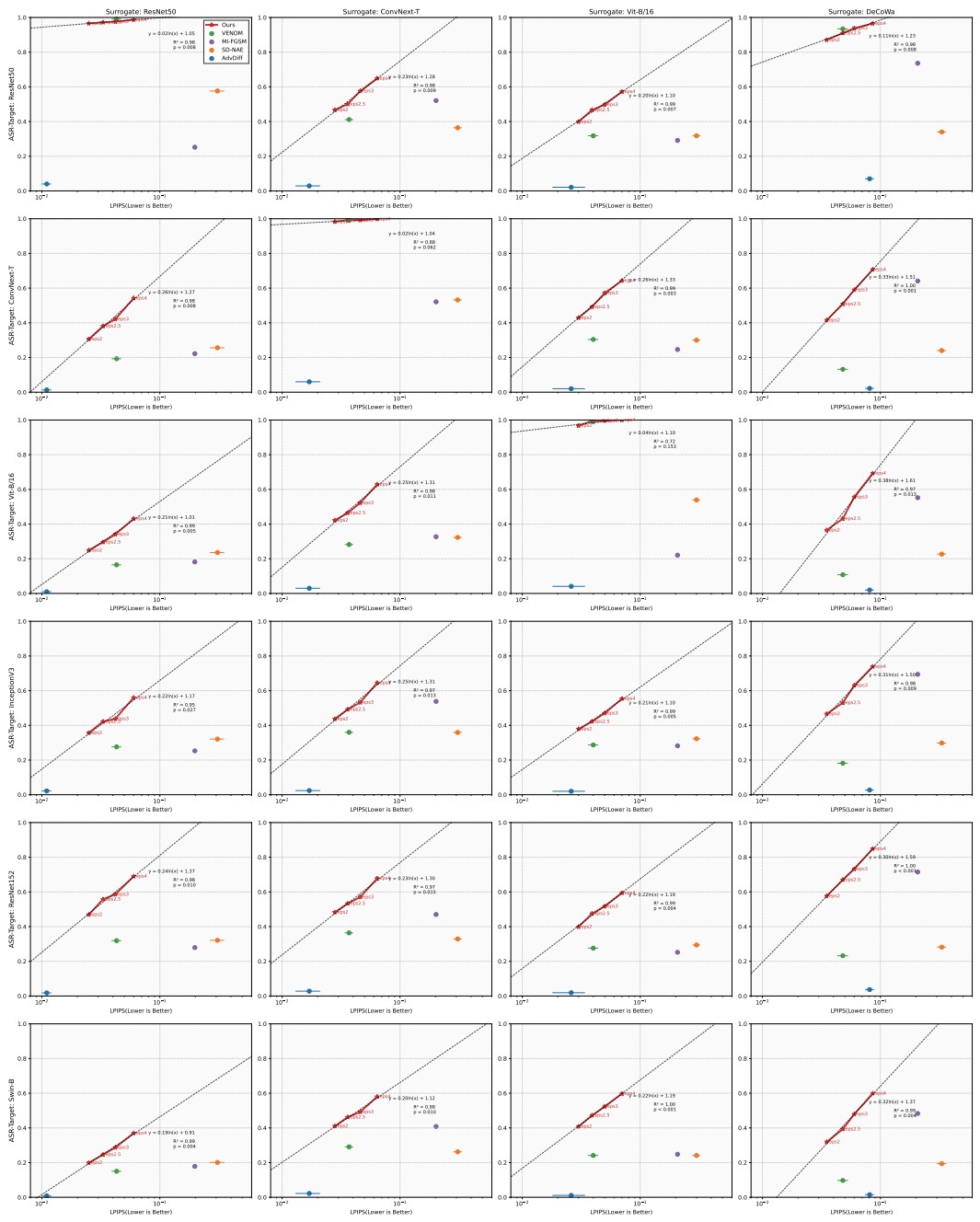

Figure 20: ASR of Abstracted Label

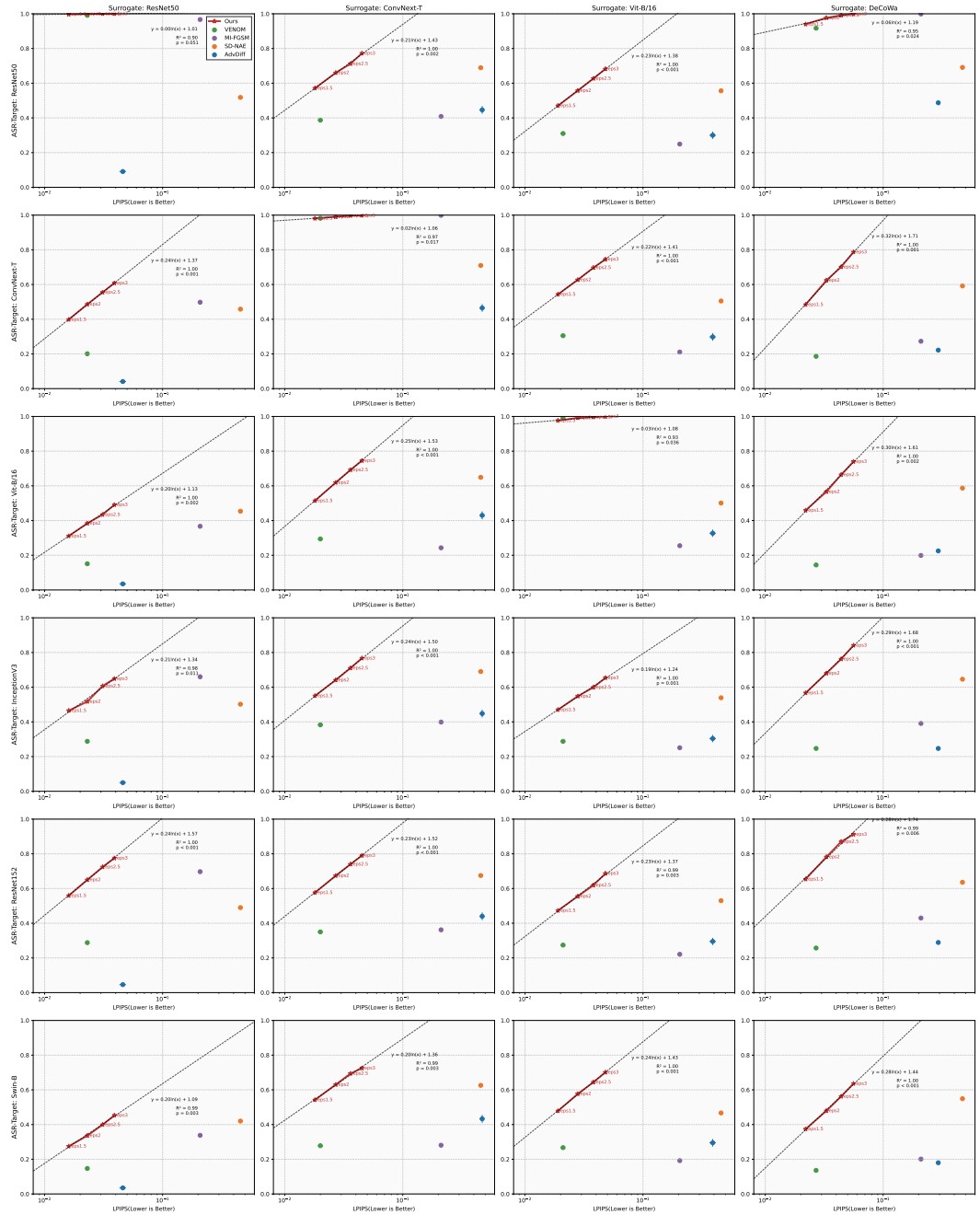

Figure 21: ASR of Original Imagenet label

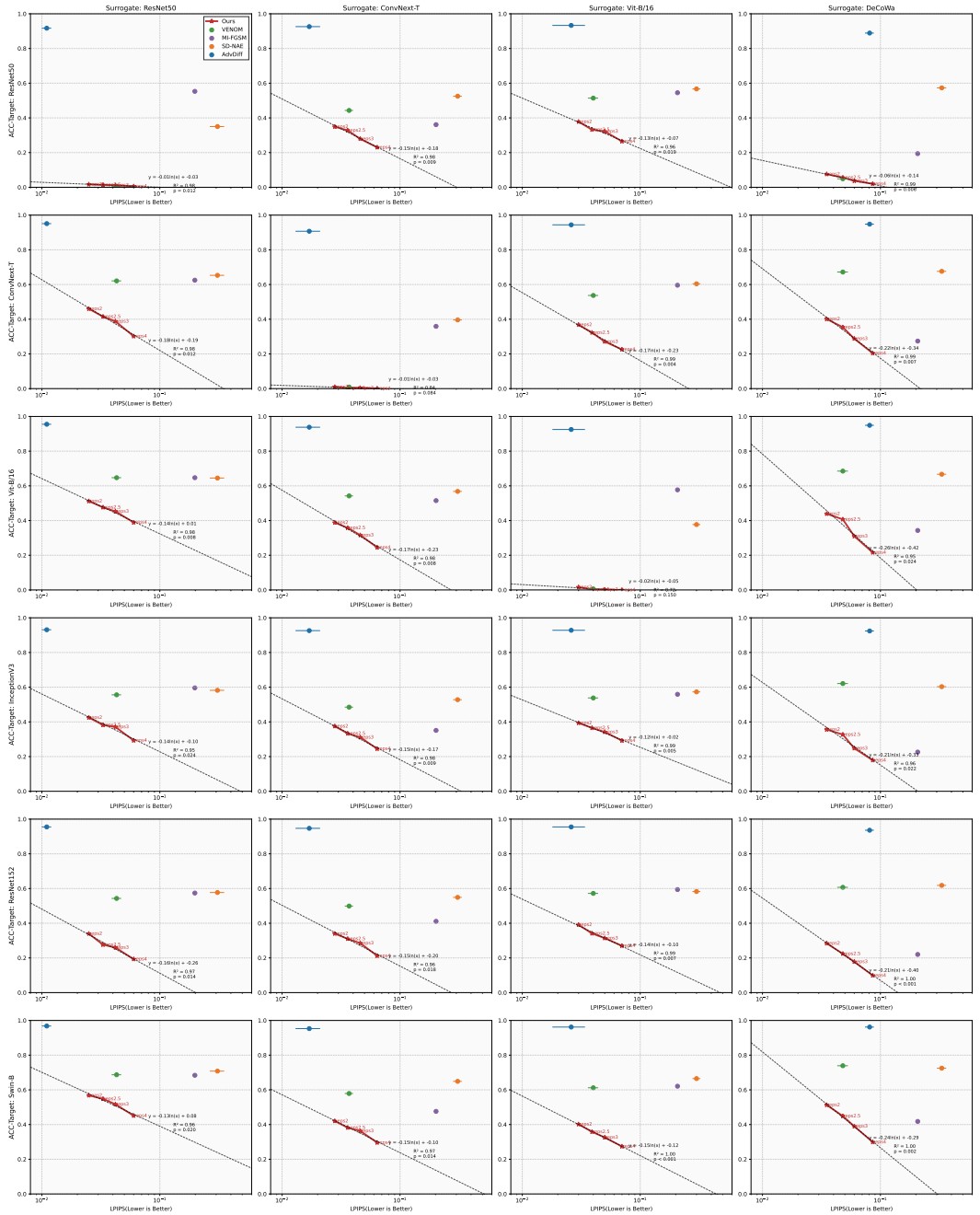

Figure 22: ACC of Abstracted Label

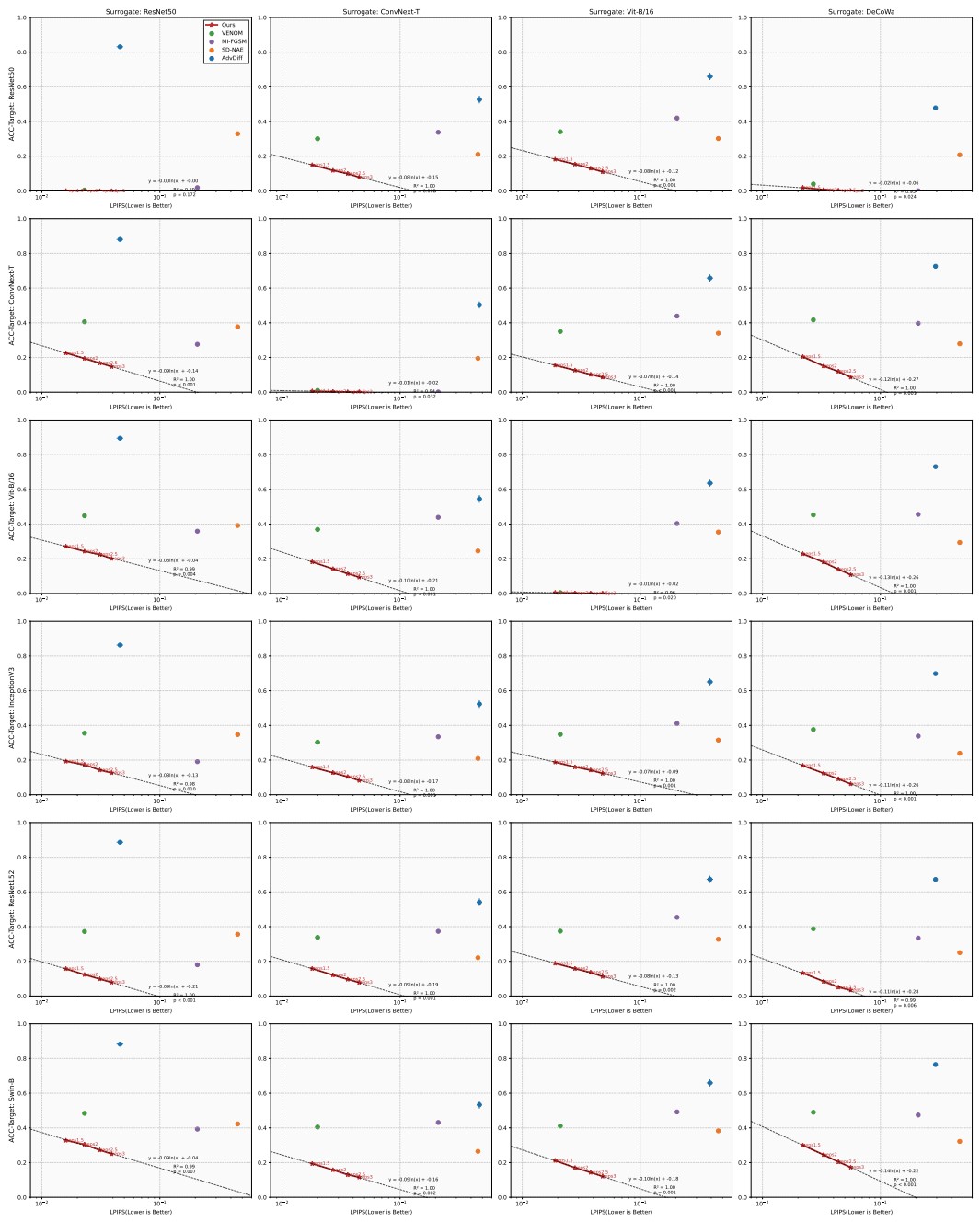

Figure 23: ACC of Original Imagenet label

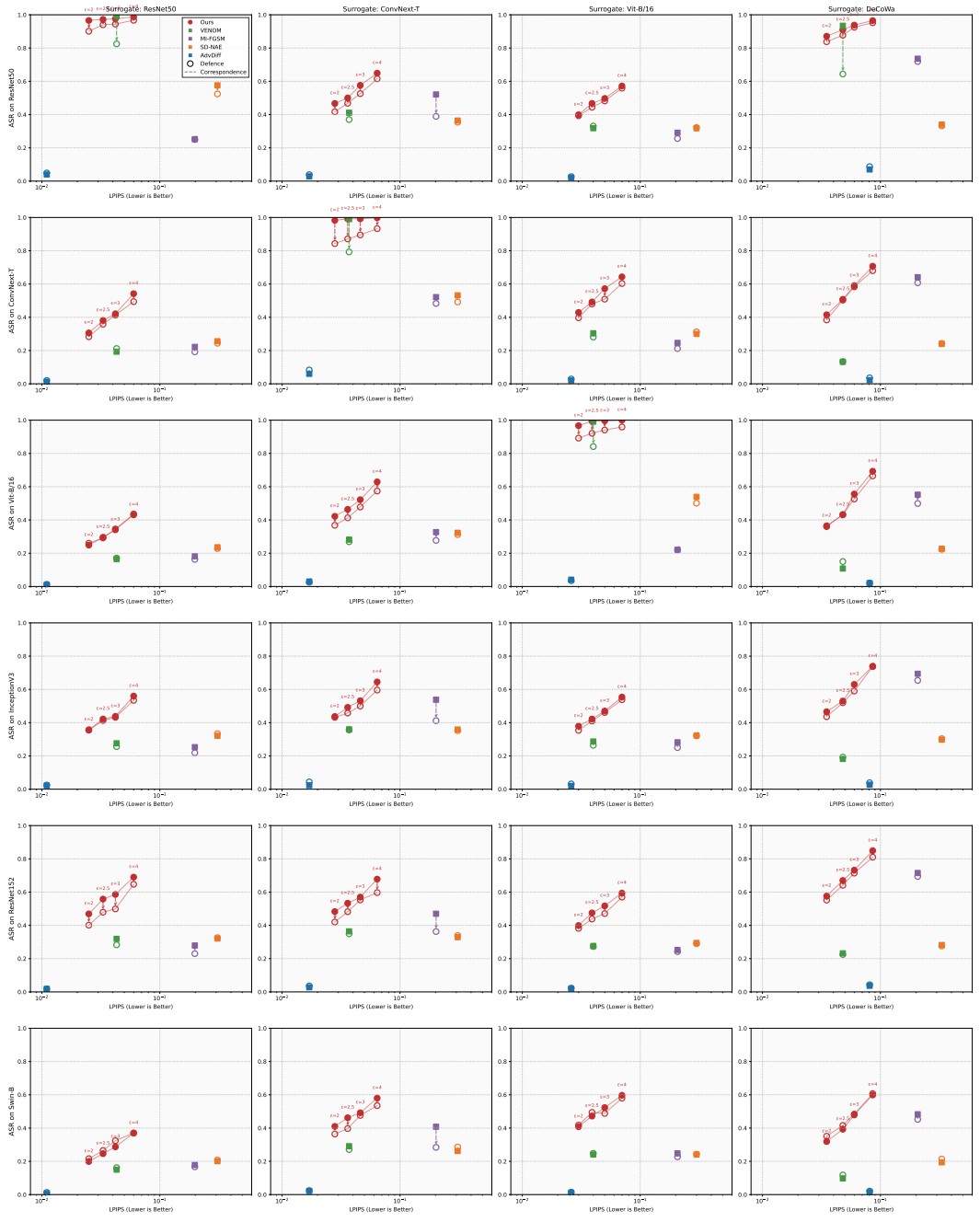

Figure 24: ASR on JPEG Defense

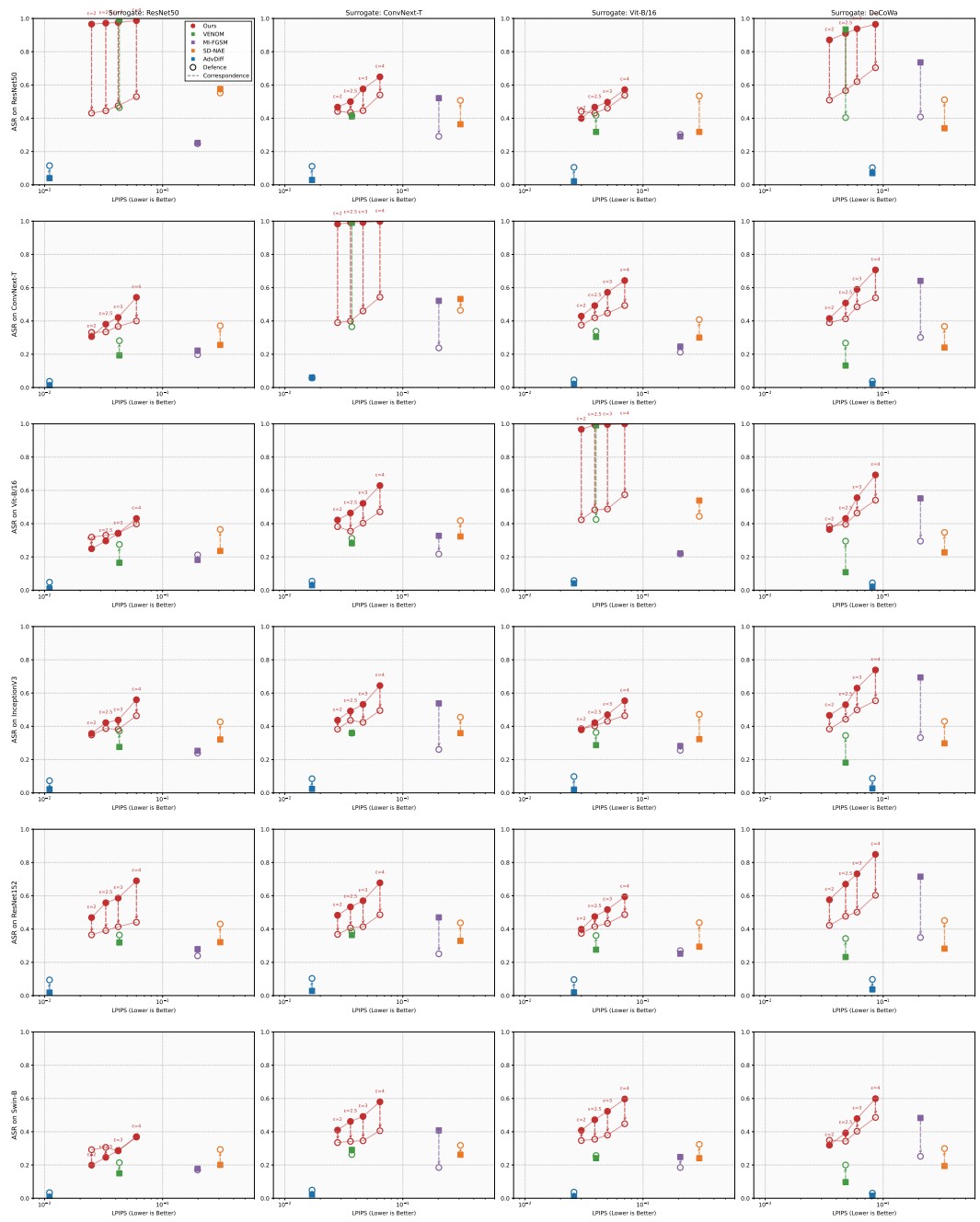

Figure 25: ASR on DiffPure Defense

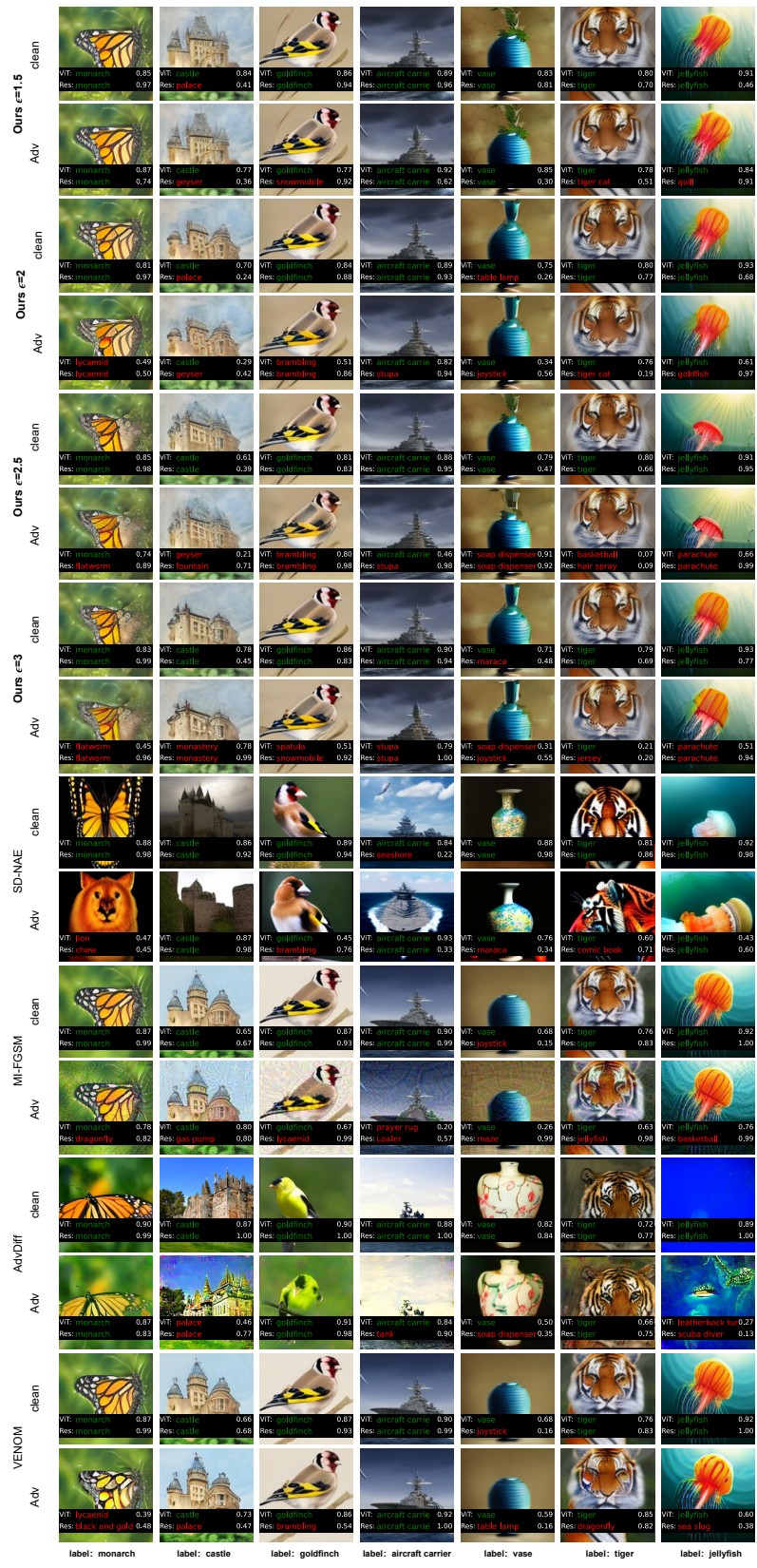

Figure 26: Visualization of 2D ImageNet-label Evasion Attacks. Surrogate model is ResNet50+DeCoWA.

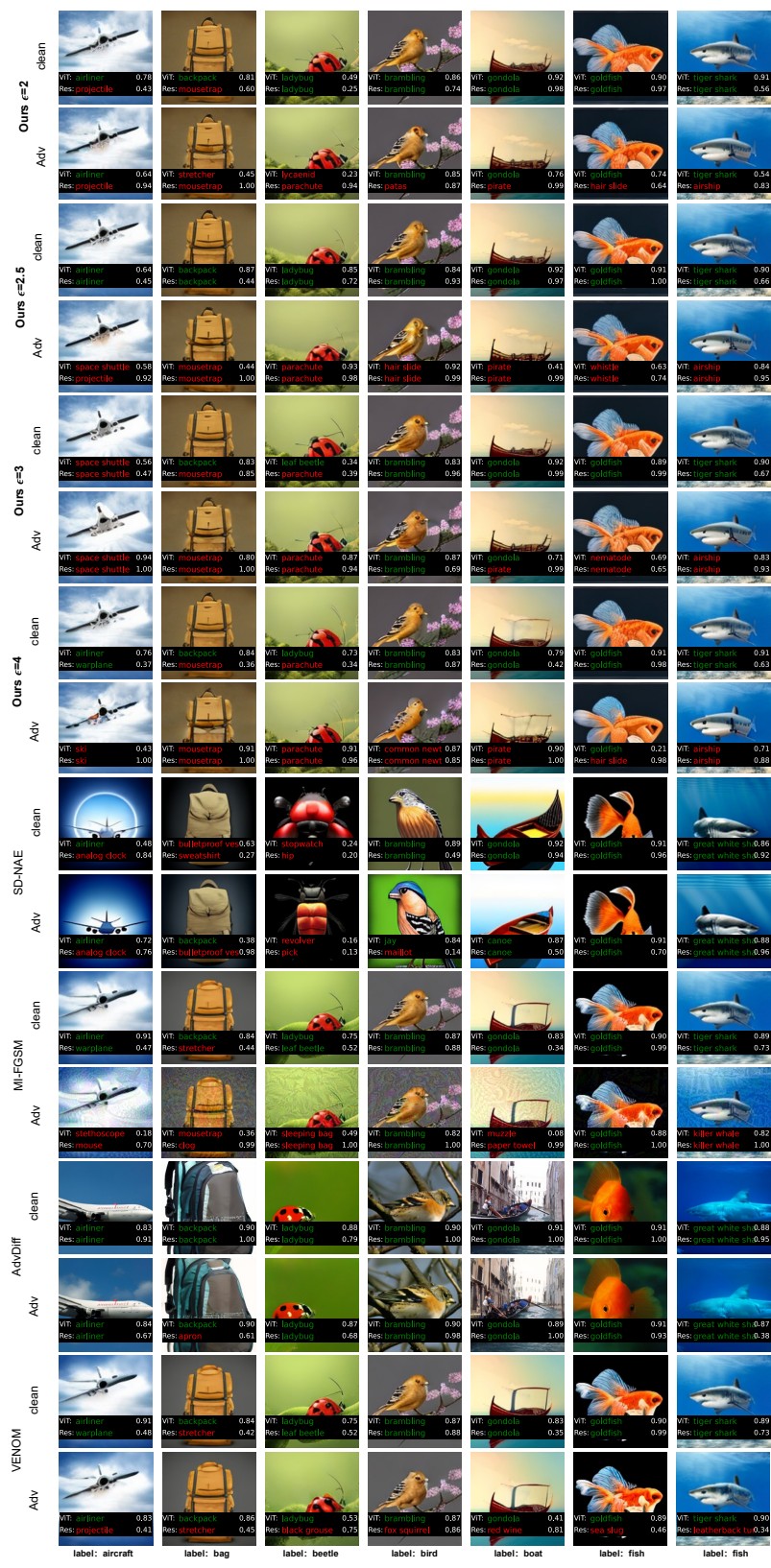

Figure 27: Visualization of 2D Abstracted-label Evasion Attacks. Surrogate model is ResNet50+DeCoWA.

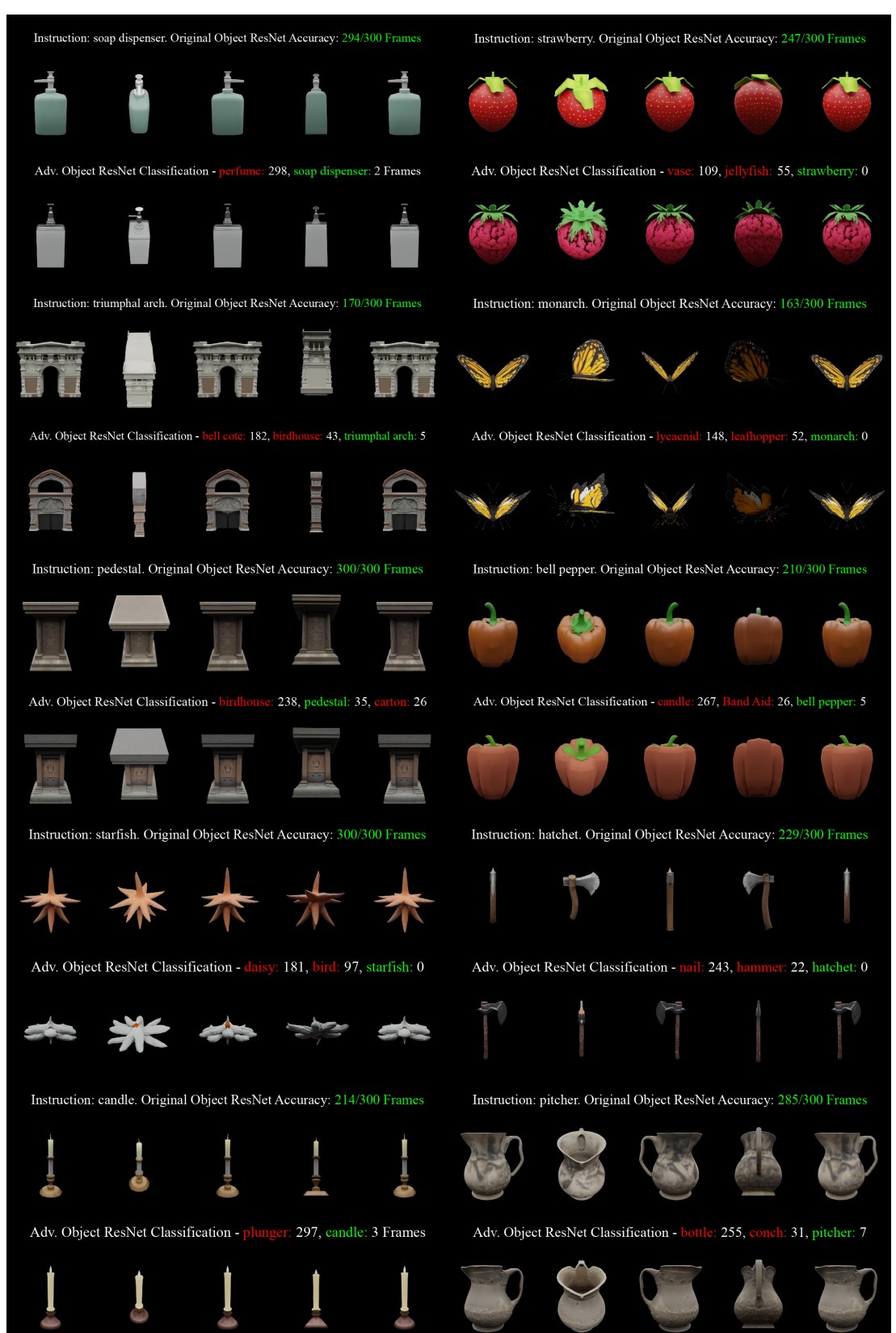

Figure 28: 3D Visual Results. Surrogate Model is ResNet50.

Table 10: Abstracted label transfer attack results.

| method | Surrogate | Metrics | | | ResNet152 | | InceptionV3 | | ResNet50 | | ViT-B/16 | | ConvNext-T | | Swin-B | |
|---|---|---|---|---|---|---|---|---|---|---|---|---|---|---|---|---|
| | | Clip$_{IQA}$ | LPIPS | MSSSIM | ASR | ACC | ASR | ACC | ASR | ACC | ASR | ACC | ASR | ACC | ASR | ACC |
| ours$_{ε=2}$ | ResNet+DeCoWA | 0.809±0.009 | 0.035±0.001 | 0.957±0.001 | 0.576±0.004 | 0.284±0.003 | 0.466±0.002 | 0.356±0.002 | 0.871±0.003 | 0.076±0.002 | 0.365±0.006 | 0.439±0.005 | 0.415±0.003 | 0.401±0.002 | 0.319±0.005 | 0.513±0.004 |
| | ResNet50 | 0.813±0.009 | 0.025±0.001 | 0.968±0.002 | 0.469±0.003 | 0.340±0.002 | 0.357±0.006 | 0.423±0.006 | 0.966±0.001 | 0.017±0.001 | 0.249±0.005 | 0.512±0.003 | 0.306±0.004 | 0.460±0.003 | 0.199±0.002 | 0.568±0.004 |
| | ViT-B/16 | 0.815±0.012 | 0.030±0.001 | 0.965±0.001 | 0.399±0.002 | 0.391±0.003 | 0.379±0.003 | 0.394±0.003 | 0.399±0.004 | 0.378±0.004 | 0.967±0.001 | 0.016±0.001 | 0.429±0.004 | 0.368±0.003 | 0.408±0.005 | 0.402±0.004 |
| | ConvNext-T | 0.814±0.009 | 0.028±0.001 | 0.967±0.001 | 0.483±0.006 | 0.340±0.004 | 0.437±0.005 | 0.375±0.004 | 0.467±0.005 | 0.349±0.003 | 0.422±0.004 | 0.388±0.003 | 0.983±0.002 | 0.010±0.001 | 0.410±0.002 | 0.420±0.002 |
| ours$_{ε=2.5}$ | ResNet+DeCoWA | 0.808±0.009 | 0.048±0.001 | 0.943±0.002 | 0.670±0.003 | 0.224±0.002 | 0.530±0.005 | 0.327±0.003 | 0.910±0.002 | 0.056±0.001 | 0.431±0.002 | 0.407±0.001 | 0.508±0.004 | 0.354±0.002 | 0.393±0.006 | 0.449±0.004 |
| | ResNet50 | 0.808±0.013 | 0.033±0.001 | 0.958±0.002 | 0.558±0.002 | 0.276±0.001 | 0.422±0.007 | 0.382±0.004 | 0.972±0.001 | 0.014±0.001 | 0.296±0.005 | 0.476±0.004 | 0.381±0.004 | 0.415±0.003 | 0.246±0.002 | 0.548±0.003 |
| | ViT-B/16 | 0.814±0.012 | 0.039±0.001 | 0.955±0.001 | 0.475±0.008 | 0.341±0.004 | 0.422±0.002 | 0.365±0.002 | 0.467±0.006 | 0.332±0.002 | 0.994±0.001 | 0.003±0.000 | 0.492±0.005 | 0.323±0.003 | 0.472±0.002 | 0.358±0.003 |
| | ConvNext-T | 0.812±0.008 | 0.036±0.001 | 0.957±0.001 | 0.533±0.005 | 0.310±0.002 | 0.492±0.008 | 0.333±0.003 | 0.500±0.005 | 0.328±0.003 | 0.464±0.004 | 0.358±0.002 | 0.492±0.005 | 0.323±0.003 | 0.462±0.004 | 0.382±0.002 |
| ours$_{ε=3}$ | ResNet+DeCoWA | 0.808±0.009 | 0.060±0.002 | 0.929±0.002 | 0.732±0.002 | 0.178±0.001 | 0.630±0.003 | 0.248±0.002 | 0.938±0.001 | 0.037±0.001 | 0.556±0.002 | 0.310±0.002 | 0.590±0.005 | 0.288±0.003 | 0.479±0.003 | 0.390±0.003 |
| | ResNet50 | 0.806±0.011 | 0.042±0.002 | 0.949±0.003 | 0.586±0.002 | 0.260±0.002 | 0.438±0.007 | 0.371±0.003 | 0.976±0.001 | 0.012±0.001 | 0.341±0.005 | 0.453±0.002 | 0.421±0.002 | 0.388±0.002 | 0.287±0.003 | 0.517±0.002 |
| | ViT-B/16 | 0.811±0.011 | 0.050±0.002 | 0.944±0.001 | 0.517±0.006 | 0.314±0.004 | 0.470±0.004 | 0.342±0.004 | 0.497±0.009 | 0.322±0.003 | 0.996±0.001 | 0.002±0.001 | 0.572±0.004 | 0.271±0.002 | 0.523±0.004 | 0.326±0.003 |
| | ConvNext-T | 0.808±0.011 | 0.046±0.002 | 0.946±0.002 | 0.570±0.007 | 0.285±0.005 | 0.532±0.006 | 0.311±0.003 | 0.576±0.008 | 0.279±0.005 | 0.522±0.004 | 0.316±0.003 | 0.992±0.001 | 0.004±0.001 | 0.492±0.006 | 0.363±0.005 |
| ours$_{ε=4}$ | ResNet+DeCoWA | 0.807±0.012 | 0.086±0.002 | 0.901±0.002 | 0.849±0.002 | 0.099±0.001 | 0.739±0.003 | 0.179±0.002 | 0.965±0.001 | 0.020±0.001 | 0.693±0.002 | 0.216±0.002 | 0.707±0.003 | 0.204±0.002 | 0.599±0.002 | 0.299±0.002 |
| | ResNet50 | 0.799±0.010 | 0.060±0.002 | 0.929±0.003 | 0.690±0.001 | 0.193±0.001 | 0.560±0.010 | 0.293±0.005 | 0.987±0.001 | 0.006±0.000 | 0.431±0.006 | 0.389±0.004 | 0.542±0.002 | 0.302±0.003 | 0.370±0.005 | 0.451±0.005 |
| | ViT-B/16 | 0.806±0.012 | 0.070±0.001 | 0.922±0.003 | 0.594±0.004 | 0.270±0.003 | 0.554±0.005 | 0.292±0.003 | 0.572±0.003 | 0.265±0.003 | 1.000±0.000 | 0.000±0.000 | 0.643±0.004 | 0.226±0.004 | 0.597±0.005 | 0.274±0.004 |
| | ConvNext-T | 0.808±0.008 | 0.064±0.002 | 0.926±0.002 | 0.678±0.003 | 0.212±0.002 | 0.645±0.005 | 0.245±0.003 | 0.649±0.004 | 0.230±0.003 | 0.629±0.003 | 0.245±0.003 | 0.998±0.001 | 0.001±0.000 | 0.580±0.006 | 0.296±0.005 |
| mifgsm | ResNet+DeCoWA | 0.474±0.013 | 0.207±0.008 | 0.870±0.006 | 0.715±0.002 | 0.220±0.001 | 0.694±0.002 | 0.226±0.002 | 0.736±0.002 | 0.194±0.002 | 0.552±0.003 | 0.343±0.003 | 0.641±0.003 | 0.274±0.003 | 0.483±0.004 | 0.418±0.003 |
| | ResNet50 | 0.551±0.014 | 0.198±0.009 | 0.885±0.005 | 0.279±0.003 | 0.574±0.002 | 0.253±0.001 | 0.596±0.002 | 0.252±0.001 | 0.553±0.003 | 0.182±0.001 | 0.647±0.002 | 0.222±0.001 | 0.625±0.004 | 0.178±0.002 | 0.684±0.002 |
| | ViT-B/16 | 0.521±0.013 | 0.207±0.008 | 0.860±0.006 | 0.252±0.004 | 0.594±0.004 | 0.282±0.004 | 0.559±0.004 | 0.291±0.002 | 0.545±0.004 | 0.221±0.002 | 0.577±0.003 | 0.246±0.002 | 0.596±0.002 | 0.248±0.002 | 0.621±0.002 |
| | ConvNext-T | 0.532±0.009 | 0.202±0.009 | 0.880±0.005 | 0.470±0.003 | 0.411±0.003 | 0.538±0.003 | 0.351±0.002 | 0.521±0.003 | 0.361±0.002 | 0.327±0.002 | 0.515±0.002 | 0.521±0.003 | 0.359±0.002 | 0.408±0.002 | 0.476±0.001 |
| venom | ResNet+DeCoWA | 0.796±0.016 | 0.048±0.005 | 0.944±0.005 | 0.232±0.004 | 0.607±0.001 | 0.182±0.002 | 0.621±0.002 | 0.934±0.002 | 0.049±0.002 | 0.108±0.002 | 0.685±0.004 | 0.132±0.002 | 0.672±0.002 | 0.097±0.001 | 0.739±0.002 |
| | ResNet50 | 0.779±0.012 | 0.043±0.004 | 0.951±0.004 | 0.319±0.004 | 0.543±0.002 | 0.276±0.004 | 0.556±0.004 | 0.991±0.001 | 0.006±0.001 | 0.165±0.003 | 0.647±0.004 | 0.193±0.003 | 0.621±0.002 | 0.150±0.001 | 0.687±0.003 |
| | ViT-B/16 | 0.780±0.016 | 0.040±0.004 | 0.958±0.004 | 0.276±0.003 | 0.572±0.002 | 0.287±0.002 | 0.538±0.003 | 0.318±0.004 | 0.514±0.003 | 0.991±0.001 | 0.006±0.001 | 0.304±0.002 | 0.537±0.001 | 0.241±0.003 | 0.612±0.002 |
| | ConvNext-T | 0.785±0.010 | 0.037±0.003 | 0.961±0.004 | 0.364±0.004 | 0.499±0.004 | 0.360±0.003 | 0.485±0.002 | 0.411±0.004 | 0.443±0.002 | 0.282±0.002 | 0.542±0.002 | 0.990±0.001 | 0.009±0.001 | 0.291±0.003 | 0.579±0.003 |
| SD-NAE | ResNet+DeCoWA | 0.782±0.009 | 0.331±0.030 | 0.568±0.034 | 0.282±0.002 | 0.618±0.001 | 0.298±0.002 | 0.603±0.003 | 0.340±0.003 | 0.573±0.003 | 0.227±0.003 | 0.667±0.002 | 0.240±0.002 | 0.676±0.002 | 0.194±0.003 | 0.725±0.003 |
| | ResNet50 | 0.771±0.014 | 0.308±0.043 | 0.599±0.050 | 0.321±0.002 | 0.577±0.002 | 0.321±0.003 | 0.582±0.003 | 0.576±0.003 | 0.350±0.001 | 0.236±0.003 | 0.645±0.001 | 0.256±0.004 | 0.653±0.003 | 0.201±0.002 | 0.708±0.002 |
| | ViT-B/16 | 0.787±0.010 | 0.300±0.023 | 0.609±0.030 | 0.294±0.002 | 0.583±0.004 | 0.323±0.004 | 0.573±0.004 | 0.318±0.004 | 0.567±0.003 | 0.539±0.003 | 0.377±0.003 | 0.300±0.003 | 0.604±0.002 | 0.241±0.002 | 0.665±0.003 |
| | ConvNext-T | 0.782±0.012 | 0.308±0.026 | 0.603±0.033 | 0.329±0.002 | 0.549±0.001 | 0.359±0.003 | 0.528±0.001 | 0.364±0.002 | 0.525±0.001 | 0.323±0.003 | 0.568±0.002 | 0.532±0.004 | 0.396±0.002 | 0.262±0.001 | 0.649±0.001 |
| adv diff | ResNet+DeCoWA | 0.629±0.007 | 0.081±0.007 | 0.934±0.007 | 0.037±0.001 | 0.936±0.002 | 0.027±0.001 | 0.924±0.001 | 0.070±0.002 | 0.889±0.003 | 0.020±0.001 | 0.949±0.001 | 0.022±0.001 | 0.948±0.002 | 0.015±0.000 | 0.962±0.001 |
| | ResNet50 | 0.621±0.007 | 0.011±0.001 | 0.992±0.001 | 0.018±0.001 | 0.955±0.001 | 0.022±0.001 | 0.931±0.001 | 0.040±0.002 | 0.917±0.003 | 0.010±0.001 | 0.955±0.002 | 0.013±0.001 | 0.951±0.001 | 0.008±0.001 | 0.968±0.001 |
| | ViT-B/16 | 0.628±0.005 | 0.026±0.008 | 0.972±0.012 | 0.019±0.001 | 0.955±0.002 | 0.020±0.001 | 0.928±0.002 | 0.021±0.001 | 0.933±0.002 | 0.041±0.002 | 0.925±0.004 | 0.020±0.001 | 0.944±0.002 | 0.011±0.001 | 0.962±0.001 |
| | ConvNext-T | 0.627±0.006 | 0.017±0.004 | 0.985±0.007 | 0.028±0.002 | 0.947±0.002 | 0.024±0.001 | 0.926±0.002 | 0.029±0.002 | 0.926±0.002 | 0.030±0.001 | 0.938±0.002 | 0.060±0.001 | 0.907±0.001 | 0.022±0.001 | 0.953±0.001 |

Table 11: Original ImageNet label transfer attack results.

| method | Surrogate | Metrics Clip$_{IQA}$ | LPIPS | MSSSIM | ResNet152 ASR | ACC | InceptionV3 ASR | ACC | Swin-B ASR | ACC | ResNet50 ASR | ACC | ConvNext-T ASR | ACC | Vit-B/16 ASR | ACC |
|---|---|---|---|---|---|---|---|---|---|---|---|---|---|---|---|---|
| ours$_{\varepsilon=1.5}$ | ConvNext-T | 0.822±0.000 | 0.018±0.000 | 0.978±0.000 | 0.576±0.001 | 0.158±0.001 | 0.550±0.003 | 0.159±0.002 | 0.542±0.003 | 0.194±0.001 | 0.571±0.002 | 0.150±0.001 | 0.981±0.001 | 0.006±0.001 | 0.514±0.002 | 0.182±0.001 |
| | ResNet+DeCoWa | 0.821±0.003 | 0.022±0.000 | 0.973±0.000 | 0.654±0.001 | 0.133±0.001 | 0.567±0.002 | 0.168±0.001 | 0.374±0.002 | 0.300±0.001 | 0.940±0.002 | 0.019±0.001 | 0.484±0.001 | 0.204±0.001 | 0.459±0.003 | 0.228±0.002 |
| | ResNet50 | 0.821±0.003 | 0.016±0.000 | 0.979±0.000 | 0.559±0.002 | 0.157±0.001 | 0.465±0.002 | 0.193±0.001 | 0.275±0.000 | 0.328±0.001 | 0.997±0.000 | 0.001±0.000 | 0.398±0.002 | 0.226±0.001 | 0.311±0.002 | 0.271±0.001 |
| | Vit-B/16 | 0.820±0.005 | 0.019±0.000 | 0.977±0.001 | 0.473±0.004 | 0.188±0.002 | 0.470±0.004 | 0.188±0.002 | 0.478±0.001 | 0.212±0.001 | 0.470±0.005 | 0.182±0.002 | 0.543±0.004 | 0.155±0.002 | 0.976±0.001 | 0.005±0.000 |
| ours$_{\varepsilon=2}$ | ConvNext-T | 0.819±0.002 | 0.027±0.000 | 0.968±0.000 | 0.674±0.001 | 0.121±0.001 | 0.641±0.003 | 0.127±0.002 | 0.630±0.002 | 0.158±0.001 | 0.660±0.001 | 0.118±0.001 | 0.990±0.000 | 0.003±0.000 | 0.619±0.002 | 0.142±0.001 |
| | ResNet+DeCoWa | 0.815±0.004 | 0.033±0.001 | 0.960±0.001 | 0.782±0.001 | 0.084±0.000 | 0.680±0.003 | 0.124±0.001 | 0.479±0.002 | 0.245±0.001 | 0.977±0.002 | 0.007±0.001 | 0.624±0.002 | 0.150±0.001 | 0.566±0.002 | 0.181±0.001 |
| | ResNet50 | 0.819±0.004 | 0.023±0.000 | 0.970±0.000 | 0.650±0.002 | 0.123±0.001 | 0.517±0.001 | 0.174±0.001 | 0.336±0.002 | 0.305±0.000 | 0.999±0.000 | 0.000±0.000 | 0.486±0.002 | 0.194±0.001 | 0.384±0.002 | 0.243±0.001 |
| | Vit-B/16 | 0.817±0.004 | 0.028±0.001 | 0.967±0.000 | 0.555±0.004 | 0.158±0.001 | 0.548±0.003 | 0.160±0.001 | 0.577±0.001 | 0.170±0.001 | 0.557±0.004 | 0.154±0.002 | 0.627±0.003 | 0.126±0.001 | 0.992±0.000 | 0.002±0.000 |
| ours$_{\varepsilon=2.5}$ | ConvNext-T | 0.817±0.003 | 0.036±0.000 | 0.958±0.001 | 0.740±0.002 | 0.096±0.001 | 0.710±0.003 | 0.104±0.001 | 0.694±0.002 | 0.130±0.001 | 0.712±0.002 | 0.100±0.001 | 0.997±0.001 | 0.001±0.000 | 0.692±0.002 | 0.114±0.001 |
| | ResNet+DeCoWa | 0.810±0.004 | 0.044±0.001 | 0.947±0.001 | 0.870±0.001 | 0.050±0.001 | 0.762±0.001 | 0.091±0.001 | 0.562±0.002 | 0.204±0.001 | 0.991±0.001 | 0.003±0.000 | 0.701±0.003 | 0.120±0.001 | 0.664±0.002 | 0.139±0.001 |
| | ResNet50 | 0.815±0.006 | 0.031±0.000 | 0.961±0.000 | 0.724±0.002 | 0.098±0.001 | 0.607±0.003 | 0.142±0.001 | 0.399±0.002 | 0.272±0.001 | 1.000±0.000 | 0.000±0.000 | 0.555±0.001 | 0.168±0.001 | 0.434±0.001 | 0.225±0.001 |
| | Vit-B/16 | 0.815±0.004 | 0.038±0.000 | 0.956±0.000 | 0.619±0.004 | 0.136±0.002 | 0.599±0.003 | 0.142±0.001 | 0.644±0.001 | 0.143±0.001 | 0.627±0.004 | 0.130±0.002 | 0.697±0.001 | 0.102±0.001 | 0.996±0.000 | 0.001±0.000 |
| ours$_{\varepsilon=3}$ | ConvNext-T | 0.814±0.003 | 0.045±0.000 | 0.947±0.001 | 0.790±0.001 | 0.078±0.001 | 0.767±0.002 | 0.082±0.001 | 0.725±0.001 | 0.116±0.001 | 0.772±0.001 | 0.079±0.001 | 0.998±0.000 | 0.001±0.000 | 0.745±0.002 | 0.094±0.001 |
| | ResNet+DeCoWa | 0.807±0.004 | 0.056±0.000 | 0.934±0.001 | 0.912±0.001 | 0.034±0.001 | 0.841±0.001 | 0.062±0.001 | 0.636±0.002 | 0.173±0.001 | 0.999±0.000 | 0.000±0.000 | 0.787±0.001 | 0.086±0.001 | 0.740±0.003 | 0.108±0.001 |
| | ResNet50 | 0.812±0.005 | 0.039±0.001 | 0.952±0.001 | 0.775±0.003 | 0.079±0.001 | 0.649±0.002 | 0.126±0.001 | 0.453±0.003 | 0.250±0.001 | 1.000±0.000 | 0.000±0.000 | 0.608±0.001 | 0.147±0.001 | 0.491±0.002 | 0.201±0.001 |
| | Vit-B/16 | 0.812±0.003 | 0.048±0.001 | 0.945±0.001 | 0.687±0.003 | 0.112±0.001 | 0.654±0.002 | 0.123±0.001 | 0.701±0.002 | 0.121±0.001 | 0.682±0.001 | 0.110±0.001 | 0.745±0.002 | 0.086±0.001 | 1.000±0.000 | 0.000±0.000 |
| MI-FGSM | ConvNext-T | 0.543±0.004 | 0.211±0.001 | 0.877±0.000 | 0.362±0.002 | 0.373±0.001 | 0.399±0.002 | 0.334±0.001 | 0.281±0.002 | 0.431±0.001 | 0.408±0.001 | 0.338±0.001 | 0.999±0.000 | 0.001±0.000 | 0.243±0.001 | 0.439±0.001 |
| | ResNet+DeCoWa | 0.548±0.003 | 0.209±0.002 | 0.880±0.001 | 0.430±0.003 | 0.334±0.001 | 0.391±0.002 | 0.338±0.002 | 0.201±0.001 | 0.474±0.001 | 0.999±0.001 | 0.001±0.000 | 0.273±0.001 | 0.397±0.000 | 0.199±0.001 | 0.456±0.001 |
| | ResNet50 | 0.535±0.003 | 0.208±0.002 | 0.869±0.001 | 0.697±0.001 | 0.180±0.000 | 0.660±0.001 | 0.191±0.001 | 0.338±0.001 | 0.393±0.001 | 0.967±0.001 | 0.019±0.000 | 0.498±0.001 | 0.276±0.001 | 0.367±0.002 | 0.359±0.001 |
| | Vit-B/16 | 0.539±0.003 | 0.205±0.001 | 0.856±0.001 | 0.221±0.002 | 0.454±0.001 | 0.251±0.002 | 0.411±0.001 | 0.192±0.002 | 0.492±0.001 | 0.249±0.001 | 0.420±0.001 | 0.211±0.002 | 0.439±0.002 | 0.255±0.001 | 0.403±0.001 |
| VENOM | ConvNext-T | 0.796±0.002 | 0.020±0.001 | 0.978±0.001 | 0.350±0.002 | 0.338±0.002 | 0.383±0.001 | 0.303±0.001 | 0.278±0.001 | 0.405±0.002 | 0.387±0.002 | 0.301±0.001 | 0.982±0.001 | 0.010±0.001 | 0.294±0.002 | 0.369±0.001 |
| | ResNet+DeCoWa | 0.805±0.002 | 0.027±0.001 | 0.968±0.002 | 0.257±0.001 | 0.388±0.002 | 0.247±0.002 | 0.376±0.001 | 0.136±0.001 | 0.490±0.002 | 0.917±0.001 | 0.040±0.001 | 0.186±0.001 | 0.417±0.001 | 0.144±0.001 | 0.453±0.001 |
| | ResNet50 | 0.795±0.003 | 0.023±0.000 | 0.972±0.000 | 0.288±0.002 | 0.372±0.002 | 0.288±0.002 | 0.355±0.001 | 0.147±0.001 | 0.484±0.001 | 0.991±0.001 | 0.005±0.000 | 0.201±0.001 | 0.406±0.001 | 0.151±0.001 | 0.448±0.000 |
| | Vit-B/16 | 0.796±0.002 | 0.021±0.001 | 0.977±0.001 | 0.274±0.002 | 0.374±0.001 | 0.288±0.002 | 0.348±0.001 | 0.267±0.001 | 0.411±0.002 | 0.310±0.001 | 0.341±0.001 | 0.305±0.002 | 0.350±0.001 | 0.990±0.001 | 0.005±0.000 |
| SD-NAE | ConvNext-T | 0.848±0.009 | 0.458±0.022 | 0.432±0.018 | 0.675±0.001 | 0.221±0.001 | 0.690±0.002 | 0.209±0.001 | 0.626±0.002 | 0.265±0.001 | 0.689±0.002 | 0.211±0.001 | 0.710±0.002 | 0.195±0.001 | 0.649±0.002 | 0.245±0.001 |
| | ResNet+DeCoWa | 0.845±0.013 | 0.470±0.018 | 0.421±0.016 | 0.636±0.001 | 0.250±0.001 | 0.646±0.001 | 0.239±0.001 | 0.549±0.001 | 0.322±0.001 | 0.691±0.002 | 0.208±0.002 | 0.592±0.001 | 0.279±0.001 | 0.587±0.001 | 0.294±0.001 |
| | ResNet50 | 0.841±0.011 | 0.457±0.020 | 0.433±0.017 | 0.490±0.002 | 0.356±0.001 | 0.502±0.001 | 0.347±0.001 | 0.420±0.001 | 0.423±0.001 | 0.518±0.001 | 0.330±0.001 | 0.458±0.002 | 0.377±0.001 | 0.454±0.001 | 0.392±0.001 |
| | Vit-B/16 | 0.844±0.009 | 0.459±0.016 | 0.441±0.014 | 0.530±0.002 | 0.327±0.001 | 0.539±0.001 | 0.315±0.001 | 0.467±0.001 | 0.383±0.001 | 0.556±0.002 | 0.302±0.001 | 0.505±0.001 | 0.340±0.001 | 0.501±0.001 | 0.354±0.001 |
| AdvDiff | ConvNext-T | 0.636±0.006 | 0.471±0.025 | 0.312±0.011 | 0.440±0.022 | 0.541±0.022 | 0.448±0.022 | 0.523±0.022 | 0.433±0.023 | 0.533±0.022 | 0.446±0.022 | 0.527±0.022 | 0.465±0.022 | 0.503±0.020 | 0.430±0.023 | 0.545±0.022 |
| | ResNet+DeCoWa | 0.597±0.004 | 0.293±0.003 | 0.761±0.003 | 0.289±0.001 | 0.672±0.001 | 0.247±0.001 | 0.698±0.001 | 0.180±0.001 | 0.765±0.001 | 0.487±0.001 | 0.479±0.001 | 0.222±0.001 | 0.726±0.001 | 0.225±0.001 | 0.731±0.001 |
| | ResNet50 | 0.634±0.004 | 0.046±0.003 | 0.939±0.001 | 0.046±0.002 | 0.887±0.001 | 0.050±0.001 | 0.863±0.001 | 0.035±0.001 | 0.883±0.002 | 0.091±0.001 | 0.831±0.001 | 0.041±0.002 | 0.881±0.001 | 0.035±0.001 | 0.895±0.001 |
| | Vit-B/16 | 0.638±0.004 | 0.390±0.025 | 0.430±0.010 | 0.295±0.021 | 0.673±0.021 | 0.304±0.021 | 0.651±0.020 | 0.295±0.022 | 0.659±0.021 | 0.300±0.021 | 0.660±0.021 | 0.298±0.021 | 0.658±0.021 | 0.327±0.021 | 0.636±0.020 |

