# OpenReview forum: "Exploring Semantic-constrained Adversarial  Example with Instruction Uncertainty Reduction"
_NeurIPS.cc/2025/Conference — NeurIPS 2025 poster_

### Official Review · Reviewer_vVyk · 2025-06-25

**Clarity:** 3
**Significance:** 3
**Originality:** 2
**Rating:** 4
**Confidence:** 5

**Summary:**

The paper "Exploring Semantic-constrained Adversarial Example with Instruction Uncertainty Reduction" introduces a novel framework, InSUR, for generating semantic-constrained adversarial examples (SemanticAEs) directly from natural language instructions. It addresses the challenges of semantic uncertainty in instructions, categorized as referring diversity, descriptive incompleteness, and boundary ambiguity. The InSUR framework proposes three key contributions: (1) a residual-driven attacking direction stabilization using the ResAdv-DDIM sampler to enhance adversarial optimization stability in diffusion models, (2) context-encoded attacking scenario constraints to improve scenario-adapted 2D and 3D SemanticAE generation by integrating external knowledge, and (3) a semantic-abstracted attacking evaluation enhancement based on label taxonomy to provide robust evaluation metrics. The paper demonstrates improved transfer attack performance for 2D SemanticAEs and achieves the first reference-free generation of 3D SemanticAEs using language-guided 3D generation models. Extensive experiments validate the effectiveness of the proposed methods across ImageNet-based tasks and novel abstracted label evasion tasks.

**Questions:**

1、Please clarify the residual-driven attacking direction stabilization mechanism. You should provide a detailed explanation of the residual term’s role in stabilizing adversarial direction. The ResAdv-DDIM sampler is a core contribution, proposed to stabilize adversarial optimization under instruction uncertainty by introducing a residual-driven attacking direction. The formulation (Eq. 2–4) adapts the DDIM process with a residual term and momentum optimization, but the technical details are dense, and the rationale for design choices (e.g., choice of momentum parameters) is not fully explained.

2、This paper should include a quantitative analysis comparing ResAdv-DDIM to standard DDIM in terms of convergence stability (e.g., loss curves, gradient variance) or attack success rate (ASR) under varying instruction diversity.

3、The paper emphasizes addressing instruction uncertainty (referring diversity, descriptive incompleteness, boundary ambiguity) but does not provide details on the natural language instructions used in experiments. The lack of information about instruction diversity (e.g., length, complexity, semantic variation) or their construction (e.g., manual, dataset-derived, or programmatic) makes it difficult to assess the robustness of the InSUR framework across varied inputs.

4、Please validate or discuss real-world applicability of 3D semanticAEs.

**Ethical Concerns:**

["NO or VERY MINOR ethics concerns only"]

**Final Justification:**

Thank you for your detailed response. I have reviewed your revisions. My original concerns have been addressed. Therefore, I am satisfied with the revisions and will maintain my original recommendation.

**Limitations:**

Yes

**Paper Formatting Concerns:**

Nan

**Quality:**

3

**Strengths And Weaknesses:**

Strengths:

1、The paper presents a well-designed multi-dimensional framework (InSUR) that systematically addresses three forms of semantic uncertainty (referring diversity, descriptive incompleteness, and boundary ambiguity). The proposed techniques, such as the ResAdv-DDIM sampler, context-encoded scenario constraints, and semantic-abstracted evaluation, are theoretically grounded and build on established methods.

2、The experimental evaluation is robust, testing the framework on ImageNet 1000-class label evasion and a novel abstracted label evasion task. The use of multiple surrogate models and transfer attack methods ensures a thorough assessment. The reported results, such as a 1.19× average attack success rate (ASR), demonstrate the framework’s effectiveness.

3、Figures 2–6 and 9–11 effectively illustrate the framework, optimization challenges, and generated SemanticAEs. The visualizations of 2D and 3D samples (Figures 10 and 11) are compelling, showcasing the naturalness and adversarial effectiveness of the generated examples.

Weaknesses:

1、While the paper mentions potential real-world applications, it lacks empirical validation in physical-world scenarios (e.g., testing 3D SemanticAEs in simulated environments). This limits the immediate practical significance of the results.

2、The 3D generation relies on the Trellis framework [41], which may reduce the perceived originality of the 3D pipeline, as it leverages an existing tool rather than proposing a fully new 3D generation method.

3、Some components, such as the use of guidance masking for 2D generation, are less novel and resemble techniques in prior diffusion-based adversarial works (e.g., AdvDiff [16]). While the integration is creative, these elements may be seen as incremental improvements.

---

> ### Author Rebuttal · Authors · 2025-07-31
>
> We are truly grateful for your valuable suggestions. Below are our responses to the questions raised.
>
> **Response to Weekness 1 & Q4: Real-world applicability of 3D SemanticAEs**
>
> In our supplementary material, we include a simulation video showing adversarial attacks under continuous camera rotation around the 3D object, demonstrating strong multi-view robustness. Following the reviewer's suggestion, we conducted a Blender-based physical simulation using sunlight illumination. The tested model is the forklift model in the supplementary video. Under the viewpoints, material and lighting settings different from the training stage, the ResNet50 classification accuracy on clean samples was 59.41%, while on adversarial samples it dropped to 20.0%. Crucially, the two models are visually indistinguishable, indicating that the generated 3D adversarial objects maintain natural appearance while achieving strong attack performance.
> Due to NeurIPS policy restrictions, we are unable to include visualizations, but we will release the simulation results in the camera-ready version.
>
> Additionally, a concurrent work at ICCV 2025 [1] (preprinted after our initial submission) demonstrates that adversarial examples in 3D Gaussian representations can effectively enable physical-world attacks. Our semantic-level control advances this direction. We discuss the relationship between that work and ours below. LLM-related application is discussed in our response to Reviewer t5zq.
>
> ---
>
> **Response to Weakness 2: Use of Trellis may reduce perceived originality**
>
> We clarify that our framework is **not limited to Trellis** and can be adapted to other diffusion-based 3D generation models—similar to how Hugging Face Diffusers supports multiple architectures. Trellis serves as one implementation backend, and our design choices prioritize compatibility with existing open-source tools. From the perspective of AI safety governance, we argue that effectively leveraging widely adopted, open models and frameworks is not only realistic but essential. It addresses the economic and scalability challenges in adversarial robustness research, enabling broader access and faster iteration.
>
> Moreover, to the best of our knowledge, no prior work had applied 3D Gaussian Splatting to adversarial example generation. We note that a concurrent ICCV 2025 paper [1] also adopts 3D Gaussian rendering and reports improved physical-world attack performance. While we omit extensive discussion of 3D reconstruction mechanics due to space constraints in the main paper, we believe that since 3D Gaussian Splatting projects 2D gradient feedback onto spherical harmonic bases, it effectively reduces optimization uncertainty and enables efficient inverse optimization of adversarial patterns.
> By integrating this with ResAdv-DDIM, we establish a *multidimensional uncertainty reduction framework*, stabilizing both the semantic (via residual approximation) and geometric (via 3D Gaussian representation) optimization. We believe this paradigm provides valuable guidance for scaling red-teaming frameworks to complex tasks.
>
> ---
>
> **Response to Weakness 3: Perceived incremental nature of some components**
>
> The core innovation of this paper lies in constructing an integrated robust attack framework through Instruction Uncertainty Reduction. Regarding component-level contributions:
>
> 1. By combining residual approximation and adaptive optimization, we establish a new technical solution for transferability enhancement, while unlocking the potential of multi-step diffusion models in improving adversarial transferability and robustness.
>
> 2. Motivated by scenario adaptability, we design an effective knowledge injection mechanism for ResAdv-DDIM from both internal sampling model and external guidance perspectives, supporting cross-task applicability.
>
> 3. To address ambiguity in evaluation, we propose an exemplar-based metric that alleviates the metric robustness problem posed by Clip-VQA-style assessment in the SemanticAE research. Detailed discussions are provided in Appendix B.3 and Appendix D.4.
>
> Regarding the use of masked guidance, we have discussed its role in Appendix A.1. Masked guidance has been primarily used in diffusion-based image editing to constrain the edited regions, i.e., constraining $x$, with limited exploration on the guidance model $\epsilon$. As stated in Appendix A.1, *in our 2D generation task, we take a further step by modeling the interaction between conditional and unconditional guidance, achieving the new function of re-distributing the spatial strength of the semantic constraint.* This approach has not yet been explored in the adversarial attack domain.
>
>
>
> ---
>
> **Response to Question 1**
>
> Thanks for the suggestion. We will provide a more detailed explanation in the camera-ready version. Specifically, at diffusion step $t$, the adversarial update $\Delta x_t$ is computed as:
> $$
> x_t \to \hat{x}_0 \to \mathcal{M}(\hat{x}_0) \to \Delta \hat{x}_0 \to \Delta x_t,
> $$
> where $\hat{x}_0 \sim \mathrm{ResApprox.}[x_t]$. In principle, the higher the number of internal sampling steps in the $\mathrm{ResApprox.}$ process, the more accurate the estimation of $\hat{x}_0$, leading to lower variance $\mathrm{Var}[\Delta x_t]$. This results in more stable optimization, as reflected by fewer abrupt drops in attack strength between consecutive denoising steps. We verify this phenomenon in the following supplementary experiments.
>
> Furthermore, our contribution goes beyond improving $\hat{x}_0$ estimation: by enabling stronger adversarial optimization in early denoising steps via ResAdv-DDIM, we enhance the transferability of the final adversarial sample. Since early steps determine the global structure of the generated image, focusing on attack optimization in early steps leads to more structural and semantically stable features, which are more likely to transfer across models.
>
> We do not modify the momentum mechanism used in prior adversarial optimization methods. To isolate the effect of our design, we keep the momentum parameter identical to the baseline (VENOM). A brief review of momentum-based transfer attacks and their mechanisms is provided in Appendix A.3, and detailed implementation is described in Appendix B.1. The choice of parameter $\xi$ is discussed in our response to Reviewer rjVJ’s Question 2.
>
> ---
>
> **Response to Question 2**
>
> Thanks for the suggestion. We analyze convergence stability by measuring the mean and fluctuation of the estimated confidence of successful attack across different denoising stages of the diffusion model. Due to NeurIPS policy restrictions, we are unable to include visual plots. The quantitative results are presented below. When $k=0$, the sampler degrades to DDIM.
>
> ---
>
> **Response to Question 3**
>
> Our language instructions are formally defined in the evaluation methodology section (Section 3.4, Eq. 10). The specific label configurations are listed in the Appendix. Regarding instruction diversity, we conduct experiments under different levels of *referring diversity* to evaluate robustness. The results are shown below.
>
> ---
> **Supplementary Experiment**
>
> We evaluate the impact of different *referencing diversity* levels and varying *residual approximation* steps $k$. The surrogate model is ResNet50, and the diffusion model is `stabilityai/stable-diffusion-2-1-base`. We define four prompt sets with decreasing referencing diversity:
>
> - P0: *a dog on the floor*
> - P1: *a dog sitting on the floor*
> - P2: *a brown Labrador sitting on the wooden floor, facing and gazing us directly*
>
> The experiment is conducted 10 trials for each setting with random seeds from 0 to 9.
> To measure optimization stability, we define **Fluctuations** as the cumulative drop in attack success rate (ASR) during the denoising process $t = [T, T-1, ..., 0]$:
> $$
> \text{Fluctuations} = \sum \|\mathrm{clip}_{\min=0}(ASR[t] - ASR[t-1])\|^2
> $$
> Additionally, we compute the **Estimated ASR**—the predicted success rate after removing adversarial optimization in later steps—as a measure of the effectiveness of the attack at step $t$.
>
> The experimental results are summarized in the table below.
>
>
> |Prompt| k| Fluctuations $\downarrow, t\in [750, 500) $ | Estimated ASR$\uparrow, t\in [750, 500)$ | Fluctuations $\downarrow, t\in [500, 250)$|Estimated ASR$\uparrow, t\in [500, 250)$ | Fluctuations $\downarrow,t\in [250, 0]$ | Estimated ASR$\uparrow,t\in [250, 0]$ |
> |-|-|-|-|-|-|-|-|
> |P0|0|0.1843|$\underline{0.611}$|0.1473|0.674|0.1451|0.817|
> |P0|1|0.2036|0.538|0.0843|0.560|0.0805|0.795|
> |P0|2|**0.1375**|0.560|**0.0322**|$\underline{0.727}$|$\underline{0.0080}$|**0.934**|
> |P0|3|$\underline{0.1421}$|**0.636**|$\underline{0.0578}$|**0.767**|**0.0042**|$\underline{0.877}$|
> |||||||||
> |P1|0|0.2110|0.561|0.0722|0.617|0.0215|0.833|
> |P1|1|0.2074|0.491|0.1225|0.524|0.0559|0.781|
> |P1|2|$\underline{0.1214}$|**0.637**|**0.0088**|$\underline{0.806}$|$\underline{0.0202}$|$\underline{0.894}$|
> |P1|3|**0.1017**|$\underline{0.635}$|$\underline{0.0170}$|**0.865**|**0.0005**|**0.940**|
> |||||||||
> |P2|0|0.1025|0.369|0.0570|0.369|0.0256|0.602|
> |P2|1|0.1687|0.355|0.0626|0.365|$\underline{0.0128}$|0.672|
> |P2|2|$\underline{0.0660}$|$\underline{0.446}$|$\underline{0.0206}$|$\underline{0.567}$|0.0241|$\underline{0.853}$|
> |P2|3|**0.0279**|**0.627**|**0.0122**|**0.814**|**0.0070**|**0.930**|
>
> Overall, as $k$ increases, fluctuations decrease and Estimated ASR improves across denoising stages. Prompts with higher specificity induce smaller fluctuations, while more ambiguous prompts lead to larger fluctuations, especially when $k=0$. Due to stronger semantic constraints, more precise prompts may result in lower average ASR. We will incorporate these analysis, along with visualization results, into the main text in the camera-ready version.
>
>
> [1] Lou T, Jia X, Liang S, et al. 3D Gaussian Splatting Driven Multi-View Robust Physical Adversarial Camouflage Generation[J]. arXiv preprint arXiv:2507.01367, 2025.

---

> > ### Comment · Reviewer_vVyk · 2025-08-05
> >
> > Thank you for your detailed response. I have reviewed your revisions. My original concerns have been addressed. Therefore, I am satisfied with the revisions and will maintain my original recommendation.

---

> ### Author Response · Authors · 2025-08-04
>
> Dear Reviewer,
>
> We sincerely appreciate the time and effort you have dedicated to evaluating our work. We have carefully addressed all the comments and feedback provided in the review, and the corresponding responses are detailed in the rebuttal section above. We hope that our clarifications and revisions have adequately resolved the concerns raised.
>
> Given the approaching deadlines for the author-reviewer discussion, we would greatly appreciate your prompt feedback on any remaining questions or concerns. Timely communication will ensure we can effectively address all issues and improve the quality of our work.
>
> Thank you once again for your valuable insights and guidance!
>
> Best regards,
> Submission 2210 Authors.

---

### Official Review · Reviewer_FdBW · 2025-07-01

**Clarity:** 2
**Significance:** 3
**Originality:** 3
**Rating:** 4
**Confidence:** 3

**Summary:**

This paper mainly try to addresses the problem of generating semantic-constrained adversarial examples (SemanticAEs) directly from natural language instructions, a challenging task that moves beyond traditional perturbation-based attacks. The authors identify a core challenge: "instruction uncertainty," which they thoughtfully categorize into three key issues. To tackle these issues, the paper proposes a multi-dimensional framework named Instruction Uncertainty Reduction (InSUR). The authors conduct extensive experiments on 2D and 3D tasks, demonstrating that InSUR significantly outperforms existing state-of-the-art methods in generating transferable and effective SemanticAEs.

**Questions:**

See weakness

**Ethical Concerns:**

["NO or VERY MINOR ethics concerns only"]

**Limitations:**

The paper does not discuss the limitations of the proposed methods within the main text.

**Quality:**

3

**Strengths And Weaknesses:**

# Strengths

1. The proposed InSUR framework is technically sound and novel. The ResAdv-DDIM sampler is a clever solution to the problem of unstable gradients in multi-step generative processes.

2. Integrating a language-guided 3D generator with an adversarial objective to produce reference-free 3D objects that can fool 2D classifiers is a major step forward for the field and demonstrates the flexibility of the proposed framework.

3. The authors compare against a strong set of recent baselines (AdvDiff, SD-NAE, VENOM) across multiple surrogate models (ResNet, ViT, ConvNeXt), demonstrating robust transferability.

# Weaknesses

1. Regarding the ResAdv-DDIM sampler (Section 3.2): Could you provide more intuition on the choice of $k$ (the number of coarse prediction steps)? Is there a principle for selecting the optimal $k$, or is it purely an empirical hyperparameter?

2. Regarding 3D generation (Table 2): The results are extremely impressive, with ASR jumping to $92.2\%$. The non-adversarial baseline already has a classification accuracy of $21.5\%$ on the target model. Does this suggest that classifiers are inherently less robust to rendered 3D objects, making them an easier target for adversarial attacks compared to real 2D images?

3. Regarding the relationship between referring diversity and descriptive incompleteness. The paper neatly separates these concepts. However, could there be an interplay? For example, if an instruction is incomplete ("image of a car"), does this increase the diversity of valid references, making the optimization problem harder? How does your framework handle this potential interaction?

---

> ### Author Rebuttal · Authors · 2025-07-31
>
> We sincerely appreciate your recognition of our work's technical rigor and innovation. Below are our responses to the question raised.
>
> **Q1:** Regarding the ResAdv-DDIM sampler (Section 3.2): Could you provide more intuition on the choice of $k$ (the number of coarse prediction steps)? Is there a principle for selecting the optimal $k$, or is it purely an empirical hyperparameter?**
>
> **R1:** In principle, within a certain range, a higher $k$ leads to stronger adversarial optimization in the early denoising steps of diffusion, resulting in higher attack transferability—but at the cost of increased computational time. We provide a detailed ablation study analyzing this trade-off in *Table 3* of the main paper (partial results shown below). In the original table, *Iter$_\mathrm{max}$* corresponds to the maximum number of coarse prediction steps. The specific scheduling strategy for $k$, i.e., how many prediction steps to use at each denoising step based on remaining timesteps, is described in *Appendix B.1 (in the supplementary material)*, with $K$ being the maximum allowed prediction step.
>
> | $K$  | Acc. ($\downarrow$) | ASR ($\uparrow$) | ClipQ ($\uparrow$) | MSSSIM ($\uparrow$) | LPIPS ($\downarrow$) | Time ($\downarrow$) |
> |------|---------------------|------------------|--------------------|---------------------|-----------------------|---------------------|
> | 0    | 43.10%              | 43.30%           | 0.794              | 0.941               | 0.055                 | 4.53±0.19           |
> | 1    | 33.50%              | 54.70%           | 0.812              | 0.942               | 0.049                 | 6.87±1.48           |
> | 2    | 31.30%              | 56.60%           | 0.816              | 0.942               | 0.049                 | 7.44±1.92           |
> | 3    | 29.20%              | 57.50%           | 0.807              | 0.943               | 0.048                 | 7.75±2.37           |
> | 4    | 27.70%              | 60.10%           | 0.813              | 0.944               | 0.047                 | 7.87±2.31           |
>
> Further mechanistic analysis on the impact of $k$ on optimization stability is provided in our response to Reviewer vVyk. We believe that the choice of $K$ should be made based on the specific application scenario, balancing effectiveness and efficiency. In practice, $K$ can be selected adaptively by integrating with AutoML frameworks, using example instructions $\mathrm{Text}$ and stability metrics to determine an appropriate $K$ for the deployment environment.
>
> ---
>
> **Q2:** Regarding 3D generation (Table 2): The results are extremely impressive, with ASR jumping to 92.2. The non-adversarial baseline already has a classification accuracy of 21.5 on the target model. Does this suggest that classifiers are inherently less robust to rendered 3D objects, making them an easier target for adversarial attacks compared to real 2D images?
>
> **R2:** We believe this is due to the nature of the task. Nevertheless, our results sufficiently validate the effectiveness of the proposed method. Specifically,
> 1. Although the average clean accuracy across the dataset is low, for certain labels, the generated 3D objects achieve high clean accuracy under multi-view sampling. On these labels, our method still successfully generates effective 3D adversarial examples (visualized in Figure 11 of the main paper and the supplementary videos).
> 2. Our $ASR_\mathrm{relative}$ metric is designed precisely to address this issue: it excludes samples where the clean version is already misclassified, eliminating the influence of inherently hard-to-recognize objects. $ASR_\mathrm{relative}$ on clean samples could be regarded as 0%.
> 3.  Ablation studies show that ResAdv-DDIM significantly improves attack performance, indicating this is a non-trivial problem.
>
>
> ---
>
> **Q3:** Regarding the relationship between referring diversity and descriptive incompleteness. The paper neatly separates these concepts. However, could there be an interplay? For example, if an instruction is incomplete ("image of a car"), does this increase the diversity of valid references, making the optimization problem harder? How does your framework handle this potential interaction?
>
> **R3:** We agree with the reviewer that these concepts are interrelated. As the reviewer points out, we view *referring diversity* as having a *direct* impact on optimization, while *descriptive incompleteness* has an *indirect* effect. From a design perspective, we believe it is reasonable to first develop optimization methods targeting referring diversity, and then define contextural knowledge embedding methods that account for issues arising from descriptive incompleteness. Specifically, referring diversity characterizes the *optimization challenge* faced by transfer attack algorithms, while descriptive incompleteness relates to how the *optimization objective* is defined. InSUR provides a collaborative design solution for these two issues. We will add further clarification on this methodological design principle at the beginning of the methodology section to better address this point.

---

> ### Author Response · Authors · 2025-08-04
>
> Dear Reviewer,
>
> We sincerely appreciate the time and effort you have dedicated to evaluating our work. We have carefully addressed all the comments and feedback provided in the review, and the corresponding responses are detailed in the rebuttal section above. We hope that our clarifications and revisions have adequately resolved the concerns raised.
>
> Given the approaching deadlines for the author-reviewer discussion, we would greatly appreciate your prompt feedback on any remaining questions or concerns. Timely communication will ensure we can effectively address all issues and improve the quality of our work.
>
> Thank you once again for your valuable insights and guidance!
>
> Best regards,
> Submission 2210 Authors.

---

### Official Review · Reviewer_rjVJ · 2025-07-05

**Clarity:** 3
**Significance:** 2
**Originality:** 3
**Rating:** 5
**Confidence:** 4

**Summary:**

This paper addresses the challenge of generating semantically constrained adversarial examples (SemanticAEs) from natural language instructions, which suffer from instruction uncertainty due to referring diversity, descriptive incompleteness, and semantic boundary ambiguity. The authors introduce the InSUR framework (Instruction Uncertainty Reduction), which combines a novel ResAdv-DDIM sampler for stabilizing adversarial optimization, context-encoded constraints for adapting to incomplete instructions in both 2D and 3D settings, and a semantic-abstracted evaluation using label taxonomies. On the ImageNet abstracted label evasion task, the proposed method achieves a 55.4% attack success rate (ASR) with the ViT-B surrogate model, outperforming prior approaches in transferability and semantic fidelity (Table 1).

**Questions:**

- In Eq. 4, how sensitive is the ResAdv-DDIM sampler to the choice of k?
 - In Eq. 6, could you clarify how the thresholds ξ₁ = 0.1 and ξ₂ = 0.01 were chosen? Were these hyperparameters tuned for attack performance, or are they fixed across all tasks/models?
 - In Line 192–193, the author’s mention “M is the guidance masking that regularizes the ....". It is unclear how M is defined.
 - In Table 2, given that generation times vary widely (e.g., up to 59s), is most of this cost from the Gaussian splatting renderer, diffusion steps, or EoT sampling? Can you attribute bottlenecks quantitatively?

**Ethical Concerns:**

["NO or VERY MINOR ethics concerns only"]

**Final Justification:**

The reviewer read through the comments of other reviewers and the rebuttal and found the rebuttal to be satisfactory. Hence, the reviewer is updating the rating to "Accept".

**Limitations:**

yes (although provided in the Conclusion section)

**Paper Formatting Concerns:**

None the reviewer could find.

**Quality:**

3

**Strengths And Weaknesses:**

**Strengths:**
 - The reviewer likes the approach of InSUR that introduces a multi-dimensional approach combining residual-based diffusion guidance, context-aware 2D/3D generation, and semantic evaluation via label taxonomies, offering a robust pipeline for reference-free, instruction-guided 3D adversarial generation.
 - The proposed method significantly outperforms prior methods (SD-NAE, VENOM, AdvDiff). For example, on the abstracted label evasion task with ViT-B surrogate, the proposed method achieves 55.4% ASR versus 33.6% for SD-NAE and 40.3% for VENOM, while maintaining high perceptual quality (e.g., LPIPS of 0.039).
 - The paper is clear and concise in writing.

----
**Weaknesses:**
 - While the paper evaluates transferability across six standard architectures (e.g., ResNet50, ViT-B/16, ConvNeXt), it does not test robustness against, foundation models like CLIP (despite using CLIP-IQA as a metric), as   semanticAEs are meant to probe misalignment in general perception models. The reviewer feels that restricting evaluation to classification networks undercuts the paper's broader alignment claims.
 - In the 3D pipeline, the choice to freeze the structured latent (z₀^slat) is heuristic. There’s no ablation to show whether optimizing over it would help or hurt performance.
 - Given that instruction-driven generation is central, it’s surprising that prompt attacks on vision-language systems like [1] ([13] in the paper) are not used as baselines.
 - Typos
    - Line 39: "SemanticAEhas" → should be "SemanticAE has" (missing space).
    - Line 151: ".. improve the approximation..." → should be "by improving".
    - Line 203: "Senario knowledge..." → should be "Scenario knowledge".
    - Line 298: "Tabel 2" → should be "Table 2".
----
[1] An LLM can Fool Itself: A Prompt-Based Adversarial Attack, ICLR 2024

---

> ### Author Rebuttal · Authors · 2025-07-31
>
> We sincerely appreciate the reviewer's positive feedback on our approach and their meticulous proofreading to improve the readability. Below are our responses to the question raised.
>
> **Response to Weakness 1: limited evaluation on foundation models.**
>
> We thank the reviewer for the comment. While our primary evaluation focuses on standard architectures, we acknowledge the importance of testing on foundation models, given the paper’s emphasis on semantic alignment. To address this, we have conducted additional experiments on VQA tasks, as detailed in our response to Reviewer t5zq, with results presented at the end of that reply.
> Furthermore, in Appendix D.4 (provided in the supplementary material), we show that our generated adversarial examples effectively perturb the CLIP-VQA evaluation metric itself, indicating that semantic assessment based on CLIP is also vulnerable to such attacks. This highlights a key challenge: evaluating adversarial examples becomes difficult when the evaluator model is not robust.
> To mitigate this dependency, we propose the $ASR_\mathrm{relative}$ metric, which reduces reliance on absolute semantic evaluation by using relative performance comparison. We believe these findings are valuable for research on alignment and even *super-alignment*, as they reveal vulnerabilities not only in perception models but also in the metrics used to assess them.
>
> ---
> **Response to Weakness 2: freeze the structured latent (z₀^slat) is heuristic**
>
> Our choice of freezing the structured latent is not purely heuristic.  The reasons are:
> 1. $z_0^{slat}$ encodes the 3D positioning data, while the accurate positioning is also rectified by $z_0$ in the *Trellis* framework. We could achieve the positioning disturbance also by optimizing $z_0$.
> 2. Estimating the adversarial feedback from the surrogate model to  $z_0^{slat}$ is not computationally efficient, since it shall undergo the entire denoising diffusion process.
> 3. The computation from $z_0^{slat}$ to $\mathbf{pos}$ contains a quantization operation, which is non-differentiable.
>
> Furthermore, $z_0^{slat}$ is an internal data structure of Trellis, and the optimization method customized specifically for this structure will be constrained by the *Trellis* framework. Besides, even with z₀^slat frozen, we still achieve favorable optimization results. We believe such optimization shall be developed together with harder attack tasks.
>
> ---
> **Response to Weakness 3: limited evaluation on prompt-based attacks.**
>
>
> Thanks for the comment. As stated in Section 3.1, our work focuses on a fixed instruction: *how to generate an image that satisfies the semantic constraints of that instruction while being adversarial*. We acknowledge that perturbing the instruction itself (i.e., prompt attacks) can also lead to adversarial effects. However, most existing prompt attacks are not designed for tasks similar to SemanticAE.
>
> Our baseline, **SD-NAE**, performs attacks by perturbing the prompt embedding, making it approximately equivalent to prompt attack methods, but more aligned with our task objective. We will include a detailed description of SD-NAE in the Appendix, and ensure it is appended to the main paper in the camera-ready version.
>
> Moreover, we discuss this distinction in Appendix A.2. Specifically, generating hard samples from language can be approached via two paradigms:
> (1) *Perturbing the input text without altering the language-guided generation model*.
> (2) *Altering the generation process by introducing adversarial capabilities into the model*.
> Our work focuses on the latter. In practice, these two approaches can be combined into a unified pipeline. Our evaluation could be regarded as assessing the effectiveness of our method as a *module* within the red-team framework.
>
> We further tested a recent method that uses prompt optimization to generate adversarial images [1]. In our local single-GPU setup, it took **several hours** to find a single adversarial prompt, which is thousands of times slower than our method and the baselines we compare against. This performance gap significantly limits its applicability in scalable red-teaming systems, and it shall be deployed in different parts of the system compared to our method.
>
>
> [1] Zhu, Xiaopei, et al. "Natural language induced adversarial images." *Proceedings of the 32nd ACM International Conference on Multimedia*. 2024.
>
> ---
>
> **Q1**: In Eq. 4, how sensitive is the ResAdv-DDIM sampler to the choice of $k$?
>
> **R1**: We conducted ablation studies comparing different $k$ value settings (Iter$_\mathrm{max}$ in Table 3 of the main paper). We will clarify it to make it clearer. As $k$ increases, attack effectiveness gradually improves, but so does computational cost. In practice, $k$ can be adjusted based on the specific application. Furthermore, following the suggestion from Reviewer vVyk, we provide additional analysis on the stability of adversarial optimization under semantic guidance instructions with varying ambiguity.
>
> ---
>
> **Q2**:  In Eq. 6, could you clarify how the thresholds $\xi_1 = 0.1$ and $\xi_2 = 0.01$ were chosen? Were these hyperparameters tuned for attack performance, or are they fixed across all tasks/models?
>
> **R2**: They are fixed across all tasks and models. In our framework, $\mathrm{Confidence} < \xi_1$ indicates that the "residual approximation" predicts the attack performance will be sufficiently strong, so the inner optimization at the current denoising step is terminated and the process proceeds to the next denoising step. When $\mathrm{Confidence} < \xi_2$, it means the predicted attack performance already exceeds the task requirement—further optimization is unnecessary, and even if subsequent denoising steps improve naturalness at the cost of reduced attack strength, this is acceptable. These conditions reduce the expected number and lower bound of optimization steps. In Appendix B.1, we detail analyze the benefits of this design on generation efficiency (Proposition B.1 and related discussions).
>
> We chose these threshold values heuristically and used them consistently across all tasks without manual tuning for attack performance or computational speed.
>
> Below, we present results using ResNet50 as the surrogate model on the Abstract Label Evasion Task. In white-box attacks, all tested threshold values achieve strong performance (accuracy after attack < 3%). In black-box attacks, smaller $\xi_1$ values yield higher transferability, because lower $\xi_1$ leads to more aggressive adversarial optimization in early denoising steps—optimization that tends to be more transferable (this is consistent with the design principle of ResAdv-DDIM, which aims to enhance early-step optimization). However, this also increases computational cost. A practical deployment could balance effectiveness and efficiency.
>
> |$ξ_1$|$ξ_2$|Resnet50-ACC$\downarrow$|ViT-B/16-ACC$\downarrow$|
> |-|-|-|-|
> |0.15|0.02|0.0224|0.2949|
> |0.15|0.01|**0.0192**|0.2885|
> |0.15|0.005|0.0214|0.2885|
> |0.1|0.02|0.0246|0.2853|
> |0.1|0.01|0.0246|0.2724|
> |0.1|0.005|0.0246|0.2714|
> |0.05|0.02|0.0224|0.2404|
> |0.05|0.01|0.0256|**0.2382**|
> |0.05|0.005|0.0246|0.2511|
>
> **Q3:** In Line 192–193, the authors mention “M is the guidance masking that regularizes the ....". It is unclear how M is defined.
>
> **R3:** The definition of $M$ is explained in Appendix B.1 (in the supplementary material). We will include the appendix after the main text in the camera-ready version and add a reference to it after this sentence. Specifically,
> $$
> \begin{equation}
>     M_{ij} := \begin{cases}
> M_\mathrm{mid} & \frac{h}{16} \le i < \frac{15h}{16}, \frac{w}{16} \le j < \frac{15w}{16}, \\\\
> M_\mathrm{edge} & \text{otherwise}.
>     \end{cases}
> \end{equation}
> $$
> In the main experiments, we set $M_\mathrm{mid} = 3.0$ and $M_\mathrm{edge} = 0.3$. We further test $M_\mathrm{edge} = \{0.0, 0.3, 3.0\}$ in the ablation study.
>
> ---
>
> **Q4:** In Table 2, given that generation times vary widely (e.g., up to 59s), is most of this cost from the Gaussian splatting renderer, diffusion steps, or EoT sampling? Can you attribute bottlenecks quantitatively?
>
> **R4:**  The variation in generation time arises from the adaptive number of optimization steps and differences in the size of sparse tensors in the Trellis framework. Attributed to the proposed early stopping mechanism, easier attack cases require fewer iterations and thus less time. The large variance in generation time reflects significant differences in attack difficulty across tasks, which highlights the effectiveness of our adaptive step design.
>
> We further conducted a timing analysis on a single GPU using 100 labels from the Imagenet-Label dataset. The results are shown below. Gradient computation and residual approximation account for the majority of the computational cost. Meanwhile, *EoT sampling* is implemented via multi-view rendering using the Gaussian splatting renderer, while the rendering parameter computation does not occupy CUDA execution time.
>
> For the single adversarial optimization iteration:
>  - Denoise Sampling: 123.321 (std=36.878)ms
>  -  Residual Approximation: 147.256 (std=76.243)ms
>  -  Gaussian Decoding: 21.547 (std=11.868)ms
>   - Gaussian Rendering: 28.556 (std=7.706)ms
>  -  Surrogate Model: 6.777 (std=0.745)ms
>   - Backward Process: 208.471 (std=78.039)ms
>
> The fluctuation in computation speed is primarily caused by inconsistent data sizes in the Sparse Tensor.
>
> For the entire adversarial optimization:
>   - Denoise Sampling: 2994.933 (std=269.841)ms
>  -  Residual Approximation: 7040.238 (std=8915.112)ms
>  -  Gaussian Decoding: 1029.394 (std=1185.636)ms
>  -  Gaussian Rendering: 1364.221 (std=1732.996)ms
>   - Target Model: 323.782 (std=423.503)ms
>  -  Backward Process: 6261.577 (std=12119.871)ms
>
> The fluctuation in computation speed is also attributed to the adaptive optimization steps. Specifically, the average count of optimization steps for the inner optimization is: 47.810 (std=63.566).

---

> ### Comment · Reviewer_rjVJ · 2025-08-02
>
> Hi Authors,
>
> Thank you for your hard work on the rebuttal. All concerns are addressed and the reviewer has updated the rating to "Accept". Best of luck!
>
> Regards, \
> Reviewer rjVJ

---

> > ### Author Response · Authors · 2025-08-04
> >
> > Dear Reviewer,
> >
> > Thank you for your positive feedback and the score adjustment. We appreciate your time and effort in evaluating our work.
> >
> > Best regards,
> > Submission 2210 Authors.

---

### Official Review · Reviewer_t5zq · 2025-07-12

**Clarity:** 3
**Significance:** 2
**Originality:** 3
**Rating:** 4
**Confidence:** 4

**Summary:**

This paper focuses on the topic of semantic-constrained adversarial examples. This paper develops a multi-dimensional instruction uncertainty reduction (InSUR) framework to generate more satisfactory SemanticAE. Their framework proposes residual-driven attacking direction stabilization, context-encoded attacking scenario constraint, guidance masking and renderer integration techniques to solve the uncertainty problems. Extensive experiments demonstrate the superiority of the transfer attack performance of InSUR.

**Questions:**

How to control the abstracting level of the antonyms of the Text. Cat is an antonym of dog, and car is also an antonym of dog, but they are different-level antonyms. How do you consider this kind of different-level antonyms?

How to guarantee the naturalness of the proposed adversarial attacks. We know that diffusion models can generate realistic images, but some of the details are unreal. Your approach utilizes the diffusion models, so how to guarantee the naturalness of the generated images?

I am not sure about the real use cases of semantic AE. Is the semantic AE possible in the real world or does the semantic AE have real world application?

I understand that the adversarial examples can be used for single object evaluation, but can you extend the approach for more complex scenarios in general VQA tasks, e.g., the image contains multiple objects? Can your adversarial examples be able to generate such adversarial images.

(Minor) Typos Line39: SemanticAEhas => SemanticAE has

**Ethical Concerns:**

["NO or VERY MINOR ethics concerns only"]

**Final Justification:**

After the rebuttal, I increase my score to borderline accept.

**Limitations:**

Yes

**Quality:**

3

**Strengths And Weaknesses:**

Strengths
+ Semantic AE is interesting
+ The approach is concrete

Weakness

-	The naturalness of adversarial examples cannot be guaranteed
-	The real use case is unclear
-	Different abstracting levels should be discussed.

---

> ### Author Rebuttal · Authors · 2025-07-31
>
> We sincerely appreciate your constructive feedback. Below are our responses to the questions raised.
>
> **Q1:** How to control the abstracting level of the antonyms of the Text. Cat is an antonym of dog, and car is also an antonym of dog, but they are different-level antonyms.
>
> **R1:**  We acknowledge that the term *antonyms* in the original manuscript (Section 3.1) was imprecisely used and could lead to misunderstanding. Specifically, $A_{\text{Text}}$ refers to output types that are conceptually *different* from the original text, not necessarily strict linguistic antonyms. In fact, in the untargeted attack scenario, these should be understood as *non-synonyms* (e.g., *cat* is not an antonym of *dog*, but it is a non-synonym). As in prior transfer-based attack works, our experiments focus on the untargeted setting to evaluate effectiveness.
>
> Our proposed benchmark already addressed the **evaluation bias** caused by overly fine-grained semantic abstraction. We agree with the reviewer that different abstraction levels inherently lead to different attack difficulties, as supported by prior work [1]. To account for this, we define the abstraction level of $A_{\text{Text}}$ via the scope of synonym coverage: broader synonym sets imply more distant "antonyms." In our experiments, we evaluate across both *Abstracted Labels* and *Refined Labels*—representing different abstraction levels—to comprehensively assess the performance of SemanticAE methods under varying semantic granularity.
>
> The reviewer’s suggestion, “Different abstraction levels should be discussed”, can be further addressed through finer-grained evaluation, which we support. We also agree that VQA provides a more general and flexible framework for multi-level evaluation, potentially extending to alignment studies. We present VQA results under different abstraction levels in our response to Q3.
>
> ---
> **Q2:** We know that diffusion models can generate realistic images, but some of the details are unreal. Your approach utilizes the diffusion models, so how to guarantee the naturalness?
>
> **R2:** While generating natural adversarial examples has been studied by many prior works, our core contribution lies not in naturalness, but in exploring and constructing adversarial examples in a broader semantic space, with a focus on key properties such as *transferability* and *robustness*. Hence, our title emphasizes *Semantically Constrained Adversarial Examples*, not *Natural* ones.
>
> In our technology contribution, diffusion models are primarily used for *on-manifold regularization* to enhance transferability. Nevertheless, we do incorporate naturalness constraints by adapting existing techniques to ResAdv-DDIM. Specifically, we constrain the difference between non-adversarial and adversarial denoising steps during diffusion sampling (as shown in Eq. 7):
> $$
> \|\|\textrm{Denoise}\_\textrm{DDIM}(x\_{t\_s - \Delta T})-\textrm{Denoise}\_\textrm{Adv}(x\_{t\_s - \Delta T})\|\|\_2 < \epsilon
> $$
> Both visualizations and quantitative results confirm that our generated samples exhibit sufficient naturalness.
>
> Moreover, we argue that the loose realistic constraint does not challenge the significance of the study, specifically:
> 1.  **From an attack system perspective**, if our method is used as a module in a real system, generated samples can be filtered using semantic plausibility checks. If both generator $A$ and discriminator $B$ (not under attack) deem an image $X$ semantically valid, we have strong justification to treat $X$ as natural. We formalize this in Appendix B.3 via a proposition linking our evaluation metric to practical utility. In real attack scenarios—whether physical attacks or jailbreaks—the targets are often automated systems; our evaluation aligns well with modeling such defenses. We further analyze attack performance under defenses in Appendix D.
> 2. **For adversarial training**, recent work [2] (ICML 2025, oral) shows that diffusion-generated samples—even with minor imperfections—can significantly improve training efficiency and model performance, enhancing scaling laws. Thus, minor realism flaws do not undermine their utility. We detailed this work in our response to Q3.
>
> The *realism* of generated images is more central to diffusion model alignment research. Our framework may also inform *Test-Time Scaling of Diffusion Model Alignment*. While recent works[3] use reward models to refine generation, we explore the inverse: what optimization structures induce robustness failures in such reward models?
>
> ---
> **Q3:** Is the semantic AE possible in the real world or does the semantic AE have real world application? I understand that the adversarial examples can be used for single object evaluation, but can you extend the approach for more complex scenarios in general VQA tasks, e.g., the image contains multiple objects? Can your adversarial examples be able to generate such adversarial images.
>
> **R3:**  Attributed to the technical improvement on the key problems, our framework has broad applications:
> 1. **Adversarial training**: Our framework can adapt to a concurrent ICML 2025 [2] work proposing diffusion-based data generation for training, which uses the target model $f_\phi$’s prediction entropy $H$ to guide diffusion sampling to generate challenging samples:
> $$\textrm{Guidance Signal} \propto \omega \nabla_{x_t} H(f_\phi(\hat{x}_{0,t}))$$ Our **ResAdv-DDIM** can enhance this pipeline by stabilizing multi-step adversarial directions through a refined estimation of $\hat{x}\_{0,t}$. This improves sampling accuracy and speed, while offering new insights into transferability. Since high-transferability adversarial examples expose non-robust features shared across models, they may offer advantages in adversarial training. We will include this discussion in the revised manuscript.
>
> 2. **Jailbreak and physical attacks**: We have already demonstrated effectiveness in cross-domain settings (2D → 3D). In complex systems, *inaccurate adversarial feedback* is a key challenge. The success of our method in 3D scenarios—where multi-view feedback is often incomplete—shows it can handle such uncertainty. We elaborate further in our response to Reviewer vVyk.
>
> 3. **Extension to complex VQA tasks**: Both the generator and the surrogate model are replaceable in our framework. Our 3D generation experiment validates the cross-task extension capability. For images with multiple objects, one can simply include multiple object descriptions in the diffusion model’s prompt. For applications in autonomous driving, our framework could guide the adversarial data generation techniques developed based on the scene generation diffusion models in the simulation pipeline.
>
> We conducted additional VQA experiments using OpenAI CLIP models {RN50, ViT-B/32, RN50x16} (with DeCoWA) as surrogates, generating adversarial examples under three different *abstraction levels*. For each answer option, InSUR generates samples satisfying the corresponding semantic constraint $\mathcal{S}_\text{Text}$ ($\text{Text} \in \{\text{Options}\}$), using stabilityai/stable-diffusion-2-1-base. We evaluate transferability across OpenAI CLIP variants (RN50, RN101, RN50x4, RN50x16, ViT-B/32, ViT-B/16) and large VLMs (LLaVA, Qwen 7B). Results show:
> - InSUR achieves strong cross-model transferability and enables weak surrogates to attack stronger models via diffusion assistance, though transfer weakens with greater semantic distance.
> - Due to the widespread adoption of ViT architectures in modern VLMs, ViT-based surrogates show stronger transfer to LLaVA and Qwen.
>
> We believe these findings are valuable for both alignment and *super-alignment* research. We will add the discussion to the camera-ready version.
>
> ---
>  **Experimental Results:**
>
>
> **Question:** "What brand is this car?", **Options:** ["Audi", "Mercedes", "Ferrari", "Toyota"]
>
> **Accuracy(%) After Attack ($\downarrow$)**
>
> |Surrogate|RN50|RN101|RN50x4|RN50x16|ViT-B/32|ViT-B/16|Llava1.5-7B|Qwen2.5VL-7B|
> | - | - | - | - | - | - | - | - | - |
> | RN50 | 2.5| 30.0| 37.5| 47.5| 55.0| 50.0| 67.5| 70.0  |
> | ViT-B/32  | 32.5 | 37.5| 45.0| 65.0| 0.0 | 40.0| 57.5   | 32.5  |
> | RN50x16 | 47.5 | 52.5| 37.5| 5.0| 70.0| 65.0| 67.5   | 57.5  |
>
> **Question:** "What is in the picture:", **Options:** ["Police car", "Ambulance", "Taxi", "School Bus"]
>
> **Accuracy(%) After Attack ($\downarrow$)**
>
> |Surrogate|RN50|RN101|RN50x4|RN50x16|ViT-B/32|ViT-B/16|Llava1.5-7B|Qwen2.5VL-7B|
> | - | - | - | - | - | - | - | - | - |
> | RN50|0.0| 32.5  | 45.0 | 50.0  | 40.0| 52.5| 70.0 | 60.0 |
> | ViT-B/32| 10.0 | 7.5  | 17.5 | 17.5 | 0.0 | 10.0| 25.0| 12.5|
> | RN50x16| 30.0 | 40.0 | 32.5| 0.0| 55.0| 52.5| 65.0| 57.5|
>
> **Question:** "What is in the picture:", Options: ["Air Liner", "Car", "Helicopter", "Cruise Ship"]
>
> **Accuracy(%) After Attack ($\downarrow$)**
>
> |Surrogate|RN50|RN101|RN50x4|RN50x16|ViT-B/32|ViT-B/16|Llava1.5-7B|Qwen2.5VL-7B|
> | - | - | - | - | - | - | - | - | - |
> | RN50 | 5.0  | 70.0  | 72.5 | 92.5 | 92.5| 85.0| 90.0| 82.5|
> | ViT-B/32  | 47.5 | 65.0  | 67.5| 77.5 | 0.0 | 62.5| 70.0 | 67.5 |
> | RN50x16 | 60.0 | 67.5  | 75.0| 10.0| 85.0| 92.5| 92.5 | 87.5 |
>
> **Overall Accuracy After Attack(%)($\downarrow$)**
>
> |Surrogate|RN50|RN101|RN50x4|RN50x16|ViT-B/32|ViT-B/16|Llava1.5-7B|Qwen2.5VL-7B|
> | - | - | - |-|-| - |-|-| - |
> | RN50 | 2.5  | 44.2| 51.7| 63.3| 62.5 | 62.5 | 75.8| 70.8 |
> | ViT-B/32| 30.0 | 36.7| 43.3| 53.3| 0.0| 37.5 | 50.8| 37.5 |
> | RN50x16| 45.8 | 53.3| 48.3| 5.0 | 70.0 | 70.0 | 75.0| 67.5 |
>
> [1] Wang, Jiakai, et al. "Generate transferable adversarial physical camouflages via triplet attention suppression." *International Journal of Computer Vision* 132.11 (2024): 5084-5100.
>
> [2] Askari-Hemmat, Reyhane, et al. "Improving the Scaling Laws of Synthetic Data with Deliberate Practice." arXiv preprint arXiv:2502.15588 (2025).
>
> [3] Jain, Vineet, et al. "Diffusion Tree Sampling: Scalable inference-time alignment of diffusion models." *arXiv preprint arXiv:2506.20701* (2025).

---

> > ### Comment · Reviewer_t5zq · 2025-08-05
> >
> > Thanks for the rebuttal, and my concerns are addressed. I will increase my scoring.

---

> ### Author Response · Authors · 2025-08-04
>
> Dear Reviewer,
>
> We sincerely appreciate the time and effort you have dedicated to evaluating our work. We have carefully addressed all the comments and feedback provided in the review, and the corresponding responses are detailed in the rebuttal section. We hope that our clarifications and revisions have adequately resolved the concerns raised.
>
> Given the approaching deadlines for the author-reviewer discussion, we would greatly appreciate your prompt feedback on any remaining questions or concerns. Timely communication will ensure we can effectively address all issues and improve the quality of our work.
>
> Thank you once again for your valuable insights and guidance!
>
> Best regards,
> Submission 2210 Authors.

---

### Note · Authors · 2025-08-11

We sincerely thank reviewers for their thoughtful feedback. This note summarizes the reviews, rebuttals, and discussions to aid the final assessment.

---

Most reviewers recognized the paper’s technical solidity and novelty, clarity of writing, and experimental significance. There was minor disagreement between Reviewers FdBW and vVyk regarding whether the 3D generation method represents a "major step forward" or "relies on existing tools." We discussed it in our response to vVyk. Moreover, we believe that the successful 2D-to-3D extension demonstrates our framework’s, especially the proposed ResAdv-DDIM's, cross-task and cross-generator extensibility, showing the capability to address diverse application scenarios.

The main concerns came from Reviewer t5zq on application scenarios, insufficient experiments on VLM alignment  (Reviewers t5zq & rjVJ) and physical attack applications  (Reviewer vVyk), the need for further algorithmic analysis (Reviewer vVyk), and issues on algorithm parameters and design details. We have addressed these by clarifying relevant parts in the paper, discussing related literature for the application scenarios, and supplementing additional experiments on VQA tasks, more parameter studies, and new analysis experiments. We emphasize that these are supplementary improvements, as three out of four reviewers affirmed that the core contribution in the original submission was already experimentally well-validated. We also fully acknowledge the value of these additions in enhancing the paper’s impact and will include them in the final version.

In summary, all active reviewers (except Reviewer FdBW’s "ack") confirmed full resolution of concerns, and will score $≥4$. Supplementary enhancements (VQA/parameter/analysis experiments) bolster impact without altering foundational validity. We thank the committee for this rigorous process and gratefully accept the decision.

---

### Decision · Program_Chairs · 2025-09-17

**Decision:**

Accept (poster)

**Comment:**

The paper proposes the InSUR framework for generating semantically constrained adversarial examples and addresses instruction uncertainty through residual-driven direction stabilization, context-encoded constraints and semantic-abstracted evaluation.
The reviewers generally agree that the paper is technically solid, novel in framing instruction uncertainty, and supported by extensive experiments in both 2D and 3D settings.
Furthermore, the rebuttal addressed reviewer concerns with additional experiments (including VQA and CLIP-based evaluations), parameter studies, and clarifications on the role of ResAdv-DDIM, Trellis integration and guidance masking.
After rebuttal and discussion, the consensus leans positive.
The work makes a meaningful contribution to the field of semantic adversarial examples.